# Residual ANTXR1+ myofibroblasts after chemotherapy inhibit anti-tumor immunity via YAP1 signaling pathway

Monika Licaj[1,2,11], Rana Mhaidly[1,2,11], Yann Kieffer [1,2], Hugo Croizer[1,2], Claire Bonneau [1,2,3], Arnaud Meng[1,2], Lounes Djerroudi[1,2,4], Kevin Mujangi-Ebeka[1,2], Hocine R. Hocine[1,2], Brigitte Bourachot [1,2], Ilaria Magagna[1,2], Renaud Leclere[4], Lea Guyonnet[5], Mylene Bohec [6], Coralie Guérin [5], Sylvain Baulande [6], Maud Kamal[7], Christophe Le Tourneau [7,8], Fabrice Lecuru[9], Véronique Becette[10], Roman Rouzier[3], Anne Vincent-Salomon [4], Geraldine Gentric [1,2,12] ✉ & Fatima Mechta-Grigoriou [1,2,12] ✉

Although cancer-associated fibroblast (CAF) heterogeneity is well-established, the impact of chemotherapy on CAF populations remains poorly understood. Here we address this question in high-grade serous ovarian cancer (HGSOC), in which we previously identified 4 CAF populations. While the global content in stroma increases in HGSOC after chemotherapy, the proportion of FAP+ CAF (also called CAF-S1) decreases. Still, maintenance of high residual CAF-S1 content after chemotherapy is associated with reduced CD8+ T lymphocyte density and poor patient prognosis, emphasizing the importance of CAF-S1 reduction upon treatment. Single cell analysis, spatial transcriptomics and immunohistochemistry reveal that the content in the ECM-producing ANTXR1+ CAF-S1 cluster (ECM-myCAF) is the most affected by chemotherapy. Moreover, functional assays demonstrate that ECM-myCAF isolated from HGSOC reduce CD8+ T-cell cytotoxicity through a Yes Associated Protein 1 (YAP1)-dependent mechanism. Thus, efficient inhibition after treatment of YAP1-signaling pathway in the ECM-myCAF cluster could enhance CD8+ T-cell cytotoxicity. Altogether, these data pave the way for therapy targeting YAP1 in ECM-myCAF in HGSOC.

Epithelial ovarian cancers represent one of the deadliest gynecologic cancers that are classified according to histological subtypes, grade and stage. High-grade serous ovarian cancer (HGSOC) accounts for 75% of total ovarian cancers. Most often, the disease progresses silently in the peritoneal cavity until an advanced stage and is associated with poor prognosis. Patients with advanced HGSOC receive a combination of platinum- and taxane-based chemotherapy prior to surgery[1]. Standard treatments also include targeted therapies, such as anti-angiogenic drugs and PARP

inhibitors[2–10]. However, despite all these treatments, more than 70% of patients still relapse.

HGSOC are complex ecosystems composed of tumor cells and of various other cell types, such as Cancer-Associated Fibroblasts (CAF), immune and endothelial cells embedded in an extracellular matrix (ECM)[11–19]. Tumor micro-environment (TME) became a major focus in new therapeutic options through the blockade of tumor vasculature and the recent development of immune checkpoint inhibitors[20–22]. Despite promising activity, these treatments have not been shown to

improve overall survival of HGSOC patients, possibly because HGSOC molecular subtypes have not yet been taken into account. Indeed, recent studies have demonstrated the existence of distinct HGSOC molecular entities based on multi-omics features[19,23–39]. Interestingly, the mesenchymal molecular subtype of HGSOC has been identified in all studies and is systematically associated with poor prognosis. Mesenchymal HGSOC exhibit tumor cells with mesenchymal features and enrichment in specific myofibroblastic CAF populations[25,30,40,41]. In particular, single cell RNA sequencing analyses (scRNAseq) show that the mesenchymal subtype of HGSOC reflects the abundance of fibroblasts rather than distinct subsets of malignant cells[42–46]. Four different CAF populations (referred to as CAF-S1 to -S4) have been recently identified in HGSOC by combining different CAF markers, including Fibroblast Activation Protein (FAP), Actin Alpha 2 Smooth Muscle (ACTA2/SMA), Integrin β1 (ITGB1/CD29) and Fibroblast Specific Protein 1 (FSP1)[40]. The myofibroblastic CAF-S1 (FAP$^{Pos}$ CD29$^{Med}$ SMA$^{Pos}$ FSP1$^{Med-High}$) and the pericyte-like CAF-S4 (FAP$^{Neg}$ CD29$^{High}$ SMA$^{High}$ FSP1$^{Med}$) are strictly detected in tumors, but not in healthy tissues, and are enriched in mesenchymal HGSOC[40]. These CAF have also been detected in several cancer types and in different species by using various methods, including single cell analysis[18,43,47–59]. SMA$^+$ CAF are well-known to promote metastases, especially in breast and ovarian cancer[40,55,60–73]. In line with these findings, both CAF-S1 and CAF-S4 enhance metastatic spread by acting on tumor cells and the surrounding ECM, respectively[68]. CAF-S1 (or FAP$^+$ CAF) have also been associated with an immunosuppressive environment in various tumor types[40,43,49,55,69,73–85]. Specifically, CAF-S1 fibroblasts have been shown to promote immunosuppression by increasing CD4$^+$ CD25$^+$ FOXP3$^+$ regulatory T lymphocyte (Treg) content and T cell dysfunction and to contribute to immunotherapy resistance in mouse and human cancers[40,47,48,50,51,53,76,80,82,86,87]. Consistent with the pro-metastatic and immunosuppressive functions of the CAF-S1 population, single cell data of CAF-S1 from cancer patients recently highlighted that this population is composed of 8 distinct cellular clusters[82]. Indeed, among the CAF-S1 (or FAP$^+$ SMA$^+$ CAF) population, we distinguish ANTXR1$^+$ and ANTXR1$^-$ CAF-S1. ANTXR1$^+$ CAF-S1 are positive for the ANTXR cell adhesion molecule 1 (ANTXR1) marker and enriched in myofibroblasts (myCAF), as ANTXR1$^+$ CAF-S1 express high levels of FAP and SMA (FAP$^{High}$ SMA$^{High}$). In contrast, ANTXR1$^-$ CAF-S1, which are negative for the ANTXR1 marker, are positive but express low to medium levels of FAP and SMA (FAP$^{Low-Med}$ SMA$^{Low}$) and are mainly inflammatory (iCAF). Among the ANTXR1$^+$ CAF-S1, the ECM-myCAF cluster is the most abundant one in tumors before treatment and is associated with primary resistance to immunotherapy, while the ANTXR1$^-$ CAF-S1 clusters are not[82]. Consistent with these findings, the ECM-myCAF cluster, which is also positive for the LRRC15 marker, is of particular relevance, as its genetic ablation in PDAC mouse models enhances the antitumor immunity of cytotoxic T cells and improves response to immune checkpoint blockade[88].

Although the role of CAF in chemoresistance is well established as a global population[89–95], the role of CAF subsets on chemotherapy response in patients has not been extensively studied. Reciprocally, the impact of chemotherapy on the different CAF populations has not yet been investigated in HGSOC. Here, we aim to fulfill this lack by addressing these questions. After chemotherapy in HGSOC patients, the global stromal content increases, concomitantly to epithelium reduction. Yet, the proportion of SMA + CAF populations (i.e., CAF-S1 and CAF-S4 populations) decreases in HGSOC following treatment. Interestingly, platinum-resistant patients with a high residual content in CAF-S1 and CAF-S4 myofibroblasts after chemotherapy survive less than patients with normal-like (CAF-S2 and CAF-S3) fibroblasts. Consistent with this observation, we found that the more the CAF-S1 content decreases following chemotherapy, the more the CD8$^+$ T lymphocyte density increases. Given the potential role of CAF-S1 in T cell content after chemotherapy, we next performed comparative

analyses of the CAF-S1 population by single-cell RNA sequencing and spatial transcriptomics from HGSOC samples before and after treatment. These analyses demonstrate that the content in the ANTXR1$^+$ CAF-S1 cluster characterized by high expression of ECM genes (hereinafter referred to as ECM-myCAF) decreases the most after chemotherapy, while the proportion of ANTXR1$^-$ inflammatory CAF-S1 (iCAF) clusters increases. Moreover, functional assays using ECM-myCAF primary fibroblasts isolated from HGSOC patients show that ECM-myCAF significantly decrease CD8$^+$ T cell cytotoxicity, revealing the direct immunosuppressive impact of the ECM-myCAF cluster on CD8$^+$ T lymphocytes, in addition to the previously reported effect of CAF-S1 on Tregs[40,80]. We identify that the mechanism mediated by the ECM-myCAF cluster on CD8$^+$ T lymphocytes is dependent on the Yes Associated Protein 1 (YAP1) co-transcription factor. Indeed, the elevated YAP1 nuclear protein level observed in ECM-myCAF at baseline in untreated HGSOC patients significantly decreases after chemotherapy. Moreover, YAP1 silencing in ECM-myCAF promotes CD8$^+$ T lymphocyte cytotoxicity, consistent with observations in patients. Thus, we show here that the decrease in ECM-myCAF content and subsequent YAP1 down-regulation after chemotherapy is associated with an increase in CD8$^+$ T lymphocyte density, suggesting that targeting YAP1 in the stroma might be a promising therapeutic avenue to favor CD8$^+$ T lymphocyte enrichment in HGSOC patients.

## Results

### Chemotherapy reshapes CAF heterogeneity in HGSOC

To evaluate the impact of standard (platinum salts- and taxanes-based) chemotherapy on TME components, we measured the content in CAF and T lymphocytes before and after chemotherapy in HGSOC patients. To do so, we first constituted a retrospective cohort of HGSOC patients with available paired samples before (at time of diagnosis) and after chemotherapy (see Table 1, Retrospective Curie 1 cohort). At diagnosis, patients had an average age of 67.6 years old (ranging from 49 to 81 years). All patients underwent initial diagnostic surgery, with sampling prior to treatment, followed by chemotherapy. 60% of patients did not relapse in the 6 months after chemotherapy responding to the actual definition of platinum-sensitive, and 40% were platinum-resistant. These different features have been previously reported in other HGSOC cohorts[37,90,93,96–98], confirming that the HGSOC cohort studied here was a representative HGSOC cohort well-adapted for comparing samples before and after chemotherapy.

We first evaluated the response to chemotherapy in each HGSOC patient by comparing epithelial and stromal content before and after chemotherapy (Fig. 1A–D). Tumor epithelium content was first assessed based on morphological criteria (Fig. 1A, C). As expected, most HGSOC exhibited a lower epithelial content after chemotherapy compared to the corresponding samples at time of diagnosis (Fig. 1A, C). We next wondered whether this global decrease in the epithelium content could be associated with a lack of epithelial features by performing immunohistochemistry (IHC) staining of EPCAM, a well-known epithelial marker (Fig. 1B, D). Although a small number of patients exhibited a decrease in EPCAM staining, epithelial cancer cells were EPCAM+ before treatment and the intensity of the staining remained high after chemotherapy in most patients (Fig. 1D), indicating that residual epithelial cells after chemotherapy expressed EPCAM at the same levels as before treatment. In contrast to the global reduction of the epithelium, the stromal content significantly increased after chemotherapy (Fig. 1E). We thus next compared CAF populations in HGSOC before/after treatment by performing IHC analyses on serial sections of HGSOC samples (Fig. 1F–I). To do so, we combined the analyses of different CAF markers, including FAP, CD29, SMA and FSP1 (see Supplementary Table S1 for antibody references) for differentiating the four CAF populations (referred to as CAF-S1 to CAF-S4) that we previously identified in several cancer types, including ovarian cancer[40,67,68,80]. We evaluated the histological score (H-score,

## Table 1 | Main patient characteristics and clinic-pathological features of HGSOC retrospective cohorts

| | Retrospective Curie 1 cohort (IHC) | Retrospective Scandare Curie 2 cohort (RNAseq) |
|---|---|---|
| **Total number of patients** | 35 with matched samples before/after treatment | 48 |
| **Date of diagnostic** | 2000–2014 | Inclusions as from 2017 |
| **Age of the patients (years)** | | |
| Median age | 67,6 | 67,5 |
| Range | 49–81 | 32–86 |
| **Histotype** | | |
| Serous | 35 (100%) | 42 (87.5%) |
| Other or NA | 0 (0%) | 6 (12.5%) |
| Grade | | |
| High | 35 (100%) | 41 (85.4%) |
| Low | 0 (0%) | 5 (10.4%) |
| NA | 0 (0%) | 2 (4.2%) |
| **FIGO stage** | | |
| I | 0 (0%) | 2 (4.2%) |
| II | 0 (0%) | 1 (2.1%) |
| III | 27 (77%) | 28 (58.3%) |
| IV | 8 (23%) | 9 (18.8) |
| NA | 0 (0%) | 8 (16.7%) |
| **Surgery (debulking)** | | |
| Partial | 14 (40%) | NA |
| Full | 21 (60%) | NA |
| Relapse | | |
| Yes | 33 (94.3%) | 29 (60.4%) |
| No | 2 (5.7%) | 17 (35.4%) |
| NA | | 2 (4.2%) |
| **Metastasis** | | |
| Yes | 7 (20%) | NA |
| No | 26 (74.3%) | NA |
| NA | 2 (5.7%) | NA |
| Levels of CA-125 before surgery | | |
| ≤ 65 | 27 (77.1%) | NA |
| > 65 | 7 (20%) | NA |
| NA | 1 (2.9%) | NA |
| BRCAness status | | |
| BRCA1 | 3 (8.6%) | 4 (8.3%) |
| BRCA2 | 2 (5.7%) | 2 (4.2%) |
| No mutation | 11 (31.4%) | 1 (2.1%) |
| NA | 19 (54.3) | 41 (85.4%) |
| **Resistance to carboplatin** | | |
| Sensitivity | 21 (60%) | NA |
| Resistant | 14 (40%) | NA |
| **CT Scan response after NAC** | | |
| Complete response | NA | 1 (2.1%) |
| Partial response | NA | 28 (58.3%) |
| Progression disease | NA | 1 (2.1%) |
| Stable disease | NA | 6 (12.5%) |
| NA | NA | 12 (25%) |
| **Delay of relapse (months)** | | |
| Median | 7,1 | 18,7 |
| Range | 0–49.5 | 1.2–37.7 |

## Table 1 (continued) | Main patient characteristics and clinic-pathological features of HGSOC retrospective cohorts

| | Retrospective Curie 1 cohort (IHC) | Retrospective Scandare Curie 2 cohort (RNAseq) |
|---|---|---|
| **Neoadjuvant chemotherapy** | | |
| Including Carboplatin and taxol | 35 (100%) | 39 (81.3%) |
| Other | 0 (0%) | 1 (2.1%) |
| NA | 0 (0%) | 8 (16.7%) |
| **Adjuvant chemotherapy** | | |
| Including Carboplatin and taxol | 18 (51.4%) | 31 (64.6%) |
| Carboplatin only | 4 (11.4%) | 0 (0%) |
| Taxol only | 2 (5.7%) | 0 (0%) |
| Others | 6 (17%) | 9 (18.8) |
| No adjuvant chemotherapy | 4 (11.4%) | NA |
| NA | 1 (2.8%) | 8 (16.7%) |

Retrospective Curie 1 cohort: Ovarian cancer patients of the retrospective Curie 1 cohort have been treated at Institut Curie Hospital Group between 2000 and 2014. All patients had paired samples collected prior and after neoadjuvant chemotherapy, allowing comparison of tissues from the same patient. Ovarian carcinomas were classified according to the World Health Organization histological classification of gynecological tumors. All patients were suffering from tumors of serous histological subtype, high-grade and advanced stage (FIGO stages III and IV), as defined by pathological analyses and clinical/radiological staging. The median age of HGSOC patients at diagnosis was 67.6 years, with a range of 49–81 years. All subjects underwent interval debulking surgery. HGSOC debulking efficiency differentiates two subgroups, the "full debulking" patients with no visible macroscopic residual disease and the "partial debulking" patients with visible macroscopic residual disease. 60% of them had a full debulking and 40% a partial one. Most of the patients had been treated with adjuvant platinum-based chemotherapy following surgery. The majority of patients (60%) was defined as platinum-sensitive, as they did not relapse 6 months after treatment. 40% of patients was platinum-resistant. CA-125 serum level before interval debulking surgery, measured as an indicator of the size of the residual tumor was higher than 65 (double the normal level) in 23% of patients. Germline mutations in BRCA1/2 genes have been analyzed by standard sequencing. Half of HGSOC patients of the cohort has been tested for BRCA mutations. 9% of patients carried a mutation on BRCA1, 6% a mutation on BRCA2 and 31% did not carry any mutation on BRCA genes. Retrospective SCANDARE Curie 2 cohort: Inclusion started from 2017. Patients have been treated at Institut Curie Hospital Group and received standard treatment according to the stage of the disease and usual procedures. The median age at diagnosis was 67.5 years. More than 85% of the patients showed a serous histological subtype of high grade. Most of the patients had been treated with carboplatin and/or taxol-based chemotherapy. Clinical categories are indicated in bold.

combining intensity of the staining and percentages of stained cells) for each aforementioned CAF marker in the stroma (Fig. 1F, G). The H-scores of the different CAF markers tested, including FAP, SMA, CD29 and FSP1, significantly decreased in CAF upon chemotherapy (Fig. 1F, G), thereby suggesting that chemotherapy might promote a shift from activated CAF (CAF-S1/CAF-S4) to less activated CAF or normal-like fibroblasts. To verify this hypothesis, we applied a decision tree algorithm (Fig. 1H) (see Methods, #Decision tree algorithm for prediction of CAF population identity and our previous publications for further details[40,80,82]) to determine the global enrichment in each CAF population per tumor before and after treatment (Fig. 1I). Before chemotherapy, HGSOC were mainly enriched in CAF-S1 and CAF-S4 populations (Fig. 1I), confirming results described in an independent cohort of HGSOC prior to treatment[40]. In contrast, the content in both CAF-S1 and CAF-S4 significantly decreased upon chemotherapy, while the proportion of normal-like CAF-S2 and CAF-S3 increased (Fig. 1I). Indeed, while 74,3% of HGSOC exhibited CAF-S1 or CAF-S4 enrichment before treatment, this proportion dropped at 42,8% after chemotherapy (P-value = 0.01 by Fisher's Exact test). Although CAF populations at diagnosis did not predict survival, residual CAF populations after chemotherapy were significantly associated with response to chemotherapy. Indeed, platinum-resistant patients with high residual CAF-S1 or CAF-S4 content after chemotherapy survived less and relapsed

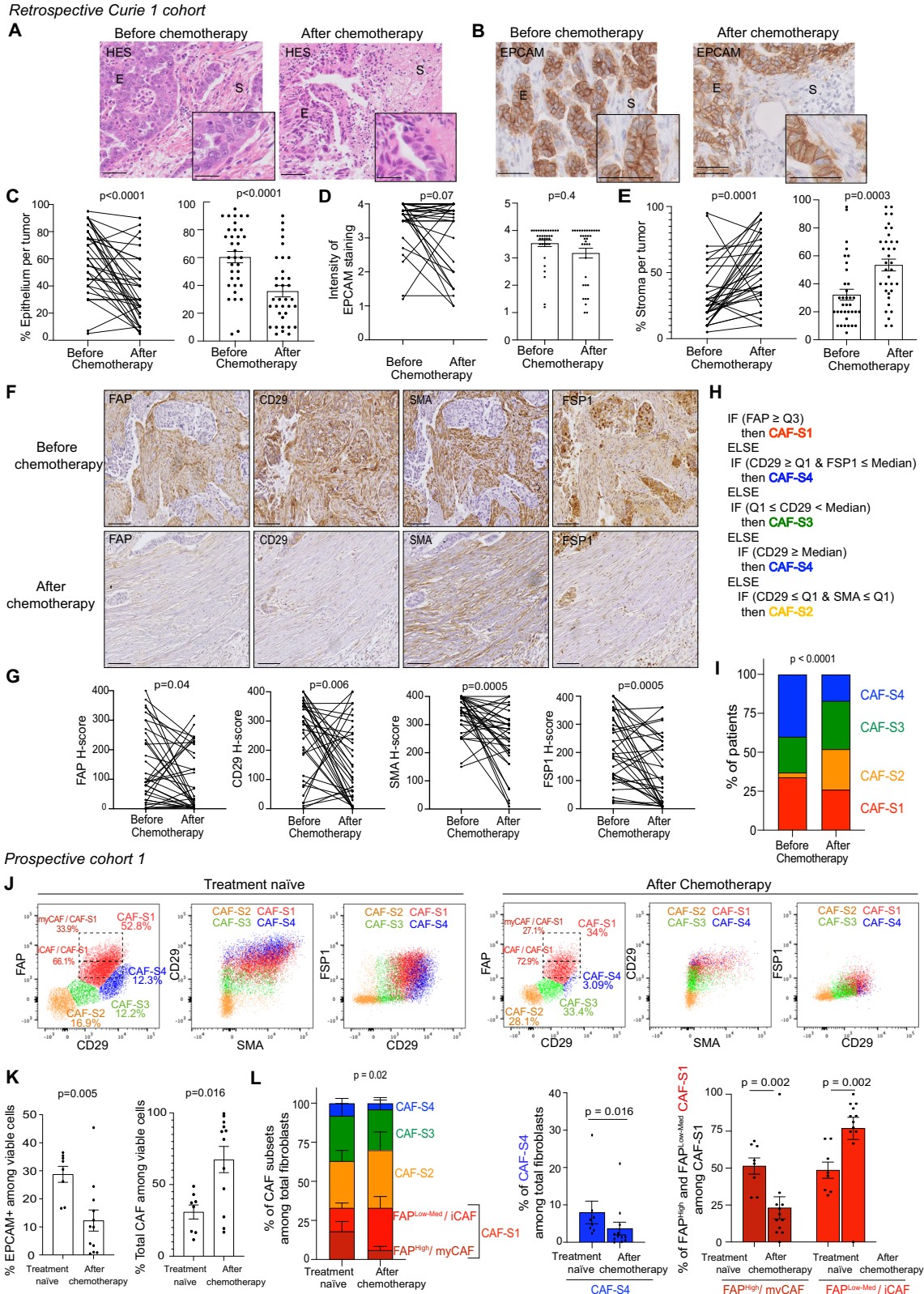

*Retrospective Curie 1 cohort*

*Prospective cohort 1*

earlier than patients enriched in CAF-S2 or CAF-S3 after treatment (*P*-value = 0.01 for overall survival and *P*-value = 0.018 for disease-free survival by Kaplan-Meier test) (Supplementary Fig. S1A). Importantly, multivariate Cox regression analysis showed that this effect was independent on the level of residual epithelial content after treatment (Supplementary Fig. S1B), thereby highlighting the interest of variations of these CAF populations after treatment.

We next sought to confirm the impact of chemotherapy on CAF populations by performing multicolor flow cytometry on freshly resected HGSOC (Table 2, Prospective cohort 1) (Fig. 1J–L). The prospective cohort 1 was composed of fresh samples collected and processed immediately after surgery either from treatment-naïve or from chemotherapy-treated HGSOC patients, as part of routine standard of care. Among viable cells isolated from these fresh HGSOC, we

**Fig. 1 | Decrease in CAF-S1 and CAF-S4 content after chemotherapy in HGSOC.** **A** Hematoxylin eosin saffron (HES) staining in paired (before and after chemotherapy) HGSOC (Retrospective Curie 1 cohort). Scale bars, 50 µm and 25 µm (insert). **B** EPCAM IHC staining. Scale bars, 50 µm and 25 µm (insert). **C** Percentage (%) of epithelium (from HES) in HGSOC before and after chemotherapy ($N = 35$ patients, $n = 70$ matched samples). Data are shown using paired (Left, two-sided paired Wilcoxon test) and unpaired (Right, two-sided Mann-Whitney test) statistical analyses. **D** Intensity of EPCAM staining in epithelial cells, ranging from 0 to 4 ($N = 35$ patients). Paired (Left, two-sided paired Wilcoxon test) and unpaired (Right, two-sided Mann-Whitney test) analyses. **E** Percentage of the fibroblastic stroma (from HES) in HGSOC ($N = 35$ patients). Paired (Left, two-sided paired Wilcoxon test) and unpaired (Right, two-sided Mann-Whitney test) analyses. **F** IHC staining of FAP, CD29, SMA and FSP1 CAF markers on serial sections of paired HGSOC. Scale bars, 50 µm. **G** CAF marker H-scores in HGSOC ($N = 35$ patients). Two-sided paired Wilcoxon test. **H** Decision tree algorithm defining CAF identity (See Methods). **I** Repartition of CAF populations enrichment in HGSOC based on the decision tree shown in (**H**) ($N = 35$ patients). Two-sided Chi-square test. **J** Flow cytometry plots showing CD29, FAP, SMA and FSP1 protein levels in viable fibroblasts from HGSOC samples collected before (Left, treatment-naïve) and after chemotherapy (Right) (See also Supplementary Fig.S1 showing the gating strategy). **K** Left, Quantifications of EPCAM$^+$ epithelial cells among viable cells ($N = 20$ HGSOC tumor samples). Two-sided unpaired t-test. Right, same as in Left for CAF among viable cells ($N = 8$ treatment naïve, $N = 12$ after chemotherapy). Two-sided Mann-Whitney test. **L** Left, % of CAF populations in HGSOC tumor samples ($N = 8$ treatment naïve, $N = 12$ after chemotherapy). Two-sided Chi-square test. Right, Bar plot showing the % of FAP$^{High}$ (myCAF) and FAP$^{Low-Med}$ (iCAF) among CAF-S1 ($N = 8$ treatment naïve, $N = 12$ after chemotherapy). Two-sided Mann-Whitney test. Data are presented as mean ± SEM.

## Table 2 | Main patient characteristics and clinic-pathological features of HGSOC prospective and spatial cohorts

| | Prospective cohort 1 (Flow cytometry) | Prospective cohort 2 (scRNAseq) | Spatial cohort |
|---|---|---|---|
| **Total number of patients** | 20 | 12 | 8 |
| Untreated | 8 (40%) | 5 (41,6%) | 4 |
| Treated | 12 (60%) | 7 (58,4%) | 6 |
| Date of inclusion | 2017–2020 | 2018–2021 | |
| **Age of the patients (years)** | | | |
| Median age | 65,46 | 67 | 61,5 |
| Range | 41–83 | 46–79 | 43–74 |
| **Histotype** | | | |
| Serous | 20 (100%) | 12 (100%) | 8 (100%) |
| **Grade** | | | |
| High | 19 (95%) | 12 (100%) | 8 (100%) |
| Low | 1 (5%) | 0 (0%) | 0 (0%) |
| **FIGO stage** | | | |
| I | 1 (5%) | 0 (0%) | 0 (0%) |
| II | 1 (5%) | 0 (0%) | 1 (12,5%) |
| III | 14 (70%) | 5 (41,6%) | 4 (50%) |
| IV | 4 (20%) | 4 (33,4%) | 3 (37,5%) |
| NA | 0 (0%) | 3 (25%) | 0 (0%) |
| **Surgery (debulking)** | | | |
| Partial | 0 (0%) | 1 (8,35%) | 1 (12,5%) |
| Full | 18 (90%) | 10 (83,3%) | 7 (87,5%) |
| NA | 1 (5%) | 1 (8,35%) | 0 (0%) |
| No debulking surgery | 1 (5%) | 0 (0%) | 0 (0%) |
| **CA-125 before surgery** | | | |
| ≤ 65 | 2 (10%) | 1 (8,3%) | 0 (0%) |
| > 65 | 18 (90%) | 8 (66,7%) | 8 (100%) |
| NA | 0 (0%) | 3 (25%) | 0 (0%) |
| **BRCAness status** | | | |
| BRCA1 | 2 (10%) | 3 (25%) | 1 (12,5%) |
| BRCA2 | 1 (5%) | 0 (0%) | 0 (0%) |
| No mutation of BRCA 1/2 | 16 (80%) | 9 (75%) | 6 (75%) |
| Other mutation (PALP2) | 1 (5%) | 0 (0%) | 1 (12,5%) |
| **Resistance to carboplatin** | | | |
| Sensitivity | 15 (75%) | 6 (50%) | 7 (87,5%) |
| Resistant | 5 (25%) | 5 (41,6%) | 1 (12,5%) |
| NA | 0 (0%) | 1 (8,4%) | 0 (0%) |

## Table 2 (continued) | Main patient characteristics and clinic-pathological features of HGSOC prospective and spatial cohorts

| | Prospective cohort 1 (Flow cytometry) | Prospective cohort 2 (scRNAseq) | Spatial cohort |
|---|---|---|---|
| **Delay of relapse (months)** | | | |
| Median | 6,95 | 5 | 9 |
| Range | 2.49–11.9 | 0–25 | 6–75 |
| **Neoadjuvant chemotherapy** | | | |
| Including Carboplatin and/or taxol | 14 (70%) | 9 (75%) | 3 (37,5%) |
| No neoadjuvant chemotherapy | 6 (30%) | 3 (25%) | 5 (62,5%) |
| **Adjuvant chemotherapy** | | | |
| Including Carboplatin and/or taxol | 17 (85%) | 11 (91,6%) | 7 (87,5%) |
| No adjuvant chemotherapy | 2 (10%) | 1 (8,4%) | 1 (12,5%) |
| NA | 1 (5%) | 0 (0%) | 0 (0%) |
| Bevacizumab | 10 (50%) | NA | NA |
| Olaparib | 2 (10%) | NA | NA |

Prospective cohorts 1 (flow cytometry) and 2 (scRNAseq): HGSOC patients of the prospective cohorts 1 and 2 (PC1, 2) have been treated at Institut Curie Hospital Group between 2017 and 2020 (PC1) and between 2018 and 2021 (PC2). All analyzed samples have been collected either prior (8 patients for PC1 and 9 for PC2) or after chemotherapeutic treatment (12 patients for PC1 and 9 for PC2). The prospective cohorts are homogenous in terms of treatments and surgery. For each patient, a surgical specimen was taken before or after chemotherapy for multicolor flow cytometry analysis (PC1) or scRNAseq (PC2). The median age of ovarian cancer patients at diagnosis was 65.4 years, with a range of 41 to 83 years (PC1) and 67 years, with a range of 46–79 years (PC2). Ovarian carcinomas were classified according to the World Health Organization histological classification of gynecological tumors. All patients were suffering from tumors of serous histological subtype, 95% of high-grade and one of low grade (5%) in PC1 and 100% of high-grade in PC2. 90% and 75% were of advanced stage (FIGO stages III and IV) in PC1 and PC2, respectively. Around 90% of patients had a full debulking, 10% had a partial one or had no debulking surgery (in case of progression of the carcinomatosis during neoadjuvant chemotherapy). Most patients were treated with a combination of surgery and chemotherapy, the latter including platinum salts and taxanes, as a first line of treatment. The majority of patients had been treated with adjuvant platinum-based chemotherapy following surgery. 75% and 50% of patients were platinum-sensitive in PC1 and PC2, respectively. CA-125 serum level before interval debulking surgery, measured as an indicator of the residual size of the tumor after neoadjuvant chemotherapy was higher than 65 (double the normal level) in 90% of the patients in PC1 and in around 70% of the patients in PC2. Almost all patients (around 85%) had no germline mutation in *BRCA1* or *2* genes. Clinical categories are indicated in bold.

identified epithelial (EPCAM$^+$), hematopoietic (CD45$^+$), endothelial cells (CD31$^+$) and red blood cells (CD235a$^+$) (Supplementary Fig. S1C). Fibroblasts were enriched in the EPCAM$^-$ CD45$^-$ CD31$^-$ CD235a$^-$ cellular fraction and further characterized using FAP, CD29, SMA and FSP1 markers (Fig. 1J and Supplementary Fig. S1C). We distinguished the four different CAF populations by flow cytometry in HGSOC prior to treatment (Fig. 1J–L), as observed by IHC. In addition, thanks to the

sensitivity of the flow cytometry method, we were able to distinguish FAP$^{Low-Med}$ from FAP$^{High}$ CAF-S1, previously shown to characterize inflammatory (iCAF) and myofibroblastic (myCAF) clusters, respectively[47,50,82]. As shown above in the retrospective cohort, following chemotherapy, we first detected a decrease in the percentage of epithelial cells among viable cells, together with a concomitant increase in total CAF content (Fig. 1K). While the global CAF content increased, we observed variations in CAF-S1 and CAF-S4 content. Indeed, the proportion of the CAF-S4 population was reduced in treated samples (Fig. 1L). In addition, flow cytometry analysis enabled us to detect more precisely that, among the CAF-S1 population, the content in FAP$^{High}$ CAF-S1 (enriched in myCAF clusters) was the most reduced after chemotherapy (Fig. 1J, L). Taken as a whole, these data show that the abundance of both CAF-S4 and FAP$^{High}$ CAF-S1 populations significantly decreases after chemotherapy in HGSOC patients.

## The increased density of CD8$^+$ T lymphocytes after chemotherapy is correlated with the decrease in CAF-S1 abundance

Based on the impact of CAF-S1 fibroblasts on immunosuppression, we next investigated Tumor infiltrating lymphocyte (TIL) density and localization before and after chemotherapy in HGSOC patients by performing CD3, CD8 and FOXP3 IHC staining in the Retrospective Curie 1 cohort (Fig. 2A). We counted the number of TILs per surface unit of stroma and epithelium before and after chemotherapy (Fig. 2B–D). We observed that CD3$^+$ and CD8$^+$ TILs significantly accumulated after chemotherapy in HGSOC, with the same tendency for FOXP3$^+$ T cells but without reaching significance (Fig. 2B). CD3$^+$, CD8$^+$ and FOXP3$^+$ TILs infiltrated more the stroma than the epithelium (Fig. 2C and Supplementary Fig. S2A). Importantly, the CD3$^+$ and CD8$^+$ T cell density reached their highest levels in the stromal compartment after chemotherapy (Fig. 2C), highlighting the potential importance of this compartment after treatment. As we detected an increase in the proportion of stroma upon chemotherapy (Fig. 1E), we tested if the increased content in TILs after treatment could be linked to this increased stromal content, which could reinforce the role of stroma in T cell density. To test this hypothesis, we normalized the number of TILs per unit surface of stroma (Fig. 2D, Stroma). Interestingly, this normalization abrogated the difference of TIL density before/after treatment in the stroma (Fig. 2D and Supplementary Fig. S2B), showing that TIL density after chemotherapy is associated with the overall enrichment in stroma and underlying the importance of the stroma in this process.

We next wondered whether TIL density after chemotherapy could be linked to the extent of CAF-S1 or CAF-S4 decrease. To compare the variations of each cellular population after *versus* before chemotherapy in each patient, we established a delta score (Δ) calculated as followed: content of the studied population after chemotherapy *minus* (−) content of the same population before chemotherapy (see also Methods #*Establishment of a delta-score measuring variations of each population by chemotherapy*). We analyzed the variations of CD8$^+$ TILs (assessed by the Δ-number of CD8$^+$ TILs per surface unit) and CAF-S1 (evaluated by the Δ-score of FAP, a CAF-S1 specific marker) in paired HGSOC patients (Fig. 2E, F and Supplementary Fig. S2C). Interestingly, this analysis showed an anti-correlation between CD8$^+$ TILs and CAF-S1, suggesting that the more CAF-S1 decrease after chemotherapy, the more CD8$^+$ TIL density increases in the tumor and thus highlighting the importance of the extent of CAF-S1 variation upon treatment. In contrast, we found no association between the proportion of TILs and the overall decrease in myofibroblastic CAF populations, i.e., when considering both CAF-S1 and CAF-S4 together (evaluated by the Δ-score of the SMA marker, a common marker of these two myofibroblastic populations) (Supplementary Fig. S2D, E). This result showed that the increase in CD8$^+$ TIL density after treatment is specifically anti-correlated with the CAF-S1 content in HGSOC. Finally, we sought to confirm this observation and to investigate the heterogeneity of the

CAF-S1 population by flow cytometry analysis of untreated and treated HGSOC (Prospective cohort 1, Table 2) (Fig. 2G–J), although these unpaired samples did not allow us to compare cellular variations before/after chemotherapy per patient. When we considered all patients (without distinguishing patients enriched in FAP$^{High}$ CAF-S1 from those enriched in FAP$^{Low-Med}$ CAF-S1), we did not detect any significant difference between treatment-naïve and chemo-treated samples (Fig. 2H). However, when we focused our analysis on FAP$^{High}$ CAF-S1 thanks to the sensitivity of the flow cytometry method, we detected an anti-correlation between the percentage of CAF-S1 among total CAF and the number of CD8$^+$ T lymphocytes among CD3$^+$ T cells (Fig. 2I). But, there was no link in any way with FAP$^{Low-Med}$ CAF-S1 (Fig. 2J) and CAF-S4 (Supplementary Fig. S2F). Altogether, these observations show that the content in CD8$^+$ TILs is negatively correlated with CAF-S1, in particular FAP$^{High}$ CAF-S1.

## Single cell analysis and spatial transcriptomics show that the content in the ECM-myCAF cluster decreases the most with chemotherapy

Data shown above suggest that CAF-S1, in particular FAP$^{high}$ CAF-S1, might be linked to CD8$^+$ TIL density. We thus next checked whether iCAF and myCAF clusters were affected upon chemotherapy. In that aim, we analyzed ANTXR1, one of the most discriminant marker between myCAF (ANTXR1$^+$) and iCAF (ANTXR1$^-$)[82], and compared ANTXR1 protein level by IHC in the retrospective Curie 1 cohort and by flow cytometry in the prospective cohort 1 (Fig. 3A–D). Both techniques showed that the content in ANTXR1$^+$ myCAF clusters was significantly reduced after chemotherapy in HGSOC patients (Fig. 3B, D), confirming that chemotherapy mainly decreases the content in myCAF clusters within the CAF-S1 population. To go deeper in the characterization of the CAF-S1 clusters (including the ANTXR1$^-$ iCAF and ANTXR1$^+$ myCAF clusters) following chemotherapy, we performed single cell RNA sequencing (scRNAseq) analysis of CAF-S1 from 12 HGSOC patients before and after treatment (Table 2, Prospective cohort 2 and Supplementary Fig. S3A). Fresh samples from the prospective cohort 2 were collected at time of surgery either from treatment-naïve or from chemotherapy-treated HGSOC patients, as part of routine standard of care. This enabled us to analyze CAF-S1 cluster identity in HGSOC patient before and after chemotherapy using scRNAseq (Fig. 3E). We first observed that ANTXR1 expression, as well as the percentage of ANTXR1$^+$ CAF-S1, were significantly decreased in treated HGSOC compared to treatment-naïve samples (Fig. 3E, Left), thereby confirming, at single cell transcriptomic level, the data obtained by IHC and flow cytometry from HGSOC. We also took advantage of the publicly available scRNAseq data from CAF isolated from untreated and chemotherapy-treated samples from an independent HGSOC cohort of patients treated at Turku University Hospital[99] and in breast cancer (BC) patients[100] (Fig. 3F, G). By this way, we confirmed that ANTXR1 expression and the proportion of ANTRX1$^+$ CAF-S1 were reduced following chemotherapy in an independent cohort of HGSOC patients (Turku cohort), as well as in BC (Fig. 3F, G), strengthening the validity of our observations both in ovarian and breast cancer.

We recently showed that the CAF-S1 population is composed of 5 myCAF clusters (ECM-myCAF, TGFβ-myCAF, Wound-myCAF, IFNαβ-myCAF and Acto-myCAF) and 3 iCAF clusters (Detox-iCAF, IL-iCAF and IFNγ-iCAF) in BC[82]. To identify the CAF-S1 clusters modulated by chemotherapy, we used the reference-based approach from Seurat to predict and annotate CAF-S1 clusters in scRNAseq data (Fig. 3H, I and Supplementary Fig. S3A–C). By this way, we observed high prediction scores of all CAF-S1 clusters in the two HGSOC independent cohorts (Curie and Turku cohorts) as well as in BC (Supplementary Fig. S3A–C, Bottom). We also confirmed the identity of the most abundant CAF-S1 clusters detected (ECM-myCAF, Detox-iCAF and Wound-myCAF) using an unsupervised method by applying consensus Non-Negative Matrix

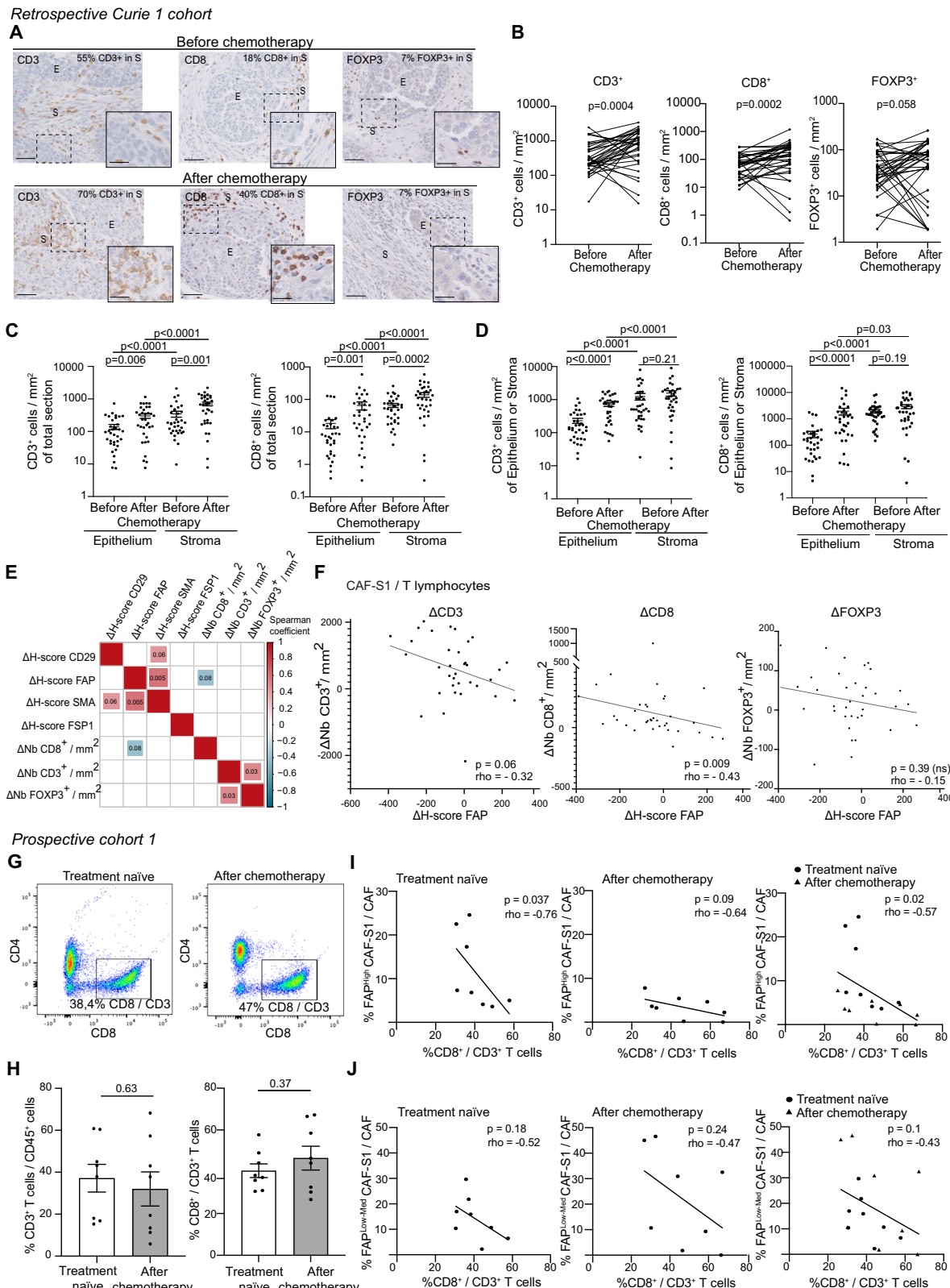

*Retrospective Curie 1 cohort*

*Prospective cohort 1*

factorization (cNMF)[101] on scRNAseq data from the Curie cohort (Supplementary Fig. S3D). Among the different CAF-S1 clusters identified, the ANTXR1[+] ECM-myCAF was the most abundant in treatment-naïve HGSOC (in both Curie and Turku cohorts) and in BC (Fig. 3H, I and Supplementary Fig. S3E). Interestingly, the ECM-myCAF was also the CAF-S1 cluster, which decreased the most -compared to all other CAF-S1 clusters- after chemotherapy in HGSOC and in BC (Fig. 3H, I and

Supplementary Fig. S3E). As the number of CAF analyzed after neoadjuvant chemotherapy in BC was much lower than from treatment-naïve samples (Supplementary Fig. S3E), we validated all these observations after down-sampling of treatment-naïve CAF-S1 cells (Supplementary Fig. S3F). In parallel with the reduction of the ECM-myCAF cluster, the Detox-iCAF cluster was the only one to be systematically detected following chemotherapy in the 3 datasets, the

**Fig. 2 | Inverse correlation between CAF-S1 and CD8⁺ T cell content upon chemotherapy. A** IHC of CD3⁺, CD8⁺ and FOXP3⁺ TILs before and after chemotherapy. Epithelial (E) and stromal (S) compartments, and stromal immune cell percentages are indicated. Scale bars, 50 μm and 25 μm (Inset). **B** Number of CD3⁺, CD8⁺ and FOXP3⁺ TILs per mm² of total HGSOC sections before and after chemotherapy (*N* = 35 patients, *n* = 70 matched samples). Two-sided Wilcoxon paired test. **C** Number of CD3⁺ and CD8⁺ TILs per mm² in HGSOC stromal and epithelial compartments, respectively (*N* = 35 patients). Two-sided Wilcoxon paired test. **D** Same as (**B**) reported to the stromal or epithelial content in each sample (*N* = 35). Two-sided Wilcoxon paired test. **E** Correlation matrix between variations (after/before chemotherapy) of FAP, SMA, FSP1 and CD29 H-scores and the number of CD3⁺, CD8⁺ and FOXP3⁺ TILs per mm² (*N* = 35). Variations are assessed by a delta score (Δ) (See Methods for calculation). Positive (red) and negative (blue) correlations with *p*-values < 0.1 (Spearman test after Benjamini & Hochberg correction for multiple testing) (*N* = 35). Square sizes are proportional to *P*-values and color intensities to the correlation coefficients (*P*-value is specified in square). **F** Correlation plots with linear regression lines comparing the variations upon chemotherapy in the content of CAF-S1 (assessed by ΔH-score FAP) and the number of CD3⁺ (Left), CD8⁺ (Middle) and FOXP3⁺ (Right) TILs per mm² of total section. Each dot represents one tumor (*N* = 35). Two-sided Spearman correlation test. **G** CD4 and CD8 protein levels in HGSOC samples (Left, treatment-naïve and Right, after chemotherapy). **H** % of CD3⁺ TILs among CD45⁺ cells (Left) and % CD8⁺ TILs among CD3⁺ T cells (Right) assessed by flow cytometry (*N* = 8 treatment naïve; *N* = 8 after chemotherapy). Two-sided unpaired Student t-test. **I, J** Correlations plots with linear regression lines between % of CD8⁺ TILs and % of FAP^High CAF-S1 (**I**) and FAP^Low-Med CAF-S1 (**J**) assessed by flow cytometry (*N* = 16 patients). Two-sided Spearman correlation test. Data are presented as mean ± SEM.

content in the Wound-myCAF cluster being also strongly increased after treatment but only in 2 out of 3 datasets (Fig. 3H, I and Supplementary Fig. S3E). We then sought to validate these findings in an independent cohort of ovarian cancer patients (containing 45 samples before treatment and 25 after chemotherapy, see Table 1, retrospective SCANDARE Curie 2 cohort) by studying bulk RNA-seq data. For this cohort of patients, long-term clinical follow-up and information on treatment responses of patients were available. Cellular composition of each sample -before and after chemotherapy- was inferred by using BayesPrism method[102] based on a high-resolution HGSOC cellular atlas that we built and annotated from both Curie and publicly available scRNAseq datasets (Supplementary Fig. S4A, B). This HGSOC atlas was composed of 49,909 cells and 24 different cell types and states, including CAF-S1 clusters, thereby constituting a comprehensive HGSOC cellular landscape (Supplementary Fig. S4A, B). In this independent HGSOC cohort, we confirmed that the proportion of CD8⁺ T lymphocytes increased after treatment (Fig. 3J). Importantly, we also validated the decrease in ECM-myCAF content and increase in Detox-iCAF after chemotherapy (Fig. 3K). Moreover, thanks to the long-term clinical follow-up available in the SCANDARE cohort, we were able to highlight that the decrease in the ECM-myCAF content and the increase in the Detox-iCAF proportion following chemotherapy were mainly detected in responder patients but not in non-responders (Fig. 3L and Supplementary Fig. S4C). Moreover, the increase of CD8 T cell density was also detected in responder patients (Fig. 3M), showing that variations of these CAF-S1 clusters are concomitant to increased CD8 + T cell content after chemotherapy. Taken as a whole, these data confirmed the clinical interest of the variations of these CAF-S1 clusters upon treatment.

Finally, we aimed to define the spatial localization and contribution of CAF-S1 clusters in HGSOC at time of diagnosis and in residual disease after chemotherapy. To do so, we used the Visium technology and compared spatial transcriptomic data (see Methods #*Spatial Transcriptomics*) from 10 different HGSOC samples collected at baseline and after chemotherapy (Fig. 3N–P). We first performed pathological annotations to distinguish epithelial and stromal compartments in these HGSOC sections (Supplementary Fig. S4D). We confirmed that the amount of stroma, as well as the number of CD3⁺ and CD8⁺ T lymphocytes, detected after chemotherapy were higher than in treatment-naïve (Supplementary Fig. S4E, F). These observations confirmed IHC and flow cytometry data and validated that the samples selected for spatial transcriptomic analyses are representative. In order to extract the signals from those compartments, we evaluated the abundance of the different CAF-S1 clusters in each spot within the sections by applying the deconvolution method cell2location[103], using the matrix of reference cell types from the HGSOC cellular atlas (Supplementary Fig. S4A, B) as input. By this way, we could explore the spatial localization of each CAF-S1 cluster in the 10 HGSOC collected and compared the baseline and residual states. We first confirmed within these tissue sections that ANTXR1 expression level and the content in ANTXR1⁺ CAF-S1 were significantly reduced after chemotherapy compared to treatment-naïve section (Fig. 3O). In these sections, we identified the different CAF-S1 clusters and confirmed that the ANTXR1⁺ ECM-myCAF cluster accumulated the most before treatment and decreased the most after chemotherapy (Fig. 3P). We also observed that the content in the Detox-iCAF cluster increased after treatment (Fig. 3P). Taken as a whole, these data highlight the relevance of the decrease in the ANTXR1⁺ ECM-myCAF cluster content after chemotherapy in various cancer types by using several complementary approaches.

## YAP1 protein levels significantly decrease in ECM-myCAF after chemotherapy

We next aimed to identify the pathways down-regulated after chemotherapy in CAF-S1. To this end, we inferred transcription factor activity (TF) from the expression of their gene targets using the DoRothEA algorithm[104]. We first observed that one of the most differential regulon modulated in CAF-S1 after chemotherapy was composed of TEAD-target genes (Fig. 4A, B and Supplementary Table S2). Indeed, the TEAD (TEA domain transcription factor) family members, including TEAD1, TEAD2 and TEAD4, were all three identified among the Top-50 transcription factors, whose activity significantly decreased after chemotherapy in CAF-S1 (Fig. 4A). We previously observed that several TEAD-target genes are up-regulated in CAF-S1 compared to CAF-S4 in HGSOC[40]. Moreover, we found that TEAD-target genes were more highly expressed in ECM-myCAF than in all other CAF-S1 clusters (Supplementary Fig. S5A). In agreement with TEAD-TF activity upon chemotherapy, TEAD-target genes were significantly down-regulated in CAF-S1 and in ECM-myCAF after chemotherapy compared to treatment-naïve samples (Fig. 4B). Consistent with these observations, spatial transcriptomic data also showed a significant down-regulation of TEAD-target genes after chemotherapy, in particular in areas enriched in ECM-myCAF (Fig. 4C, D). YAP1 and its paralogue TAZ (encoded by *WWTR1*) are transcriptional co-activators that bind and promote the activity of the TEAD transcription factors. Several YAP1/TEAD-target genes were strongly expressed in CAF-S1 fibroblasts[40,80], in particular in the ECM-myCAF cluster (Supplementary Fig. S5A), suggesting that YAP1 could be a key regulator of TEAD transcription factors in CAF-S1 in HGSOC. We thus next sought to validate this hypothesis by testing if the YAP1 protein level was decreased in CAF-S1 after chemotherapy in the retrospective Curie 1 cohort of HGSOC patients (Fig. 4E, F). We observed that YAP1 histological scores (H-scores) significantly decreased in HGSOC stroma following chemotherapy, while no significant change was observed in epithelium (Fig. 4E, F; validation of YAP1 antibody specificity in Supplementary Fig. S5B, C). Interestingly, YAP1 H-scores in stroma were correlated with those of FAP and ANTXR1, specific markers of CAF-S1 and ECM-myCAF, respectively (Fig. 4G, H). In contrast, stromal YAP1 H-scores were not correlated with SMA, marker of both CAF-S1 and CAF-S4 populations (Supplementary Fig S5D). As YAP1 localization in the nucleus is key for its

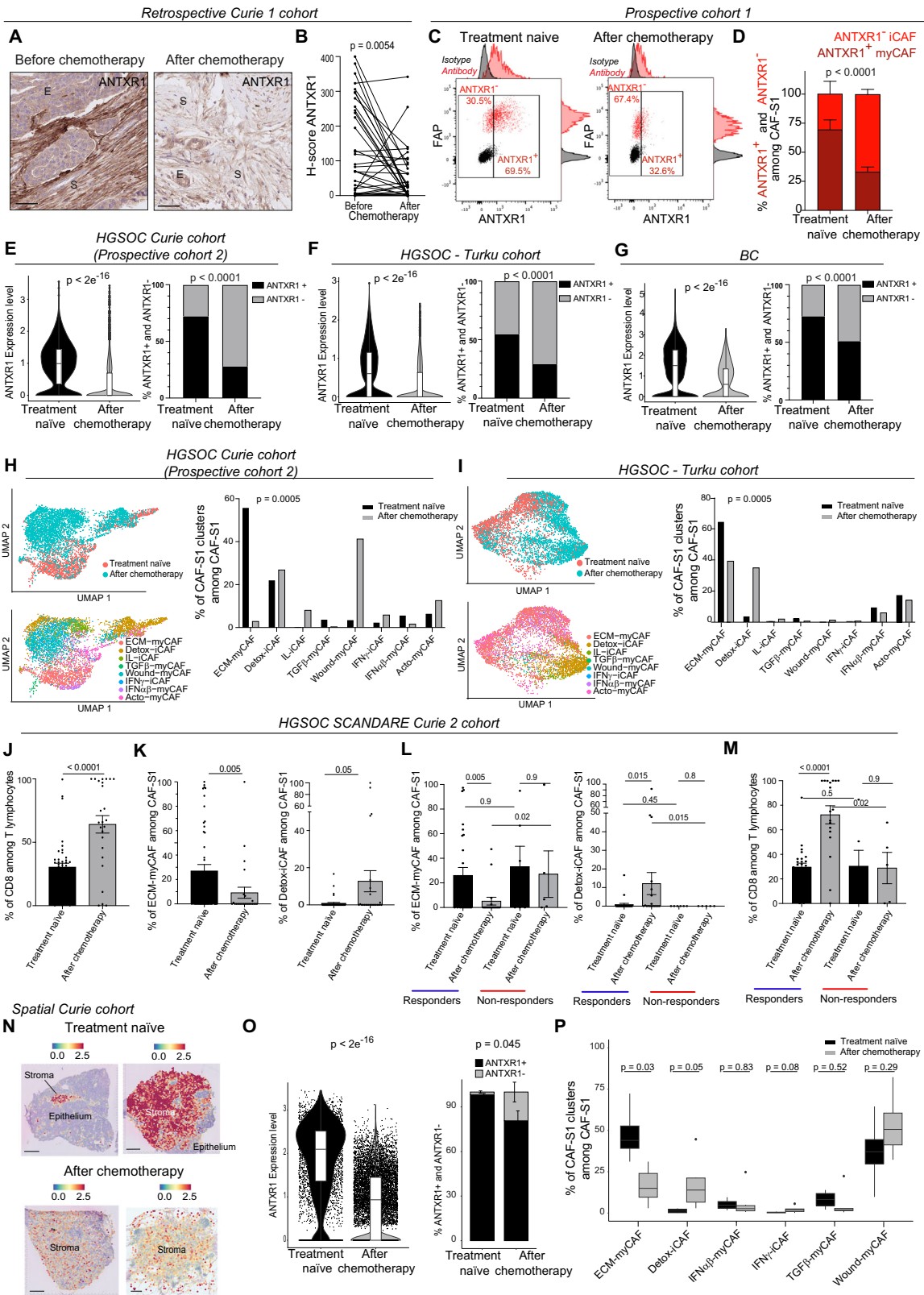

interaction with TEAD transcription factors, we next analyzed YAP1 accumulation in nuclei and found that YAP1 nuclear staining was significantly reduced after chemotherapy in stroma and in cancer cells (Fig. 4I, J). Moreover, the decrease in nuclear YAP1 staining in stroma following chemotherapy was correlated with the one of FAP and ANTXR1 (Fig. 4K, L), but not with SMA (Supplementary Fig. S5E). Thus, the reduced expression of YAP1 and TEAD-target genes after

chemotherapy was concomitant to the decrease in the content of ANTXR1⁺ CAF, marker of ECM-myCAF, suggesting that depletion of this CAF-S1 cluster after treatment may explain -at least in part- this decrease in YAP1/TEAD-signaling pathway. In addition, we also considered that chemotherapy might also reduce YAP1 protein levels in the residual ANTXR1⁺ CAF after treatment. To test this hypothesis, we analyzed the co-localization of YAP1 and ANTXR1 staining in CAF

**Fig. 3 | The proportion of the ANTXR1⁺ ECM-myCAF cluster is the most reduced by chemotherapy in HGSOC. A** ANTXR1 staining in paired HGSOC. Scale bars, 50 µm. **B** ANTXR1 H-scores (N = 35 patients). Two-sided paired Wilcoxon test. **C** Flow cytometry plots showing FAP and ANTXR1 proteins to distinguish ANTXR1⁺ FAP⁺ myCAF and ANTXR1⁻ FAP⁺ iCAF. **D** Percentage of ANTXR1⁺ myCAF and ANTXR1⁻ iCAF among CAF-S1 population in HGSOC (N = 7 treatment-naïve, N = 9 after chemotherapy). Two-sided two-way ANOVA test. **E–G** ANTXR1 expression from scRNAseq (Left) and % of ANTXR1⁺ CAF-S1 among total CAF-S1 (Right) in HGSOC Prospective Curie cohort 2 (N = 12, 5 treatment-naïve and 7 after chemotherapy) (**E**), HGSOC Turku cohort (N = 11 paired before/after treatment) (**F**) and breast cancer (BC) cohort (N = 42, 31 treatment-naïve and 11 after chemotherapy) (**G**). Two-sided Wilcoxon test (**E–G**, Left) and two-sided Fischer's Exact test (**E–G**, Right). **H** Left, UMAP of 5 618 CAF-S1 from HGSOC Curie cohort colored by treatment status (Up) or by CAF-S1 clusters predicted using label transfer (Bottom). Right, % of CAF-S1 clusters among CAF-S1. Two-sided Fisher's exact test. **I** Same as in

(**H**) for 5 658 CAF-S1 from HGSOC Turku cohort. **J** Percentage of CD8+ TILs in the retrospective SCANDARE Curie 2 cohort (N = 70 samples: 45 treatment-naïve, 25 after chemotherapy). Two-sided Mann-Whitney test. **K** Same as (**J**) % of ECM-myCAF (Left) and Detox-iCAF (Right). **L, M** Same as (**J, K**) according to response to chemotherapy for ECM-myCAF (**L**, Left), Detox-iCAF (**L**, Right) and CD8+ TILs (**M**). Two-sided Mann-Whitney test. **N** ANTXR1 expression in HGSOC Visium sections at time of diagnosis and in residual disease. Scale bars, 1 mm. **O** ANTXR1 expression (Left) and mean proportion of deconvoluted ANTXR1⁺ (black) and ANTXR1⁻ (grey) CAF-S1 among CAF-S1 (Right) in stromal spots of Visium sections (N = 4 Treatment naïve, n = 13 438 spots and N = 6 After chemotherapy, n = 24 205 spots). Two-sided Wilcoxon test. **P** Percentage of each deconvoluted CAF-S1 cluster in the 10 Visium sections. Two-sided Wilcoxon test. Data are presented as mean ± SEM. In boxplot the center line, box limits and whiskers indicate the median, upper and lower quartiles and 1.5x interquartile rage.

before and after chemotherapy (Fig. 4M). We found that the few residual ANTXR1⁺ CAF after treatment displayed a decreased YAP1 staining compared to ANTXR1⁺ CAF before treatment (Fig. 4M, N). Indeed, while the majority of ANTXR1⁺ CAF were YAP1⁺ before treatment, the few residual ANTXR1⁺ CAF were predominantly YAP1⁻/Low (Fig. 4O). Taken as a whole, our results indicate that the reduction of YAP1 staining after chemotherapy was mainly due to the concomitant loss of the ECM-myCAF population, but also to the reduction of YAP1 staining in the few residual CAF-S1 detected after treatment.

### ECM-myCAF spatially segregated away from CD8 + T lymphocytes in HGSOC

As we observed that CD8⁺ density increased in HGSOC upon treatment (Fig. 2), we hypothesized that the loss of ECM-myCAF and the decrease of YAP1 expression could be responsible for CD8⁺ T cell enrichment. To get an overview of the spatial organization of CD8⁺ T lymphocytes and ECM-myCAF, we first took advantage of spatial transcriptomic data and cellular HGSOC atlas we built (Supplementary Fig. S4A, B). We mapped the localization and the abundance of each cell population by performing deconvolution using the Cell2location algorithm[103]. We applied non-negative matrix factorization (NMF) analysis to identify patterns of co-localization of cell types and states and found that the ECM-myCAF cluster spatially segregated away from CD8⁺ T lymphocytes (Fig. 5A–C). We confirmed this observation by evaluating the distances of CD8⁺ T lymphocytes to the closest ECM-myCAF-enriched or -depleted spots (Fig. 5D). To gain deeper insights into the spatial organization of CD8⁺ T lymphocytes and ANTXR1⁺ CAF at protein level and at single cell resolution, we performed multiplex imaging of ANTXR1, CD8 and cytokeratin proteins in HGSOC (Fig. 5E). After cell segmentation, we were able to classify each cell type (ANTXR1⁺ and ANTXR1⁻ CAF, CD8⁺ T lymphocytes and pan-cytokeratin⁺ cancer cells) based on intensities of their specific proteins using an unsupervised approach (Fig. 5F, G) (See also Methods #*Pan-cytokeratin, CD8 and ANTXR1 multiplex immunofluorescence imaging and colocalization analysis*). We next compared the spatial co-localization of ANTXR1⁺ and ANTXR1⁻ CAF with CD8⁺ T lymphocytes by applying a co-occurrence method implemented in Squidpy[105], which computes the probability of detecting a cluster of interest (here either ANTXR1⁺ or ANTXR1⁻ CAF) depending on the presence of another cluster (here CD8⁺ T lymphocytes) within an increasing radius. Spatial co-occurrence analysis of these multiplex imaging revealed that ANTXR1⁺ CAF were spatially distant from CD8⁺ T lymphocytes and did not colocalize together, while ANTXR1⁻ CAF were more likely to colocalize with CD8⁺ T cells (Fig. 5H). We confirmed this spatial segregation between ANTXR1⁺ CAF and CD8⁺ T lymphocytes by performing IHC co-staining of ANTXR1 and CD8 in the retrospective Curie 1 cohort of HGSOC (Fig. 5I, J).

We then hypothesized that the YAP1⁺ ANTXR1⁺ CAF population could play a role in CD8⁺ T cell exclusion. Based on the IHC results from

the retrospective Curie 1 cohort, we observed that YAP1 protein levels in the stroma was negatively correlated with the number of CD8⁺ T lymphocytes in HGSOC after chemotherapy (Fig. 5K). In contrast, there was no link between CD8⁺ T cell content and YAP1 staining in the epithelium (Fig. 5L), suggesting that YAP1 in the stroma but not in the epithelium might affect CD8⁺ T cell density. We thus next performed serial staining of YAP1, ANTXR1 and CD8 and integrated YAP1 in the spatial co-occurrence analysis. We found that the segregation observed between ANTXR1⁺ CAF and CD8⁺ T lymphocytes (shown in Fig. 5J) was mainly associated with YAP1⁺ ANTXR1⁺ CAF and that the neighborhood of YAP1⁺ ANTXR1⁺ CAF was specifically depleted of CD8⁺ T lymphocytes (Fig. 5 M, N). Altogether, these data suggest that YAP1 in ANTXR1⁺ CAF, enriched in ECM-myCAF, spatially segregate away from CD8⁺ T lymphocytes in HGSOC.

### YAP1 expression in ECM-myCAF inhibits CD8⁺ T cell cytotoxicity

We next aimed to investigate the role of YAP1 in ECM-myCAF on CD8⁺ T lymphocytes by performing functional assays. We sought to compare the impact of YAP1 silencing in ECM-myCAF on CD8⁺ T cell cytotoxic activity, and used Detox-iCAF as control. To do so, we first isolated Detox-iCAF and ECM-myCAF primary fibroblasts from HGSOC patient samples and evaluated their impact on CD8⁺ T lymphocytes isolated from peripheral blood mononuclear cells (PBMC) of healthy donors. We first verified the identity of these fibroblasts in culture by flow cytometry using specific markers (Supplementary Fig. S6A) and by bulk RNAseq (Supplementary Fig. S6B–D) and validated that they corresponded to the ECM-myCAF and Detox-iCAF clusters, respectively. We then analyzed the impact of these two CAF-S1 clusters on CD8⁺ T cell cytotoxicity, considering both markers and activity. We found that, upon co-culture, ECM-myCAF significantly increased the percentage of PD-1⁺ CD8⁺ T lymphocytes, while Detox-iCAF did not (Fig. 6A and Supplementary Fig. S7A). Moreover, this increase in PD-1⁺ CD8⁺ T cells by ECM-myCAF was concomitant to the reduced percentages of granzyme B⁺, perforin⁺ and IFN-γ⁺ CD8⁺ T lymphocytes (Fig. 6B–D and Supplementary Fig. S7A). Here again, the impact on the percentages of CD8 T cells positive for cytotoxic markers was specific of the ECM-myCAF fibroblasts and not detected with Detox-iCAF (Fig. 6B–D and Supplementary Fig. S7A). Interestingly, we confirmed that ECM-myCAF but not Detox-iCAF reduced CD8⁺ T cell cytotoxic activity (Fig. 6E). Indeed, we observed that CD8⁺ T lymphocytes showed reduced capacity to kill cancer cells after co-culture with ECM-myCAF but not with Detox-iCAF (Fig. 6E), thereby confirming ECM-myCAF-mediated immunosuppression on CD8⁺ T lymphocyte cytotoxicity. Interestingly, YAP1 silencing (Supplementary Fig. S7B for siRNA efficacy) in ECM-myCAF reversed ECM-myCAF-mediated effects on cytotoxic CD8⁺ T cells, while it had no impact when using Detox-iCAF (Fig. 6A–E). Finally, we tested if these CAF-S1 clusters could modulate CD8⁺ T cell migration by performing transwell assay (Fig. 6F). Interestingly, Detox-iCAF enhanced CD8⁺ T cell migration,

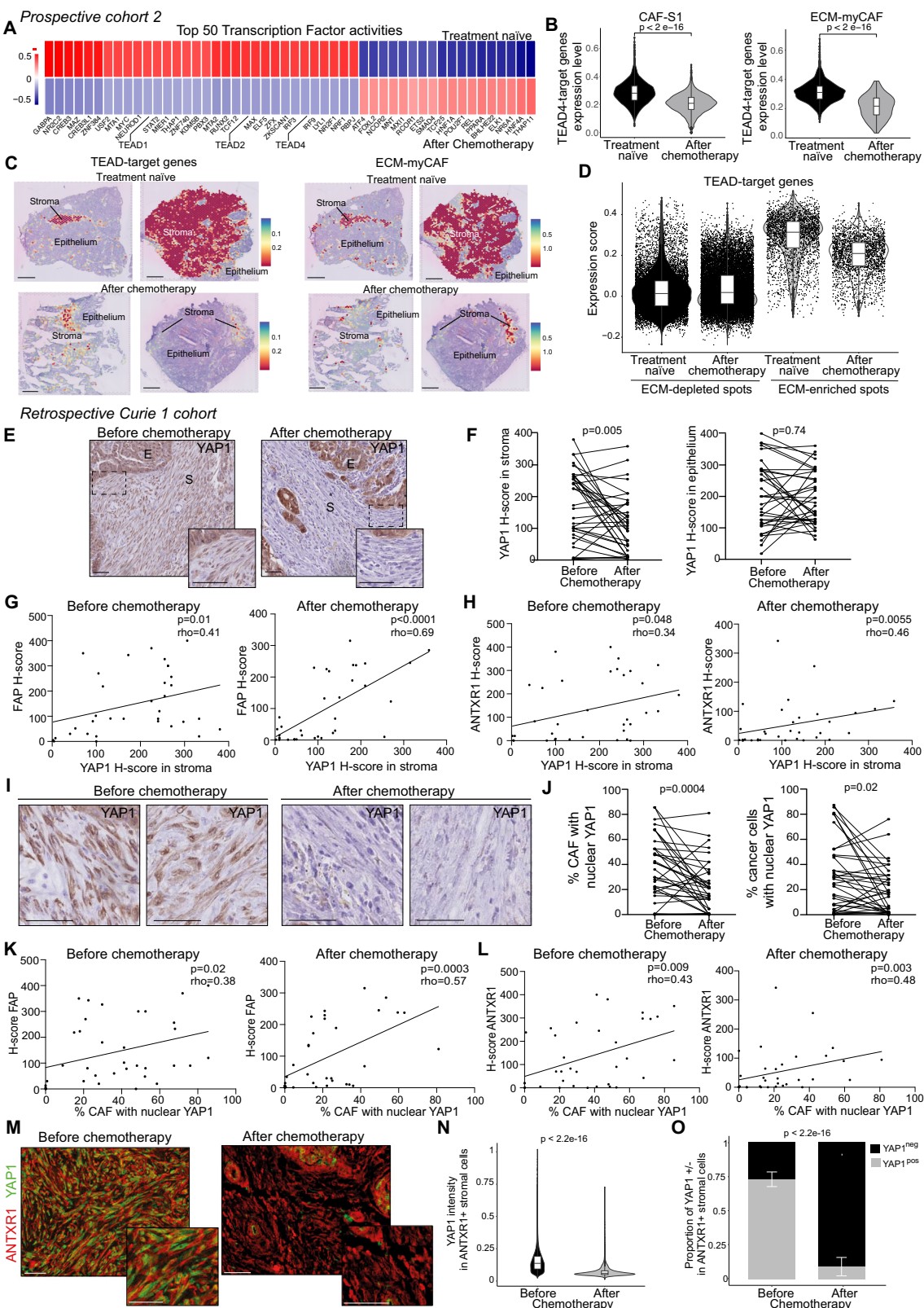

while ECM-myCAF fibroblasts did not (Fig. 6F). Altogether, these observations show complementary roles of the two CAF-S1 clusters on CD8 T lymphocytes: while Detox-iCAF attracts CD8$^+$ T lymphocytes, ECM-myCAF dampen their cytotoxic identity and functions through a YAP-1 dependent mechanism. Among well-established YAP1-TEAD-target genes, CYR61 (Cysteine-rich angiogenic inducer 61, also called CCN1) was highly expressed by CAF-S1, and more specifically by the

ECM-myCAF cluster (Supplementary Fig. S7C). As this gene encodes a secreted protein, we considered that CYR61 could be instrumental for ECM-myCAF to act on CD8$^+$ T lymphocytes. We thus silenced CYR61 in ECM-myCAF primary fibroblasts (Supplementary Fig. S7D for siRNA efficacy) and found that CYR61 silencing in ECM-myCAF restored the percentages of granzyme B$^+$, perforin$^+$ and IFN-γ$^+$ CD8$^+$ T lymphocytes and reduced the proportion of PD-1$^+$ CD8$^+$ T cells (Supplementary

**Fig. 4 | Down-regulation of YAP1/TEAD-dependent pathway in ECM-myCAF after chemotherapy in HGSOC. A** The 50 most variables TFs before and after chemotherapy in the CAF-S1 HGSOC Curie scRNAseq dataset. **B** Expression of TEAD-target genes in CAF-S1 (Left, $n = 1968$ Treatment-naïve, $n = 3650$ After chemotherapy) and in ECM-myCAF (Right, $n = 1099$ Treatment-naïve, $n = 107$ After chemotherapy). $N = 12$ HGSOC patients. Two-sided Mann Whitney test. **C** Expression score of TEAD-target genes (Left) and abundance of deconvoluted ECM-myCAF per spot (Right) in Visium sections. Scale bars, 1 mm. $n = 13\,438$ Treatment naïve spots; 24,205 after chemotherapy spots). **D** Expression score of TEAD-target genes in stromal spots ($n = 2410$ spots). $N = 4$ Treatment-naïve; 6 after chemotherapy sections. Two-sided Wilcoxon test. **E** YAP1 staining in paired HGSOC. Scale bars, 50 μm. **F** YAP1 H-scores in stroma (Left) and epithelium (Right) ($N = 35$ patients). Two-sided paired Wilcoxon test. **G** Correlation plots with linear regression lines between H-scores of FAP and YAP1 in the stroma before (Left) and after chemotherapy (Right). Each dot represents one tumor ($N = 35$ patients). Two-sided Spearman correlation test. **H** Same as in (**G**) for ANTXR1 and YAP1 in stroma. **I** Representative views of nuclear YAP1 staining in paired HGSOC. Scale bars, 50 μm. **J** Left, % of CAF with nuclear YAP1 staining among total YAP1⁺ CAF, Right, % of cancer cells with nuclear YAP1 staining before and after chemotherapy ($N = 35$ patients). Two-sided paired Wilcoxon test. **K** Correlations with linear regression lines between FAP H-scores and % CAF with nuclear YAP1. Each dot represents one tumor ($N = 35$ patients). Two-sided Spearman correlation test. **L** Same as in (**K**) for H-scores of ANTXR1. **M** Merged staining of serial IHC for ANTXR1 and YAP1 before and after treatment. Scale bars, 50 μm. **N** YAP1 mean intensity in ANTXR1 + CAF in paired patients ($n = 83,101$ stromal cells analyzed). Two-sided Mann-Whitney test. **O** % of YAP1⁺ and YAP1⁻ ANTXR1 + CAF in paired patients ($n = 83\,101$ stromal cells analyzed). Two-sided Fisher's exact test. In boxplot the center line, box limits and whiskers indicate the median, upper and lower quartiles and 1.5x interquartile rage.

Fig. S7E). Consistent with these observations, CD8⁺ T lymphocytes, which have been co-cultured with ECM-myCAF silenced for CYR61, showed an increased capacity to kill cancer cells (Supplementary Fig. S7F), suggesting that the secreted factor CYR61, a well-known YAP1-target gene, might mediate ECM-myCAF immunosuppressive activity on CD8⁺ T cells. Taken as a whole, these data suggest that the down-regulation of YAP1 and downstream TEAD-signaling pathway observed in ECM-myCAF after chemotherapy might be key in CD8⁺ T cell enrichment in HGSOC patients.

## Discussion

Although stromal heterogeneity prior to treatment is now well established, little is known about the impact of chemotherapy on CAF diversity in HGSOC. Here we show that chemotherapy re-shapes the content in FAP + CAF (CAF-S1), which can modulate cytotoxic CD8⁺ T cell density through a YAP1-dependent mechanism (see Summary, Fig. 7). While chemotherapy increases stromal abundance[92,106–108], we found that chemotherapy also deeply modifies the composition of CAF populations in HGSOC. Indeed, chemotherapy reduces the proportion of the CAF-S1 (FAP⁺ SMA⁺) and CAF-S4 (FAP⁻ SMA⁺) myofibroblast populations to the benefit of normal-like fibroblasts. Similarly, chemotherapy was shown to repress the expression of FAP in CAF in PDAC[109], confirming the impact of treatment on CAF-S1 in another tumor type. As we detected the different FAP⁺ CAF-S1 clusters -first identified in breast cancer[82]- in HGSOC, we wondered if chemotherapy could exert a differential impact on these CAF-S1 clusters. We first observed that chemotherapy increases the proportion of ANTXR1⁻ iCAF, while the content in ANTXR1⁺ myCAF clusters decreases. The content in iCAF was also shown to be significantly increased in PDAC and colorectal cancer after treatment[95,110,111]. Moreover, a recent scRNAseq analysis from ovarian cancer revealed that chemotherapy increases the prevalence of a stress phenotype in cancer cells that is associated with an inflammatory stroma[99]. Similarly, chemotherapy-treated breast cancer cells reprogram CAF toward an inflammatory state[72,112]. By combining several complementary approaches, including IHC, flow cytometry, scRNAseq and spatial transcriptomics, we confirmed the increase in ANTXR1⁻ iCAF after treatment and identified the Detox-iCAF cluster as systematically detected after treatment in scRNAseq datasets analyzed from independent cohorts of HGSOC and BC patients. Moreover, we went a step further by highlighting that, among all ANTXR1⁺ myCAF clusters, the content in the ECM-myCAF cluster decreases the most after treatment. We validated this decrease in two independent cohorts of HGSOC patients, as well as in BC patients, showing this observation is relevant in different cancer types. Consistent with our findings, maintenance of a CAF signature composed of genes enriched in ECM-encoding genes is associated with poor overall survival in PDAC patients[95]. Similarly, resistance to treatment is associated with the abundance of myofibroblastic CAF after treatment in several cancer types[108,113–119]. We observed that platinum-resistant HGSOC patients with high residual myofibroblast content

after chemotherapy relapse earlier and survive less than patients with normal-like fibroblasts, highlighting the importance of the decrease of myCAF content after treatment. Moreover, we show that the decrease in the ECM-myCAF content and the increase in the Detox-iCAF proportion following chemotherapy were mainly detected in responder patients but not in non-responder patients of the SCANDARE cohort. Previous studies revealed that myCAF and iCAF exhibit a certain degree of plasticity according to culture conditions[50,57,88,110,120]. Taxanes- and platinum salts-based chemotherapy may reshape CAF-S1 cluster phenotype and promote the conversion of myCAF into iCAF. Nevertheless, we cannot exclude that the decrease in ECM-myCAF content after chemotherapy does not result from an increased apoptosis or a lower proliferation rate of this population, compared to iCAF.

CAF activation by chemo- and radiotherapy improves the ability of cancer cells to acquire stemness properties, and promotes tumor growth and invasion[92,108,112,116,121–123]. Here, we highlight another pathophysiological consequence on immunosuppression of high ECM-myCAF content in post-treatment HGSOC patients. Effective therapy is known to re-activate tumor-targeting immune responses by increasing cancer cell immunogenicity and inhibiting immunosuppressive circuitries[83,119,124–128]. However, the impact of chemotherapy on the different CAF populations, and in turn, their effect on T cell content after chemotherapy are poorly understood. Based on the dose-response relationship between CD8⁺ TIL and HGSOC survival[81], understanding factors that increase cytotoxic CD8⁺ T cell density after treatment is key to unravel outcome diversity in HGSOC patients. Here, we show that CD8⁺ T lymphocyte enrichment upon chemotherapy is associated with the reduction of the ANTXR1⁺ ECM-myCAF cluster. Indeed, the more this ANTXR1⁺ cluster is maintained after treatment in HGSOC, the less CD8⁺ TIL re-infiltrate the tissue. Chemotherapy was reported to down-regulate TGFβ-signaling pathway in CAF[109,129], while TGFβ upregulation in CAF is associated with immune evasion[83,88,94,108,119,125,130–132]. Here, besides TGFβ, we identified the YAP1/TEAD-dependent pathway in ECM-myCAF as a mediator of cytotoxic T-cell exclusion. YAP1-signalling pathway is associated with various hallmarks of oncogenesis in many cancer types, included HGSOC[133–136]. Although YAP1 functions in both cancer and immune cells are well-established[133–139], our study highlights its differential expression and functions in CAF populations. We observed that YAP1/TEAD-signaling pathway is specifically up-regulated in the ECM-myCAF cluster. Moreover, after chemotherapy, high YAP1 staining in this CAF population is associated with reduced CD8⁺ T cell density, while this is not the case for tumor cells. YAP1 function is required for CAF to remodel ECM production and increase tumor stiffening[133,140–142], which is consistent with the potential role of YAP1 in stroma as a physical barrier reducing T cell content. In addition, we show that YAP1 in ECM-myCAF also acts directly on CD8⁺ T lymphocytes by inhibiting their cytotoxicity. Thus, our study adds another layer of interest in YAP1 functions by highlighting its high expression in ECM-myCAF and its function in the regulation of cytotoxic CD8⁺ T lymphocyte content after

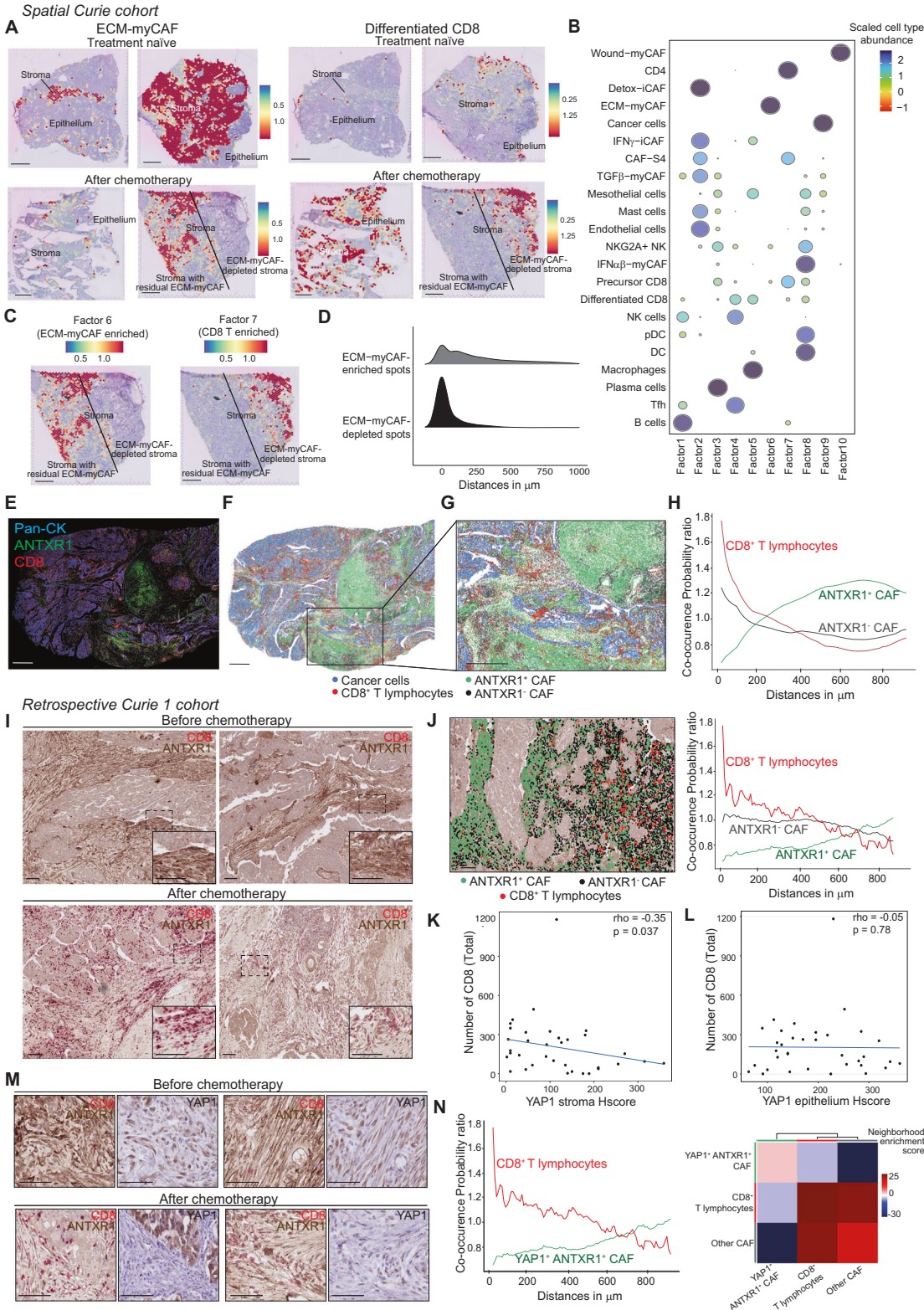

chemotherapy. In conclusion, although it might be also interesting to investigate the function of TAZ after chemotherapy, our study suggests that YAP1 might be a promising therapeutic target in combination with chemotherapy for HGSOC treatment.

Spatially-defined TME anchored in stromal plasticity and tumor immunity at time of diagnosis has recently been discovered in PDAC and associated with chemoresistance[143]. To better capture the impact

of chemotherapy on spatial distribution of TME components, we performed deconvolution of spatial transcriptomic data from pathology-annotated HGSOC sections. We characterized the spatial heterogeneity of the CAF-S1 clusters and CD8+ T cells, as well as YAP1/TEAD-signaling, by integrating histological features and transcriptomics on paired HGSOC sections before and after chemotherapy. As shown by IHC, flow cytometry and scRNAseq analyses, we

**Fig. 5 | ECM-myCAF segregate away from cytotoxic CD8 + T lymphocytes.**
**A** Abundance of deconvoluted ECM-myCAF (Right) and differentiated CD8[+] TILs (Left) per spot in Visium sections. The bottom right section shows distinct pathological responses after treatment, one non-responding with high proportion of residual ECM-myCAF (Left part) and one responding with a reduced ECM-myCAF content (Right part). **B** Non-negative matrix factorization of the deconvolution output on 4 Treatment-naïve; 6 after chemotherapy Visium sections. Colors and sizes of circles indicate the scaled cell type abundance. **C** Spatial distribution and score intensity of ECM-myCAF- and CD8[+] T cell-enriched factors (Factor 6 and 7) on a HGSOC sample after treatment. **D** Distribution of the closest distances between CD8-enriched spots and ECM-myCAF-enriched or depleted spots ($n = 10$) in a radius of 1 mm. **E** Co-staining of Pan-cytokeratin, ANTXR1 and CD8 in a HGSOC section after treatment. Scale bar = 1 mm. **F** Cell segmentation of the section shown in (**E**). Colors represent clustering results identifying cancer cells (blue), CD8[+] TILs (red), ANTXR1[+] (green) and ANTXR1[-] CAF (black). Scale bars = 1 mm. **G** Same as in (**F**) area considered for co-occurrence analysis shown in (**H**). **H** CD8[+] TILs co-occurrence probability ratio (*See Methods*) with ANTXR1[+] and ANTXR1[-] CAF in (**G**). **I** ANTXR1 (brown) and CD8 (red) IHC co-staining in paired HGSOC sections. Scale bars, 50 μm. **J** Left, HGSOC residual sample segmentation. Scale bar = 50 μm. Right, CD8[+] TILs co-occurrence probability ratio with ANTXR1[+] and ANTXR1[-] CAF in the Left section. Correlation plot with linear regression line between H-score of YAP1 in the stroma (**K**) or in the epithelium (**L**) and the number of total CD8[+] TILs in HGSOC after chemotherapy. Each dot represents one tumor ($N = 35$). Two-sided Spearman correlation test. **M** ANTXR1 (brown) and CD8 (red) IHC co-staining with YAP1 stained on serial paired HGSOC sections ($N = 2$ paired HGSOC). Scale bars, 50 μm. **N** Left, Co-occurrence probability ratio between CD8[+] T lymphocytes and residual YAP1[+] ANTXR1[+] CAF in HGSOC after treatment. Right, Neighbors enrichment score computed on the spatial connectivity graph between CD8[+] T lymphocytes, YAP1[+] ANTXR1[+] CAF, and other CAF.

---

validated spatially and transcriptionally the CAF-S1 cluster variation before and after treatment. Indeed, ECM-myCAF decreases after chemotherapy, while iCAF are enriched in treated TME. Importantly, the combination of these biological and computational approaches enabled us to highlight that the specific decrease of YAP1/TEAD-dependent signaling in ECM-myCAF-enriched areas after chemotherapy is associated with CD8[+] T cell content within the TME. In conclusion, by combining IHC, flow cytometry, scRNAseq, spatial transcriptomics and functional assays, we provide here a full characterization of the quantity, diversity and spatial localization of CAF populations before and after treatment in HGSOC. In addition, we highlight the role of YAP1 expressed by the ECM-myCAF cluster on CD8[+] T cell cytotoxicity following chemotherapy.

## Methods
### Cohorts of HGSOC patients: inclusion and ethics
For both retrospective and prospective cohorts of patients, there was no interference with standard clinical practice. Samples were available for research use and not needed for diagnosis. Analysis of tumor samples was performed according to the relevant national law providing protection to people taking part in biomedical research. All patients included in the retrospective Curie 1 cohort were diagnosed before 2014. At that time, the principle of non-opposition was the French legal requirement. In line with this, all patients treated at Institut Curie were informed orally and through an informative flyer, that their biological samples, collected through standard clinical practice, could be used for research purposes, and that by not opposing this use, they accept it. A certificate attesting that patients have read the booklet is signed by each patient and included in her medical record. Patients' refusal, expressed either orally or written, was considered and excluded from our study. For the prospective cohorts of patients taking in charge from 2017/2018, we aligned ourselves with the General Data Protection regulation (GDPR) when it was voted in France in 2019, Institut Curie provides a systematic explicit consent form that every patient needs to sign before the samples are used for research. Then, all HGSOC patients included in prospective cohorts in our study signed this consent. The Biological Resource Centre (BRC) is integrated to the Pathology Department headed by Dr. A. Vincent-Salmon. BRC is authorized to store and manage human biological samples according to French legislation. The BRC has declared defined sample collections that are continuously incremented as and when patient consent forms are obtained (AC-2021-4366 authorization number from the French Health Ministry). The BRC follows all currently required national and international ethical rules, including the declaration of Helsinki. The BRC has also been accredited with the *AFNOR NFS-96-900 quality label*. In addition, the BRC collections have been declared to the *CNIL* (*Approval Nb: 1487390 delivered February 28th, 2011*). All data collected were made pseudo-anonymous for further analyses and privacy was therefore protected. Finally, the

Institutional Review Board and Ethics committee of the Institute Curie Hospital Group approved all analyses realized in this study (*Approval of the Tumour Micro-environment project given February 12th, 2014*), as well as the National Commission for Data Processing and Liberties (*Approval Nb: 1674356 delivered on March 30, 2013*), authorizations obtained by Dr. F. Mechta-Grigoriou. Finally, the Committee for the Protection of Persons (CPP) expressed a favorable opinion to Dr. F. Mechta-Grigoriou's studies on tumor heterogeneity and plasticity (*Approval Nb: ID-RCB: 2020-A00048-31, November 3rd, 2020*).

### Retrospective Curie 1 cohort information
We selected patients with an HGSOC treated with a sequence of laparoscopy for diagnostic (pathological analysis on surgical samples) and staging, neoadjuvant chemotherapy, and then interval debulking surgery followed by adjuvant chemotherapy. We collected and embedded into paraffin tumors at diagnosis and at interval debulking surgery (incomplete pathological response to chemotherapy). 35 HGSOC patients diagnosed between 2000 and 2014, treated with neoadjuvant chemotherapy, followed at the Institute Curie Hospital Group, and exhibiting in term an incomplete pathological response to chemotherapy (microscopic or macroscopic tumor residue at the interval debulking surgery) were selected for inclusion on this study. 70 samples before-and-after neoadjuvant chemotherapy from HGSOC patients have been collected as part of routine standard of care and analyzed by immunohistochemistry (IHC). HGSOC samples referred to as "before chemotherapy" were collected during the diagnostic surgery prior to any treatment either at time of diagnosis or before neoadjuvant treatment. Samples referred to as "after chemotherapy" were collected at time of interval debulking surgery for all patients. Neoadjuvant chemotherapy was based on platin and taxane conventional treatment for HGSOC patients. Clinical data were collected in medical reports. When it was available, the germinal BRCA status was also collected in medical reports. Overall survival and disease-free survival have been defined as followed: overall survival is defined as the time from diagnostic to death or last follow-up of the patient. Disease-free survival is the period of survival without regional or distant relapse or death, meaning the period of time going from the last platinum injection of adjuvant chemotherapy to the first event occurring among regional or distant relapse or death. Relapse at 6 months was defined as: date of relapse (progression or metastasis or death if it occurred before relapse) – date of the last platinum salt injection (Disease recurrence within six months of the last chemotherapy administration. If the event appears before 6 months: relapse = yes, otherwise relapse = no.

### Retrospective SCANDARE Curie 2 cohort information
SCANDARE (NCT03017573) is a prospective biobanking study in which tumor tissues were collected after patients' consent has been obtained. SCANDARE was approved by a national ethics committee (CPP Ile-de-France 3) and the French National Agency

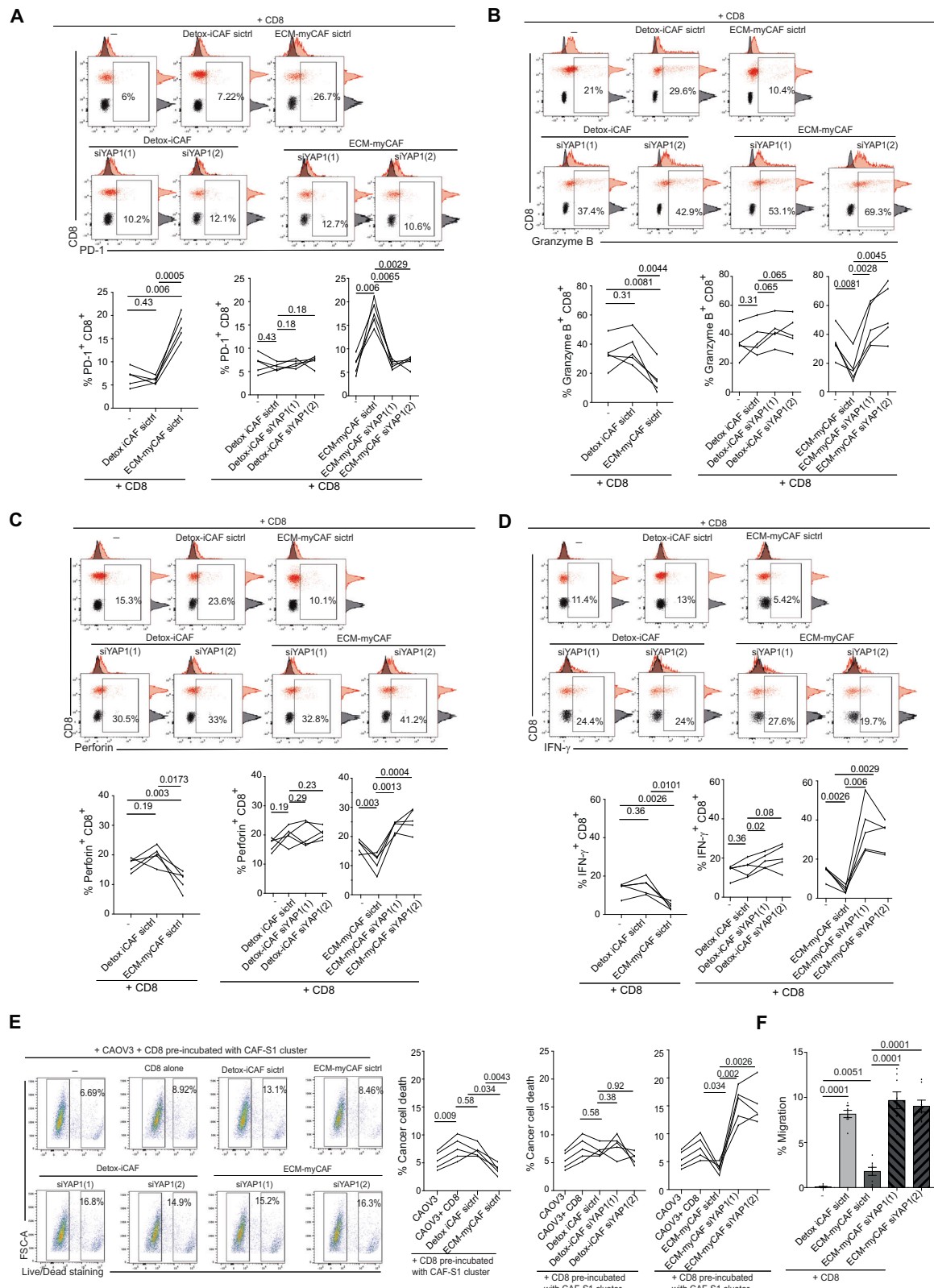

for the Safety of Medicines and Health Products (ANSM) in June 24th, 2016. By participating in SCANDARE, patients receive standard treatments and agree to get additional sampling done along their disease evolution. Patient inclusions started from April 2017. 70 tumor tissues (45 before and 25 after treatment) were obtained from 48 ovarian patients treated at Institut Curie. All patients received standard treatment according to the stage of the disease and usual procedures. Efficacy of treatment was assessed by CT scan according to RECIST v.1.1 criteria or PET scan, defining complete response, partial response, patients with stable disease and patients showing tumor progression. Patients with complete or partial response were considered as responders, patients with stable or progression disease were considered as non-responders.

**Fig. 6 | ECM-myCAF dampen CD8⁺ T lymphocyte cytotoxicity through a YAP-1 dependent mechanism. A** Up, Representative flow cytometry plots showing CD8 and PD-1 protein levels in control condition (CD8⁺ T cells alone) (−) or co-cultured with Detox-iCAF or ECM-myCAF primary fibroblasts transfected with an untargeted siRNA (siCtrl) or with two different siRNA targeting YAP1 (siYAP1(1), siYAP1(2)). The population of interest (CD8⁺ PD-1⁺) is represented in red and the isotype control in black. Bottom, Bar plots showing the % of PD-1⁺ T cells among CD8⁺ T lymphocytes alone or in presence of Detox-iCAF or ECM-myCAF (Left) and transfected with siCtrl or siYAP1 (Middle and Right). Data are mean ± SEM (*n* = 5 independent experiments). *P*-values from paired Wilcoxon test. **B**–**D** Same as (**A**) for Granzyme B⁺ (**B**), Perforin⁺ (**C**) and IFN-γ⁺ (**D**) CD8⁺ T lymphocytes. *P*-values from paired Student

*t*-test. **E** Left, Representative flow cytometry plots showing CAOV3 cell death after 24 h of incubation with CD8⁺ T lymphocytes pre-incubated with Detox-iCAF or ECM-myCAF primary fibroblasts transfected with an untargeted siRNA (siCtrl) or with two different siRNA targeting YAP1 (siYAP1(1), siYAP1(2)). Right, Bar plots showing the % of cancer cell death after incubation with CD8 + T lymphocytes. Data are mean ± SEM (*n* = 5 independent experiments). *P*-values from paired Student *t*-test. **F** Bar plots showing the % of migration of CD8 + T lymphocytes after 24 h of transwell co-culture with Detox-iCAF or ECM-myCAF primary fibroblasts transfected with an untargeted siRNA (siCtrl) or with two different siRNA targeting YAP1 (siYAP1(1), siYAP1(2)). Data are mean ± SEM (*n* = 8 independent experiments). P-values from unpaired Student t-test.

## Cohort for spatial transcriptomic analyses

10 tumor tissues, including 4 treatment-naïve samples and 6 samples after chemotherapy, were obtained from 8 HGSOC patients treated at Institut Curie. All patients received standard treatment according to the stage of the disease and usual procedures.

## Prospective cohorts 1 and 2 information

Prospective cohort 1 was dedicated to flow cytometry analyses and Prospective cohort 2 to single cell RNA sequencing. Fresh samples from the two prospective cohorts were collected at time of surgery either from treatment-naïve or from chemotherapy-treated HGSOC patients, as part of routine standard of care. Prospective cohort 1 is composed of 20 HGSOC patients, including 8 untreated and 12 treated patients, and Prospective cohort 2 of 12 HGSOC patients, including 5 treatment-naïve samples and 7 samples collected after chemotherapy. Definitive inclusion of patients in these prospective cohorts was confirmed after complete evaluation by a pathologist. For the prospective cohorts of patients taking in charge from 2017/2018, we 1) aligned ourselves with the General Data Protection regulation (GDPR) when it was voted in France in 2019, Institut Curie provides a systematic explicit consent form that every patient needs to sign before the samples are used for research or 2) included patients enrolled in the SCANDARE bio-banking study (NCT03017573).

Clinical features of retrospective and prospective cohorts are detailed in Tables 1 and 2.

## Sex and gender reporting

All patients suffering from HGSOC and analyzed in our study are women, a variable which has been self-reported by patients.

## Flow cytometry on HGSOC samples

20 fresh human HGSOC primary tumors were obtained directly from the operating room and cut into small fragments by a mechanical and enzymatic digestion in $CO_2$-independant medium (Gibco #18045-054) supplemented with 2 mg/ml collagenase I (Sigma #C0130), 2 mg/ml hyaluronidase (Sigma #H3506) and 25 mg/ml DNaseI (Roche #11284932001) during 45 min (min) at 37 °C with shaking (180 rpm). After tissue digestion, cells were filtrated through a 40 µm cell strainer (Falcon #352340) and resuspended in PBS+ solution (PBS, Gibco #14190; EDTA 2 mM, Gibco #15575; Human serum 1%, BioWest #S4190-100). Cells were next separated into two groups for analyzing panels of CAF subsets and immune cells, separately. For characterization of CAF populations, cells were stained with an antibody mix for the detection of both cell surface and intracellular staining containing anti-EPCAM-BV605 (1:50; BioLegend, #324224), anti-CD45-APCCy7 (1:20; BD Biosciences, #BD-557833), anti-CD31-PE/Cy7 (1:100; BioLegend, #303118), anti-CD235a-PerCP5.5 (1:50; Biolegend, #349110), anti-FAP-APC (1:100; R&D Systems, #MAB3715) conjugated with the fluorescent dye Zenon APC Mouse IgG1 labeling kit (1/100, ThermoFisher Scientific #Z25051), anti-CD29-Alexa700 (1:100; BioLegend, #303020) and anti-SMA-Alexa594 (1:25; R&D Systems, #IC1420T-025), anti-FSP1-PE (1:25; BioLegend, # 370004) and anti-

ANTXR1-AF405 (1:25; Novus Biologicals, #NB100-56585AF405). For each CAF marker, the isotype control antibody was: iso-anti-FAP (primary antibody, 1:200, R&D Systems, #MAB002), iso-anti-CD29 (1:100, BioLegend, #400144), iso-anti-SMA (1:25, R&D Systems, #IC003T), iso-anti-FSP1 (primary antibody, 1:20, BioLegend, #400139) and iso-anti-ANTXR1 (1:25; Novus Biologicals, #IC003T). For surface staining, cell suspensions were stained immediately after dissociation of samples during 15 min at RT with the antibody mix in PBS+ solution and 2.5 µg/ml DAPI (Thermo Fisher scientific, #D1306) was added just before flow cytometry analysis. For intracellular staining, cells were stained with a violet live/dead marker (1:1000, Thermofisher Scientific #L34955) for 10 min at room temperature (RT) in PBS (Gibco #14190) to exclude dead cells and then fixed in 4% paraformaldehyde (PFA) (Electron Microscopy Sciences, #15710) for 20 min at room temperature (RT). After a rapid washing step with PBS+ solution (PBS supplemented with EDTA 2 mM and 1% Human serum), cells were stained with an antibody mix during 30 min at room temperature (RT). Antibodies are suspended in PBS+ solution with 0.1% of Saponin (Sigma-Aldrich #S7900) and corresponding isotype control mix antibodies are used for each experiment. For surface and intracellular staining, signals were acquired on the LSRFortessa analyzer (BD biosciences) for flow cytometry analysis. At least $5 \times 10^5$ events were recorded. Compensations were performed using single staining on anti-mouse IgG and negative control beads (BD biosciences, #552843) for each antibody. Data analysis was performed using FlowJo version 10.4.2 (LLC). Cells were first gated based on forward (FSC-A) and side (SSC-A) scatters (measuring cell size and granularity, respectively) to exclude debris. Dead cells were excluded based on their positive staining for Live/Dead. Gating included EPCAM⁻, CD45⁻, CD31⁻, CD235a⁻ cells, to remove epithelial (EPCAM⁺), hematopoietic (CD45⁺), endothelial (CD31⁺) and red blood cells (CD235a⁺). Cells from the negative fraction were next examined using CAF markers, including FAP, CD29, αSMA, FSP1 and ANTXR1. Each cell population was selected using the same gating strategy for all the patients' samples analyzed by flow cytometry. Minor variations in the fluorescence intensity of CD45⁺ and EPCAM⁺ populations could be observed in acquisitions over the few years that were required to collect all samples of the Prospective cohort 1. For characterization of T lymphocytes, the protocol was the following: Among the 20 HGSOC samples analyzed for CAF subsets, 16 were characterized at the meanwhile for the immune content. HGSOC fresh tissues were collected and digested, as described above. Cells were stained with an antibody mix for the detection of cell surface staining containing anti-CD45-APC-Cy7 (1:20; BD Biosciences, #557833), anti-CD3-AlexaFluo700 (1:40; BD Biosciences, #557943), anti-CD8-PE/Alexa610 (1:80; Thermofisher, #MHCD0822) and anti-CD4-APC (1:20; Miltenyi Biotec, #130-113-210). Cells were stained with a violet live/dead marker (1/1000, Thermofisher Scientific #L34955) for 10 min at RT in PBS (Gibco #14190) to exclude dead cells. Cells were incubated for 20 min at RT with the antibody mix in PBS + solution. CD45⁺ cells were detected by flow cytometry, and data analysis performed using FlowJo version 10.4.2 (LLC), as described above.

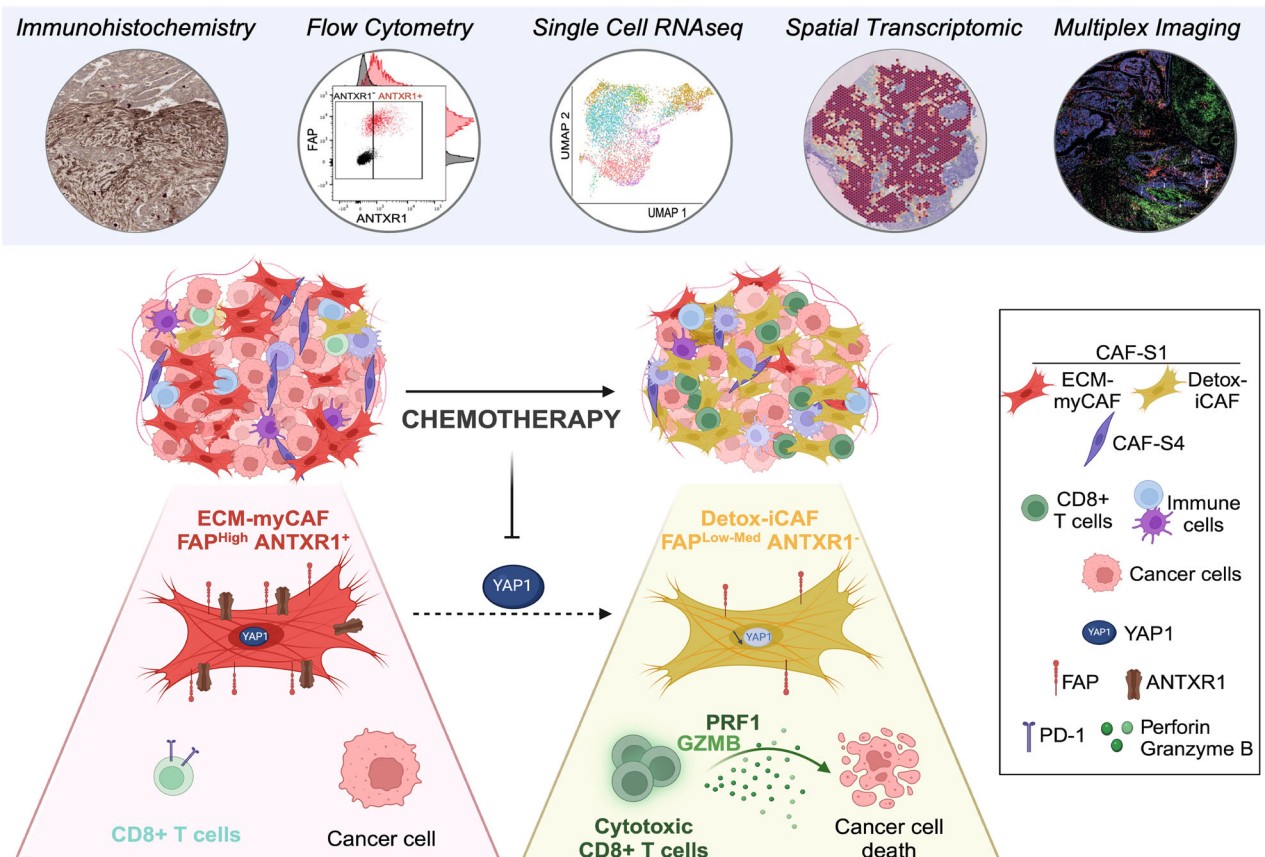

**Fig. 7 | Summary: impact of chemotherapy on CAF-S1 and CD8 T cell density in HGSOC.** HGSOC TME is heterogeneous and composed of different CAF subsets, including the FAP⁺ CAF-S1 and FAP⁻ CAF-S4 myofibroblastic populations. Chemotherapy globally increases the stromal content, but reduces the proportions of both CAF-S1 and CAF-S4 myofibroblasts. The combination of several high-throughput technologies shows that, among the CAF-S1 population, the content in the ANTXR1⁺ ECM-myCAF cluster decreases the most following chemotherapy, while the proportion of the ANTXR1⁻ detox-iCAF cluster increases. Concomitantly, chemotherapy increases CD8⁺ T lymphocyte density. Indeed, ECM-myCAF reduce CD8⁺ T cell cytotoxicity through the YAP1 signaling pathway, thereby explaining that the more the content in ECM-myCAF decrases after chemotherapy, the more CD8⁺ T cell infiltrate the tumor upon treatment. Figure 7 was created with Biorender.com.

The list of the antibodies and the respective dilutions used are detailed in Supplementary Table S1.

## Immunohistochemistry (IHC) staining on HGSOC sections

A total of 70 HGSOC samples, as representative sections of the tumor, were selected by pathologists. Serial sections of paraffin-embedded tissues of 3 µm thickness were stained on the Lab Vision Autostainer (Thermo Fisher Scientific). Dewaxing and antigen retrieval on slides prior to immunohistochemical staining were performed with EnVision FLEX Target Retrieval Solution (high- or low-pH, as required -see Supplementary Table S1- Dako, #K800421 or #K800521) on the Lab Vision PT Module (Thermo Fisher Scientific). Antigen detection was done using the streptavidin-peroxidase protocol (Vectastain ABC kit; Vector Labs #PK-6101/6102/6104) plus detection with 3,3′-diaminobenzidine for 5 min (DAB, Dako, #K3468). List of antibodies and respective conditions used are detailed in Supplementary Table S1. Counterstaining was performed with Mayer hematoxylin freshly prepared (Dako, #S3309). Tissue sections were then submitted to serial gradients of xylen and mounted with coverslip in an automatic device (Sakura, Tissue-Tek DRS). Epithelial and stromal content was first evaluated by clinicians based on morphological criteria on Hematoxylin and Eosin (H&E) stained sections considering that epithelium and stroma areas correspond to 100% of tissue. Ovarian cancer cells were characterized by a stratified, disorganized and enlarged epithelial cells with atypical nuclei morphology, following the OMS criteria[144–146]; stroma appeared as a typical elongated shape compartment. In addition, we performed specific staining of each compartment. The following antibodies and respective conditions were used: anti-EPCAM (pH=6, 1 h, 1:200, Dako #M0804), anti-FAP (pH=6, 1 h, 1:150, Vitatex #MABS1001), anti-CD29 (pH=6, 1 h, 1:150, Abcam #ab3167), anti-FSP1 (pH=6, 1 h, 1:450, Abcam #ab27957), anti-αSMA (pH=6, 30 min, 1:350, Dako #M0851), anti-YAP1 (pH=9, 1 h, 1:800, Cell Signaling, #14074) and anti-ANTXR1 (pH=9, 20 min, 1:400, Abcam #ab241067). Counterstaining was performed with Mayer hematoxylin freshly prepared (Dako, #S3309). Tissue sections were then submitted to serial gradients of xylen and mounted with coverslip in an automatic device (Sakura, Tissue-Tek DRS). Co-immunohistochemistry using anti-CD8 and anti-ANTXR1 antibodies was performed on a LEICA BOND RX automate. Leica epitope retrieval 2, pH=9 (LEICA, #AR9640) was used prior to each staining. Anti-CD8 antibody (30 min, 1:200, Dako, #M7103) and anti-ANTXR1 (60 min, 1:400, Abcam #ab241067) were revealed using BOND Polymer Refine RED Detection kit (LEICA, #DS9390) and BOND Polymer Refine Detection kit (LEICA, #DS9800) respectively, according to manufacturer instructions.

## Quantification of IHC staining

For each slide, staining of EPCAM (marker of cancer epithelial cells) and CAF markers was evaluated as a histological score (H-score)

defined by staining intensity (from 0 to 4) multiplied by the percentage of stained cells (either epithelial cells or fibroblasts) (from 0% to 100%). The YAP1 H-score was established independently in the stromal and epithelial part of the tumor as a result of % ((YAP1$^+$ CAF in the stroma) x (Intensity of YAP1$^+$ staining in the stroma)) and % ((YAP1$^+$ CAF in the epithelium) x (Intensity of YAP1$^+$ staining in the epithelium)). To evaluate YAP1 nuclear staining, we defined the proportion of cells (considering either CAF or cancer cells) with nuclear YAP1 among total YAP1$^+$ cells (either CAF or cancer cells). To measure YAP1 nuclear staining in CAF and cancer cells, respectively, we applied the following product: ((% of CAF with nuclear YAP1 staining) * (% of YAP1$^+$ CAF among total CAF)) / 100 and ((% of cancer cells with nuclear YAP1 staining) * (% of YAP1$^+$ cancer cells among total cancer cells)) / 100. For quantification of each marker, the whole section was considered and evaluated. For quantification of T lymphocytes, five representative areas of 0.105 mm$^2$ for each slide and per each immune marker were evaluated. The number of CD3$^+$, CD8$^+$ or FOXP3$^+$ T lymphocytes were evaluated in each compartment (considering either cancer cells, CAF or both compartments). The evaluation was assessed by the following product: (Number of T lymphocytes (CD3$^+$, CD8$^+$ or FOXP3$^+$ cells) in the epithelium or stroma / total area of the section). The number of T lymphocytes (CD3$^+$, CD8$^+$ and FOXP3$^+$) were next standardized per surface unit of stroma or epithelium. To evaluate the number of T lymphocytes in the stroma or epithelium, we counted the number of T cells (considering either cancer cells or CAF) divided by the area of the section (either cancer or CAF area). The evaluation was assessed by the following product: (Number of T cells (CD3$^+$, CD8$^+$ or FOXP3$^+$ cells) in the epithelium or stroma / the area of the epithelium or stroma in the section). The evaluation of histological scores (H-scores) of all CAF markers and the quantifications of immune cells were carried out in a non-blinded manner by two independent researchers, including a pathologist. All quantifications gave very consistent results regardless of the person.

### Pan-cytokeratin, CD8 and ANTXR1 multiplex immuno-fluorescence imaging and colocalization analysis

HGSOC section of paraffin-embedded tissues of 3 µm thickness was used for multiplex analysis. Dewaxing and antigen retrieval on slide prior to staining were performed with EZ-AR 2 Elegance AR buffer solution (EDTA based pH 8.5 – BioGenex, #HK547-XAK) for 5 min at 95 °C followed by 5 min at 107 °C. Slide was first stained with anti-ANTRX1 primary antibody (Abcam, #ab246321) for 1 h at room temperature followed by argofluor conjugated secondary antibody (Rarecyte, #52-1025-602) incubation for 30 min at room temperature and then with argofluor conjugated anti-pan-cytokeratin and anti-CD8 antibodies (Rarecyte, #52-1015-801 and #52-1048-501) for 2 h at room temperature. Slide was then rinsed and incubated with Hoechst 33342 (Thermo Fisher, #H3570). The whole slide was scanned on the Orion instrument (RareCyte). After importing the whole slide image in QuPath v0.4.3[147], cell segmentation was done using the Cell detection method with default parameters. After measuring the intensity of the markers in each cell, the cell detections and associated features were exported as a GeoJSON file. A anndata object was created using the python package scanpy for subsequent clustering with the function scanpy.tl.leiden with a resolution of 0.1. Cell type identification was done by identifying upregulated proteins in each cluster and by overlaying the segmentation on the image. Co-occurrence probability of clusters was computed using the function gr.co-occurrence from the Squidpy package[105] and plotted with the function pl.co-occurrence according to the CD8 cluster. The co-occurrence probability corresponds to the probability to observe a cell type of interest with a CD8$^+$ T lymphocyte reported to the probability to find the cell type of interest, computed at increasing distance. Cancer cells were removed from the plot after computing the co-occurrence probability ratio for a simplified presentation of the results.

### In silico analysis of colocalization of IHC staining on serial sections

Regions of interest from either serial IHC stained for ANTXR1 and YAP1 or serial IHC stained for both ANTXR1/CD8 and YAP1 were registered using the Fiji plugin BUnwarpJ by manually identifying common features. Registered images were imported in QuPath and epithelial regions were masked by training a pixel classification model as implemented in QuPath. The stromal region was then segmented using Cell Detection on the Hematoxylin channel with default parameters. After quantifying the DAB staining intensity in the first image (YAP1), the cell detections were then transferred to the other registered image to quantify -in the same cell- the subsequent staining using the DAB channel (for ANTXR1), or DAB and Fast Red for ANTXR1 and CD8 co-staining respectively. Measurements were then exported and analyzed in R to plot YAP1 intensity in ANTXR1$^+$ CAF and the proportion of YAP1$^{+/-}$ ANTXR1$^+$ CAF, or exported in python for colocalization analysis. Cell mean DAB staining intensity of ANTXR1 and YAP1 were used to separate the segmented cells into YAP1$^{+/-}$ and ANTXR1$^{+/-}$ cells. Threshold for positivity was assessed using the Single Measurement classifier tool in QuPath. Co-occurrence probability of clusters was computed using the function gr.co-occurrence from the Squidpy package[105] and plotted with the function pl.co-occurrence according to the CD8 + T cell cluster, other cell types than the one of interest were removed from the plot after computing the co-occurrence probability ratio for a simplified presentation of the results. Neighborhood enrichment analysis was conducted after building a connectivity matrix with the function gr.spatial_neighbors, computed with gr.nhood_enrichment and plotted with pl.nhood_enrichment from the Squidpy package, with default parameters. Merged staining of ANTXR1 and YAP1 was performed by deconvoluting the DAB staining of the registered region of interest (ROI) with the Color deconvolution method integrated in Fiji[148]. ANTXR1 and YAP1 were colored in red and green, respectively, and the lookup table was inverted in Fiji to highlight the staining. The images were then merge.

### Decision tree algorithm for prediction of CAF population identity

CAF enrichment for each HGSOC sample was established applying an algorithm previously developed and published, which takes as input H-scores of CAF markers[40,68,80]. In brief, this decision tree was based on marker intensity thresholds, first defined in a learning dataset on the distribution (1st quartile, median and 3rd quartile) of each marker from flow cytometry data (intensity of each CAF marker in each CAF subset) from fresh HGSOC samples and next transposed to IHC H-scores.

### Establishment of a delta-score (Δ) measuring variations of each cell population by chemotherapy

The delta score was developed using IHC quantifications in untreated and treated samples collected from the same patient. Variations are assessed by a delta score (Δ) calculated as: Variable (H-score or T cell number) after chemotherapy – Variable (H-score or T cell number) before chemotherapy. This method enabled us to investigate the impact of chemotherapy on the content of each cellular population reported to their initial abundance before treatment.

### Single cell RNA sequencing (scRNAseq) data processing: HGSOC Curie cohort

CAF populations were isolated from 13 HGSOC before or after treatment using the same digestion protocol as described above (#*Flow cytometry on HGSOC samples*). Cells were stained during 15 min at room temperature immediately after dissociation with an antibody mix containing anti-EPCAM-PE (1:50; Biolegend #324205), anti-CD45-APCCy7 (1:20; BD Biosciences, #BD-557833), anti-CD31-PE/Cy7 (1:100; BioLegend, #303118), anti-CD235a-FITC (1:50;

Biolegend #324205), anti-FAP-APC (1:100; R&D Systems, #MAB3715) conjugated with the fluorescent dye Zenon APC Mouse IgG1 labeling kit (1/100, ThermoFisher Scientific #Z25051), anti-CD29-Alexa700 (1:100; BioLegend, #303020). DAPI (2.5 μg/mL; Thermo Fisher Scientific, #D1306) was added just before flow cytometry sorting on the BDFACS ARIA III sorter (BD Biosciences). After isolation, CAF cells were immediately collected into RNase-free tubes (Thermo Fisher Scientific, #AM12450) precoated with DMEM (GE Life Sciences, #SH30243.01) supplemented with 10% FBS (Biosera, #1003/500). At least 2 to 4000 cells were collected per sample. In these conditions, cell concentration was checked in control samples and was of 200,000 cells/ml. Single-cell capture, lysis, and cDNA library construction were performed using Chromium system from 10X Genomics, with the following kits: Chromium Single Cell 3′ Library & Gel Bead Kit v2 kit (10X Genomics, #120237), v3 kit (10X Genomics, #PN-1000092) and v3.1 Dual index (10X Genomics, #1000269). Generation of gel beads in Emulsion (GEM), barcoding, post GEM-reverse transcription cleanup and cDNA amplification were performed according to the manufacturer's instructions. Targeted cell recovery was ranging from 2 to 3000 cells per sample to retrieve enough cells, while preserving a low multiplet rate. Cells were loaded accordingly on the Chromium Single cell chips, and 12 cycles were performed for cDNA amplification. cDNA quality and quantity were checked on Agilent 2100 Bioanalyzer using Agilent High Sensitivity DNA Kit (Agilent, #5067-4626) and library construction followed according to 10X Genomics protocol. Libraries were next run on the Illumina NovaSeq 6000 with a depth of sequencing of 50,000 reads per cell. Preprocessing of raw data was performed using Cell Ranger software pipeline (versions 3.1.0 and 6.0.0). This step included demultiplexing of raw base call (BCL) files into FASTQ files, reads alignment on human genome assembly GRCh38 using STAR, and counting of unique molecular identifier (UMI). Low-quality cells were defined based on the distribution of number of detected genes (non-zero count) and fraction of mitochondrial genes and filtered out. 6974 CAF from 13 HGSOC samples were then analyzed using Seurat R package (version 4.0.5). After merging, normalization (function *NormalizeData*) and scaling (function *scaleData*), integration was performed using FastMNN wrapper implemented in Seurat (function RunFastMNN). The 30 most informative principal components (PCs) were used for computing Uniform Manifold Approximation and Projection for dimension reduction (UMAP). 5,618 CAF-S1 cells from 12 patients were finally isolated based on FAP expression using CellSelector function from Seurat. Annotation of CAF-S1 clusters in HGSOC was done by using Label Transfer algorithm from Seurat (functions *FindTransferAnchors* and *TranferData)*. scRNAseq dataset of more than 18,000 CAF-S1 fibroblasts from breast cancer previously annotated[82] was used as reference and HGSOC scRNAseq dataset generated in this study as query.

### Single-cell RNA sequencing (scRNAseq) data processing: HGSOC Turku cohort

Count data from scRNAseq dataset from HGSOC before and after chemotherapy was retrieved from Gene Expression Omnibus with accession code GSE165897[99] and analyzed using Seurat pipeline. Normalization and patient integration was performed using FastMNN wrapper implemented in Seurat (function RunFastMNN). Cells annotated as stromal cells were isolated and 7964 cells annotated as CAF-1, CAF-2, CAF-3 were conserved for analysis. 5658 CAF-S1 cells were identified based on both *FAP* expression and CAF-S1 gene signature[82]. Annotation of CAF-S1 clusters was done by using Label Transfer algorithm from Seurat (functions *FindTransferAnchors* and *MapQuery*). scRNAseq dataset of more than 18,000 CAF-S1 fibroblasts from breast cancer previously annotated[82] was used as reference.

### Single cell RNA sequencing (scRNAseq) data processing: breast Cancer cohort

Publicly available scRNAseq dataset from breast cancer before and after chemotherapy[100] were retrieved from http://biokey.lambrechtslab.org and analyzed using Seurat pipeline. Cells annotated by the authors as Fibroblasts from treatment-naïve samples and samples treated by neoadjuvant chemotherapy (before anti-PD1 treatment) were isolated from the whole dataset. These cells were normalized and scaled with *nCount_RNA* added for regression. Highly variable genes were identified using *FindVariableFeatures* (nFeature=2000) and used to compute PCA. Graph-based clustering was done using the 20 top principal components at resolution 0.3. 7959 fibroblasts were defined as CAF-S1 by selecting cluster with high *FAP* gene expression. Annotation of CAF-S1 clusters was finally done by using Label Transfer algorithm from Seurat (functions *FindTransferAnchors* and *MapQuery*). scRNAseq dataset of more than 18,000 CAF-S1 fibroblasts from breast cancer previously annotated[82] was used as reference.

### Consensus non-negative matrix factorization

Identification of gene expression programs from scRNAseq data was performed using consensus Non-Negative Matrix factorization (cNMF) algorithm described in ref. 101 and implemented in Python (https://github.com/dylkot/cNMF). Range from 5 to 13 factors (K) were tested with 200 iterations for each K. Consensus estimate was obtained by setting optimal K to 10 considering the trade-off between stability and error as described by the authors. Local-density-threshold was fixed to 0.1 using diagnostic plot, default parameters were used otherwise. Heatmap and clustering (correlation distance and Ward.D2 method) was applied on the usage matrix (cells x K) and pathway analysis was performed using Metascape tool on the first 200 most contributing genes for each factor from the gene expression program matrix.

### DoRothEA analyses

Inference of transcription factor (TF) activity from the gene expression of their target (regulon) was assessed by using VIPER v1.32 and DoRothEA v1.10 R packages. In brief, the DoRothEA algorithm[104] constructs a network of TF-gene interactions based on prior knowledge from publicly available databases and literature. To increase the accuracy of the results, only Regulons with a high confidence level (A, B and C) were included in the analysis.

### Construction of HGSOC cellular Atlas

To build a HGSOC cellular atlas needed for subsequent deconvolution of bulk RNA-Seq and spatial transcriptomic data, we collected two publicly available scRNAseq datasets[45,99]. We performed additional filtering of the cells based on the number of detected genes, counts and computed mitochondrial percentages of reads. For the first dataset[99], we kept cells with more than 500 features and less than 9000 detected. Moreover, we removed cells with a mitochondrial percentage above 20%. We then performed a fastMNN integration using the Seurat wrapper function *RunFastMNN* splitting by patientID. After scaling, computing a PCA on the 2000 most variable features and a UMAP on the first 20 principal components, we annotated the major cell types based on expression of specific markers (EPCAM for epithelial cells, FAP for CAF-S1, CALB2 for mesothelial cells, IGHG1 for plasma cells, MS4A1 for B cells, CD45 for immune cells, CD3D for lymphocytes, CD68 for myeloid cells, KLRD1 for NK cells and KLRC1 for NKG2A+ NK cells). We then removed doublets using the R package scDblFinder version 1.8.0 using the function *scDblFinder* with samples set to patient ID and with default parameters. We then isolated the fibroblast cluster and did a label transfer from the CAF-S1 dataset[82] to the atlas using the Seurat functions *FindTransferAnchors* and *MapQuery* with normalization.method set to LogNormalize, reference

reduction to PCA and using the first 30 dimensions. For the annotation of lymphocytes, we did a label transfer using as reference a high resolution scRNAseq dataset of lymphocytes[149] with the same parameters described above for fibroblasts. We then grouped all T follicular helper cell clusters under the name Tfh, all CD4 annotated clusters as CD4+ T cells. CD8+ KLF2+, CD8+ GZMK+ and CD8+ XCL1+ were annotated as precursor CD8 T lymphocytes, as described in the original paper[149]. Finally, CD8+ GZMH+ and CD8+ LAYN+ were grouped under the term differentiated CD8 T cells. As two major cell types were missing from the dataset, namely the endothelial cells and the CAF-S4, we recovered a second scRNAseq dataset[45] from which we filtered out cells with less than 1000 and more than 9000 features detected, as well as cells with more than 20% of mitochondrial reads. Furthermore, we removed doublets using the same strategy than for the first dataset. We identified endothelial cells using PECAM1 expression and CAF-S4 using RGS5 expression. We then injected these cells in our atlas and not the other cell types to avoid cluster batch effect within cluster. We then rescaled the data, recomputed the variable genes, PCA and UMAP for a final total dataset of 49,909 cells and 35,546 features covering 24 different cell types.

## Bulk RNA-seq from the retrospective SCANDARE Curie 2 cohort

70 samples were processed for RNA extraction using kit (miRNeasy Mini Kit, Qiagen #217004) following the manufacturer's instructions. DNAse treatment was done after RNA extraction, following manufacturer's intructions (Macherey Nagel, #740971). RNA integrity and quality were analyzed using Agilent 4200 TapeStation system. The library was prepared following the stranded mRNA prep Ligation-Illumina protocol according to the supplier's recommendations. Briefly, the key steps of this protocol were successively, starting from 500 ng of total RNA: purification of PolyA (containing mRNA molecules) using magnetic beads attached to poly-T oligonucleotides, fragmentation using divalent cations at high temperature to obtain fragments of approximately 300 bp, cDNA synthesis, and finally ligation of Illumina adapters and amplification of the cDNA library by PCR. Sequencing was then performed on the Illumina NovaSeq sequencer (150 bp paired end). To process the data, a nextflow pipeline developed by the Institut Curie bioinformatic platform was used to process RNA-seq from raw sequencing reads to count table. Briefly, the overall quality of raw sequencing data was first checked using FastQC (v1.11.9). Reads were then aligned on a ribosomal RNAs database using bowtie (2.4.2) and on the human reference genome (hg38) with STAR (2.7.6a). Additional controls on aligned data are performed to infer strandness (RSeQC 4.0.0), complexity (Preseq 3.1.1), gene-based saturation, read distribution or duplication level using Bioconductor R package DupRadar. The aligned data were then used to generate a final count matrix with all genes and all samples.

## Deconvolution of bulk RNA-seq data

Cell type composition of 70 bulk RNA-seq data from the retrospective Scandare Curie 2 cohort were inferred using BayesPrism algorithm version v2.0. Raw count matrix of 49,909 cells from our HGSOC cellular atlas was used as input for prior information. Labels were derived from the annotations of the 24 cell types and states described above. Mitochondrial and ribosomal protein coding genes were removed as these genes are expressed at high magnitude and not informative in distinguishing cell types. MALAT1 and genes from X and Y chromosomes were also removed following indications from BayesPrism's authors. To reduce batch effects and speed up computation, we performed deconvolution only on protein coding genes. Default parameters to control Gibbs sampling and optimization were used. Final estimation of cell type fraction in each bulk RNA-seq sample was recovered using the updated theta matrix and used for downstream analysis.

## Spatial transcriptomics

Spatial transcriptomics analysis was conducted on two HGSOC samples obtained from a single patient at baseline and at resection after neo-adjuvant treatment, chosen based on their tissue structure and RNA quality (RIN ≥ 7). The Visium Spatial Tissue Optimization Slide and Reagent Kit (10X Genomics; #PN-1000193) was used according to manufacturer's instructions. Briefly, sections were fixed, stained and then permeabilized at different time points to capture mRNA, and the reverse transcription was performed to generate fluorescently labeled cDNA. The permeabilization time that resulted in the highest fluorescence signal with the lowest background diffusion was chosen. The best permeabilization time for ovarian tissue was 18 min. Cryostat sections of 10 μm of thickness were cut and placed on Visium Spatial Gene Expression slides (10X Genomics, PN-1000184). The slide was incubated for 1 min at 37 °C, then fixed with methanol for 30 min at −20 °C followed by Hematoxylin and Eosin (H&E) staining and images were taken under a high-resolution microscope. After imaging, the coverslip was detached by holding the slide in water and the slide was mounted in a plastic slide cassette. The spatial gene expression process including tissue permeabilization, second strand synthesis and cDNA amplification was performed according to the manufacturer's instructions (10X Genomics; #CG000239). cDNA quality was next assessed using Agilent High sensitivity DNA Kit (Agilent, #5067-4626). The spatial gene libraries were constructed using Visium Spatial Library Construction Kit (10X Genomics, PN-1000184). To process data, SpaceRanger software v1.2.2 (10X Genomics) was used for processing spatial transcriptomic data. The raw base call (BCL) files were demultiplexed and mapped to the GRCh38 reference genome. Loupe Browser from 10X Genomics was used to align the barcoded spot patterns, perform spots selection on the tissue and do the pathological annotation. Seurat v4.1.0 was used to perform log2 normalization, scaling, and dimension reduction on the resulting count matrices. The expression score of the TEAD-target genes was computed using the Seurat function *AddModuleScore* with default parameters. For deconvolution of the data, estimation of the abundance of each cell population in each spot of the Visium samples was achieved using cell2location version 0.1[103], implemented in Python3. To compute the reference cell type signatures, the HGSOC atlas described above was used as input to the *RegressionModel* with categorical_covariate_keys set to *patient ID* after removing the IL-iCAF and Acto-myCAF from the reference. The spatial mapping of cell types was performed by supplying each Visium section to cell2location and setting N_cells_per_location to 15 and detection_alpha to 200 after manual examination of the tissue. The number of epoch was set to 30,000. All feature plots were generated with alpha set to 0·1 and max cutoff set to "q95". For cell type quantification in situ, the sum of rounded estimated cell type number given by cell2location in each spot was computed per section. ANTXR1− CAF were computed as the sum of Detox-iCAF and IFNγ-iCAF in each spot and ANTXR1+ CAF as Wound-myCAF, ECM-myCAF, TGFβ-myCAF and IFNαβ-myCAF. At least one ECM-myCAF per spot is identified by deconvolution in 2410 spots out of 24,205 spots analyzed after treatment (average abundance of ECM-myCAF per spot = 2.3; average abundance of all cells per spot = 10). At least one CD8 + T lymphocyte per spot is identified by deconvolution in 5172 spots out of 24,205 spots analyzed after treatment (average abundance of CD8+ T cells per spot = 0.67; average abundance of all cells per spot = 11.3). To identify underlying patterns of cell types and structures in the data, Non-Negative Matrix Factorization (NMF) was applied on the merged cell abundance matrices estimated by cell2location using the R package NMF with the number of factor set to 10.

## Isolation of primary CAF-S1 fibroblasts for functional assays

Previous studies revealed that myCAF and iCAF exhibit a certain degree of plasticity according to culture conditions[50,57,88,110,120]. Based

on these previous data, we cut fresh HGSOC samples from the operating room into small pieces and incubated them either on plastic dishes (Falcon, #353003) or on dishes coated with type I collagen at a final concentration of 9 µg/ml (Institut De Biotechnologie Jacques Boy, #207050357) to maintain ECM-myCAF and iCAF identities, respectively (see # below for validation of their identities). CAF were then maintained in DMEM (Gibco #11965092) supplemented with 10% FBS (Biosera, #FB-1003/500), and 1%penicillin (100 U ml⁻¹) and streptomycin (100 µg ml⁻¹) (Gibco #15140122) at 37 °C in 5% of $CO_2$ and 1.5% of $O_2$ for at least 2 weeks to let fibroblasts spread and expand. The media was renewed 2-3 times per week. When fibroblasts reached 50% of confluency, they were detached with trypLE and plated in new plastic plates or collagen-coated plates using DMEM supplemented as above. All experiments were performed with fibroblasts until passage 10 to avoid CAF senescence. The identity of primary CAF-S1 fibroblasts in culture was next validated by flow cytometry. To do so, cells at 75% confluency were collected after trypsin treatment, resuspended in 50 µl of PBS + solution at a final concentration of $5 \times 10^4$ cells and stained for 20 min at RT using an antibody mix containing anti-FAP-APC (1:100, R&D systems #MAB3715), anti-CD29-Alexa700 (1:100, BioLegend #303020) and anti-ANTXR1-AF405 (1:25, Novus Biologicals #NB100-56585AF405), with their respective isotype control antibodies (Alexa Fluor 700 Mouse IgG1 κ Isotype Ctrl Antibody, 1:100, BioLegend, #400144, Mouse IgG1 κ Isotype Control, R&D Systems, 1:200, #MAB002, Mouse IgG1 Alexa Fluor 405-conjugated Antibody, 1:100, R&D systems, #IC002V). Cells were next analyzed on the LSRFORTESSA analyzer (BD Biosciences). At least $1 \times 10^4$ events were recorded. Compensations were performed using single staining on anti-mouse IgG and negative control beads (BD Biosciences #552843) for each antibody and on ArC reactive beads (Molecular Probes #A10346) for Live/Dead staining. Data analysis was performed using FlowJo version 10.4.2. Cells were first gated based on forward (FSC-A) and side (SSC-A) scatters to exclude debris. Single cells were next selected based on SSC-A versus SSC-H parameters.

### Characterization of CAF-S1 cluster identity upon culture in plastic- or collagen-coated dishes by RNA sequencing

To validate by RNA sequencing the ECM-myCAF and Detox-iCAF identity of fibroblasts isolated from HGSOC patients and maintained either on plastic- or on collagen-coated dishes, as described above, RNAs were extracted using Qiagen miRNeasy Kit (Qiagen, #217004) according to the manufacturer's instructions. RNA integrity and quality were analyzed using the Agilent RNA 6000 nano Kit (Agilent Technologies, #5067-1511). cDNA libraries were prepared using the TruSeq Stranded mRNA Kit (Illumina, #20020594) followed by sequencing on NovaSeq (Illumina). The overall quality of raw sequencing data was first checked using FastQC (v0.11.9). Reads were then aligned on a ribosomal RNA database using bowtie (2.4.2) and on the human reference genome (hg38) with STAR (2.7.6a). Additional controls on aligned data were performed to infer strandness (RSeQC 4.0.0), complexity (Preseq 3.1.1), gene-based saturation, read distribution or duplication level using Bioconductor R package DupRadar. The aligned data were then used to generate a final count matrix with all genes and all samples. Only genes with at least one read in at least 5% of all samples were kept for further analyses. Normalization and differential analysis between plastic and collagen-cultured fibroblasts were conducted with DESeq2 R package. Gene Set Enrichment analysis (GSEA) was then performed using Preranked module based on DESeq2 output and two gene signatures: The NABA Matrisome described in[150] and an inflammatory gene list (compiled from chemokines ligands, chemokine receptors, interferons and Interleukines groups from the Hugo Gene Nomenclature Committee (HNGC), https://www.genenames.org). Heatmap of expression scaled across all samples using GSEA core enriched genes (233 genes from the NABA Matrisome signature, Supplementary Table S3, and 25 genes from the

inflammatory signature, Supplementary Table S4) was build using pheatmap R package.

### Silencing experiment using small-interference RNA (siRNAs)

For the short interfering RNA (siRNA) experiment, $5 \times 10^4$ primary CAF-S1 fibroblasts isolated from HGSOC per condition were used. Plating and transfection of cells were performed the same day. Cells were transfected with 10 nM siRNA in a 24-wells plate. Control was non-targeting siRNA (siCTR, AllStars negative control, #1027281). YAP1 silencing was performed with two distinct siRNAs targeting YAP1 ((*YAP1(1)S* 5′-UGA-GAA-CAA-UGA-CGA-CCA-A-3′ and *YAP1(1)AS* 5′-UUG-GUC-GUC-AUU-GUU-CUC-A-3′ *YAP1(2)S* 5′-CCA-CCA-AGC-UAG-AUA-AAG-A-3′ and *YAP1(2)AS* 5′-CCA-CCA-AGC-UAG-AUA-AAG-A-3′). CYR61 silencing was also achieved using two different siRNA targeting CYR61 (CAA-GAA-CGT-CAT-GAT-GAT-CCA and AGG-GCA-CAC-CTA-GAC-AAA-CAA) (QIAGEN, FlexiTube GeneSolution, #1027416, GS3491 for CYR61). Transfections were performed using DharmaFECT 2 Transfection Reagent (Horizon Discovery, #T-2005-01) according to manufacturer's instructions. Efficient YAP1 and CYR61 silencing were observed after 48 h and maintained along co-culture experiment.

### Ovarian cancer cell line culture

The human ovarian cell line CAOV3 were propagated in DMEM (Gibco #11965092) supplemented with 10% FBS (Biosera, #FB-1003/500), and 1%penicillin (100 U ml⁻¹) and streptomycin (100 µg ml⁻¹) (Gibco #15140122) at 37 °C in 5% of $CO_2$ and 20% of $O_2$. Cell line identity was verified by Short Tandem Repeat (STR) DNA profiling (Promega #B9510). CAOV3 are from American Type Culture Collection, ATCC.

### Isolation of CD8⁺ T lymphocytes

CD8⁺ T cells were isolated from peripheral blood of healthy donors (with informed consent) obtained from "Etablissement Français du Sang", Paris, Saint-Antoine Crozatier blood bank through an approved convention with Institut Curie (Paris, France). Peripheral blood mononuclear cells (PBMC) were isolated using Lymphoprep procedure (Stemcell #07861). After isolation, PBMC could be stored overnight at 4 °C in RPMI (Hyclone #SH30027.01) supplemented with 10% FBS (Biosera #FB-100031/50) and 1% penicillin and streptomycin (Gibco #15140122) at the following concentration $1 \times 10^8$ cells/ml. $10 \times 10^7$ PBMC were then used to isolate CD8⁺ T lymphocytes by using EasyPrep Human CD8⁺ T cell Isolation Kit (Stemcell, #17953) and high gradient magnetic cell separation column "The Big Easy" Easy Sep Magnet (Stemcell, #18001 S) according to the manufacturer's instructions.

### CD8⁺ T lymphocytes - CAF-S1 functional assay

$5 \times 10^4$ CAF-S1 fibroblasts were plated on 24-wells plate (Falcon, #353047) in DMEM (HyClone, #SH30243.01) with 10% FBS (Biosera, #FB-1003/500) at 1.5% $O_2$ during 36 h. CAF-S1 are either transfected with siYAP1(1), siYAP1(2), siCYR61(1), siCYR61(2) or untargeted siRNA (siCtrl). The medium was replaced by fresh DMEM supplemented with 1% FBS just before adding $2.5 \times 10^5$ CD8⁺ T lymphocytes. Co-cultures of CAF-S1 fibroblasts with CD8⁺ T cells were incubated for 24 h at 37 °C in 5% $CO_2$ and 20% $O_2$. After incubation, CD8⁺ T lymphocytes were collected and analyzed by flow cytometry. Cells were first stained for 10 min at RT with a violet live/dead marker (1:1000 BD Bioscience #565388) to exclude dead cells. After washing with PBS+ solution, T lymphocytes were incubated during 15 min at RT with a pool of fluorescent-conjugated primary antibodies recognizing anti-CD45-BUV395 (1:100; BD Biosciences #557833), anti-CD3-AlexaFluor700 (1:50; BD Biosciences #557943), anti-CD8-BV510 (1:50; BD Biosciences, #563919) and anti-PD1-BV421 (1:20; BD Biosciences #562516). For intracellular staining of perforin, granzyme B and IFN-γ, cells were fixed in 4% paraformaldehyde for 15 min at RT and then incubated with an antibody pool containing anti-Perforin-FITC (1:25; BD Biosciences

#563764) and anti-Granzyme B-PE (1:25; BD Biosciences, #561142) and anti-IFN-γ BV786 (1:50; BD Biosciences, #563731) or the corresponding mouse IgG2b-FITC isotype control (1:25; BD Biosciences, #5565383), mouse IgG1-PE isotype control (1:25; BD Biosciences, #555749) and BV786 mouse IgG1 isotype control (1:100; BD Biosciences, #563330). Antibodies are resuspended in PBS+ solution with 0,2% of saponin (Sigma-Aldrich #S7900). Cell analysis was performed on LSRFOR-TESSA (BD Biosciences). At least $1 \times 10^5$ events were recorded. Compensations were performed using single staining on anti-mouse IgG and negative control beads (BD Biosciences #552843) for mouse antibodies, on AbC Total compensation beads (Molecular Probes #A10513) for rat antibody and on ArC reactive beads (Molecular Probes #A10346) for Live/Dead staining (1:1000, Thermo Fisher, #L34955). Data analysis was performed using FlowJo version 10.4.2. Cells were first gated based on forward (FSC-A) and side (SSC-A) scatters. Dead cells were excluded based on their positive staining for live/dead marker. Cells were then gated on CD45+, CD3+, CD8+ cells and next examined for expression of Granzyme B, perforin, IFN-γ and PD-1.

### Transwell migration assay

For migration assay, $5 \times 10^4$ primary ECM-myCAF or Detox-iCAF fibroblasts were plated in the lower chamber of a transwell plate (0.4 µm pore size, Corning HTS Transwell 24 wells #CLS3413) in 500 µl of DMEM supplemented with 10% heat-inactivated FBS and 1% Penicillin streptomycin at 1.5% $O_2$ overnight. After cell adherence, CD8 + T lymphocytes ($2.5 \times 10^5$ cells in a volume of 50 µl DMEM supplemented with 1% FBS) were added in the upper chamber and incubated for 24 h at 37 °C 20% $O_2$. After incubation, CD8 + T lymphocytes in the upper and lower chamber were recovered separately. 0.5 µl of 10 µm carboxylated beads (Polyscience #18133) and DAPI (3 µM) were added to each sample before counting. Cell counting was performed by Flow Cytometry using precision beads for normalization and represented as percentage of migration, calculated as the ratio of the cell number in the lower chamber by the total number of T lymphocytes.

### Protein extraction and western blot

For protein extraction, cells were washed with cold sterile PBS (Gibco #14190) and lyzed with 100 µl of Laemmli buffer (BioRad, #1610737) supplemented with DTT (Thermoscientific, #11896744). The solution was next boiled at 95 °C for 5 min. Samples were next sonicated for 10 min (10 cycles of 30 s ON/30 s OFF) and centrifuged during 10 min at 13,000 x g at 4 °C. The protein extract was then short-term stored at −80 °C. For western blots, a volume of 10 µl of proteins was loaded onto a NuPAGE Novex 4–12% bis tris midi protein gel 26 wells (Invitrogen, #WG1403BOX). The migration was done in 1X NuPAGE® MOPS SDS Running Buffer (for Bis-Tris Gels only) (Invitrogen, #NP0001) in electrophoresis. The proteins were then transferred to a 0.45 µm nitrocellulose membrane (GE Healthcare #10600002) and incubated overnight with primary antibodies diluted in TBS-Tween 0.1% complemented with 5% BSA at 4 °C overnight. Human YAP (D8H1X) Rabbit monoclonal antibody (1:1000, Cell Signaling #14074), Human CYR61 Rabbit polyclonal (1:1000, Novusbio #NB100-356) and Actin (1:10.000; Sigma #A5441).The next day, after 3 washes during 5 mins in TBS-Tween 0,1% at RT, the membrane was incubated with secondary antibody anti-mouse (Jackson ImmunoResearch Laboratories, INC., #115-035-003) or anti-rabbit (Jackson ImmunoResearch Laboratories, INC., #115-035-003) in TBS-Tween 0.1% + 5% BSA or 5% dried milk (Regilait #731142) for 1 hr at RT. Three additional washes with PBS-Tween 0,1% were done before protein detection. The membrane was incubated for 1 min at RT with ECL (ratio 1:1) (Western Lightning Plus-ECL, PerkinElmer, #NEL105001EA). The detection of the signal was done in Chemidoc apparatus for detecting chemiluminescence. The protein bands were analyzed by ImageJ software for protein quantification.

### Statistical analysis

The graphical representation of the data and statistical analyses were done using R environment (https://cran.r-project.org) and GraphPad Prism software. Bar plots or scatter plots are represented with mean ± standard error of the mean (SEM). For violin plots with boxplots, the lower and upper hinges correspond to the first and third quartiles. Median is represented within the boxplot. The upper whisker extends from the hinge to the largest value no further than 1.5* IQR from the hinge (where IQR is the inter-quartile range). The lower whisker extends from the hinge to the smallest value at most 1.5* IQR of the hinge. Data beyond the end of the whiskers are called "outlying" points and are plotted individually. Statistical tests used agree with data distribution. To assess the normality of the distribution of variables, we first applied the Shapiro–Wilk test. According to normality Shapiro-Wilk test, parametric or non-parametric two-tailed tests were applied. The correlation coefficient and its significance between two independent variables were evaluated by Spearman's correlation test. Survival curves were established using the Kaplan-Meier method and compared with the Log-rank test using *survival* R package. Stratification of patients for survival analyses was performed using median value of the stromal content distribution and CAF-S1 or CAF-S4 enrichments (defined by our algorithm, see *#Development of a decision tree algorithm for prediction of CAF population identity*). All applied statistical tests are indicated in the legends. Differences were considered to be statistically significant when p-values were ≤ 0.05. No replicates were generated for primary patient specimens

### Reporting summary

Further information on research design is available in the Nature Portfolio Reporting Summary linked to this article.

## Data availability

Processed data are available on Figshare. The processed data generated in this study have been deposited in the Figshare database under the following links: https://doi.org/10.6084/m9.figshare.22147166 (scRNA seq), https://doi.org/10.6084/m9.figshare.22147103 (spatial transcriptomic) and https://doi.org/10.6084/m9.figshare.25047746 (bulk RNAseq for CAF in culture). Raw sequencing data (Fastq files from scRNAseq, spatial transcriptomic data and bulk RNAseq for CAF in culture) are available from the European Genome-Phenome Archive platform (https://ega-archive.org) under controlled access: EGAS50000000136. The controlled access is required as raw data contain identifying patient information. Data access can be granted via the EGA with completion of an institute data transfer agreement. In addition, bulk RNA sequencing data from the Retrospective Scandare Cohort 2 are available under the controlled EGA number: EGAS50000000145. Count data from scRNAseq dataset from HGSOC before and after chemotherapy was retrieved from Gene Expression Omnibus with accession code GSE165897[99]. Publicly available scRNAseq dataset from breast cancer before and after chemotherapy were retrieved from https://lambrechtslab.sites.vib.be/en/single-cell[100]. Publicly available scRNAseq dataset from treatment-naïve HGSOC patients were recovered from https://lambrechtslab.sites.vib.be/en/high-grade-serous-tubo-ovarian-cancer-refined-single-cell-rna-sequencing-specific-cell-subtypes[45]. Data used to generate the graphs in figures are available as Source Data file and on Figshare under the https://doi.org/10.6084/m9.figshare.24800064. The remaining data are available within the Article, Supplementary Information or Source Data file. Source data are provided with this paper.

## Code availability

Codes used for this study are available on Figshare under the https://doi.org/10.6084/m9.figshare.24271369 and on Zenodo under the https://doi.org/10.5281/zenodo.10555644.

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

## Acknowledgements

We are grateful to Agathe Peltier and Virginie Mieulet for fruitful discussions and critical reading of the manuscript. We also thank Philemon Sirven for helping in collecting samples, Laetitia Fuhrmann, Charlotte Martinat and Benjamin Renouf for assembling clinical data of the patients and Laetitia Lesage and André Nicolas for their help at the experimental pathology platform. We thank the coordination team of the SCANDARE study in the Department of Drug development and Innovation and the clinical research direction of the Institut Curie Hospital. M.L. was supported by the Ligue Nationale Contre le Cancer and the Foundation ARC during her PhD. G.G. and F.M.-G. are permanent scientists at Inserm. R.M. was supported by the Simone and Cino del Duca Foundation, the Foundation ARC (SIGN'it 2019 program) and the Foundation de France (00119142/WB–2021-36276). Y.K. was supported by the Institut National du Cancer INCa (INCa-DGOS-9963; INCa-11692) and SIRIC (INCa-DGOS-4654). The experimental work was supported by grants from the Ligue Nationale Contre le Cancer (*Labelisation*), Inserm, Institut Curie (Incentive and Cooperative Program Tumor Microenvironment PIC TME/T-MEGA, PIC3i CAFi), ICGex (ANR-10-EQPX-03), SIRIC (INCa-DGOS-4654), INCa (STROMAE INCa-DGOS-9963, CaLYS INCa_11692, ChemoCAF, INCa_16086; CAFHeros INCa_16101), the transational ERA-NET TRANSCAN-3 and the ARC Foundation (Chrysalis - ARCPARTN-TRANS2022080005422), the Foundation "Chercher et Trouver" and the association "Christelle Bouillot". F.M.-G. acknowledges both the "French Pink Ribbon Association" and the "Simone and Cino del Duca Foundation" for attribution of their respective Grand Prix. F.M.-G. is very grateful to all her funders for providing support throughout the years.

## Author contributions

F.M.-G. conceived all the project, initiated in part thanks to discussions with A.V.-S. on tumor composition before/after chemotherapy. F.M.-G. and G.G. designed the concept of experiments. M.L. initiated experiments for this project, R.M performed experiments and acquired data, together with G.G., H.R.H., B.B., and I.M. G.G. performed single-cell experiments, and spatial transcriptomics with R.L. C.B., R.R., V.B., M.K., C.L.T., F.L., and A.V.-S. built cohorts of patients. A.V.-S. and L.D. provided human samples and expertise in pathology analyses. S.B. and M.B. provided expertise in new-generation sequencing and L.G. and C.G. for cytometry and multiplex experiments. Y.K., H.C., A.M., and K.M.-E. performed bioinformatic and statistical analyses. F.M.-G. supervised the project and wrote the paper with G.G., M.L., R.M., H.C., and Y.K., with suggestions from all authors. Writing-review & editing: F.M.-G., G.G., R.M., H.C., and Y.K. To note, M.L. and R.M. contributed equally as co-first authors; Y.K., H.C., and C.B. contributed equally as co-second authors.

## Competing interests

F.M.-G. received research support from Innate-Pharma, Roche, Fondation Roche and Bristol-Myers-Squibb (BMS). Other authors declare no potential conflict of interest.

## Additional information

[1]Institut Curie, Stress and Cancer Laboratory, Equipe labélisée par la Ligue Nationale contre le Cancer, PSL Research University, 26, rue d'Ulm, F-75248 Paris, France. [2]Inserm, U830, 26, rue d'Ulm, Paris F-75005, France. [3]Department of Surgery, Institut Curie Hospital Group, 35 rue Dailly, 92210 Saint-Cloud, France. [4]Department of Diagnostic and Theragnostic Medicine, Institut Curie Hospital Group, 26, rue d'Ulm, F-75248 Paris, France. [5]Cytometry platform, PSL University, Institut Curie, 75005 Paris, France. [6]ICGex Next-Generation Sequencing Platform, PSL University, Institut Curie, 75005 Paris, France. [7]Department of Drug Development and Innovation, Institut Curie Hospital Group, 26, rue d'Ulm, F-75248 Paris, France. [8]INSERM, U900, Paris-Saclay University, Institut Curie, 35 rue Dailly, 92210 Saint-Cloud, France. [9]Breast, gynecology and reconstructive surgery Department, Institut Curie Hospital Group, Paris Cité University, 26, rue d'Ulm, F-75248 Paris, France. [10]Department of Diagnostic and Theragnostic Medicine, Institut Curie Hospital Group, 35 rue Dailly, 92210 Saint-Cloud, France. [11]These authors contributed equally: Monika Licaj, Rana Mhaidly. [12]These authors jointly supervised this work: Geraldine Gentric, Fatima Mechta-Grigoriou. ✉e-mail: geraldine.gentric@curie.fr; fatima.mechta-grigoriou@curie.fr

