## [Peer Review File · Nature Communications]

Residual ANTXR1+ myofibroblasts after chemotherapy
inhibit anti-tumor immunity via YAP1 signaling pathwayREVIEWER COMMENTS

Reviewer #1 (Remarks to the Author): with expertise in single-cell transcriptomics, ovarian cancer

Licaj et al. study the effects of chemotherapy in high-grade serous ovarian cancer (HGSOC) on cancer associated fibroblasts (CAFs), and the potential effects of these changes on HGSC patients' prognosis and immune response. The authors inspect chemotherapy-induced changes in CAF subtypes and their effects on the immune response by 1) immunohistochemistry (IHC) in a retrospective cohort of 35 patients with paired specimens, 2) by FACS and single-cell RNA-sequencing (scRNA-seq) in prospective cohorts of 20 or 12 non-paired samples, respectively, 3) by spatial transcriptomics analysis of one patient prior and after chemotherapy, and 4) using published scRNA-seq data, complemented with 5) analysis of CD8+ T cell gene expression in HGSC-derived CAF co-cultures. While it is very important to address the effects of chemotherapy to CAFs and their potential immunomodulatory consequences, and the authors do find interesting associations between changes in the immune cells and CAFs, the manuscript is not entirely convincing in its current form.

Major comments:

1. The the manuscript is difficult to follow, as the authors shift between different CAF subtype levels: first CAF-S1 to -S4, and later either 2 (myCAF/iCAF) or 8 CAF-S1 subtypes, wherein the latter levels are quite mixed from Fig. 3 onwards. More importantly, the markers and definitions used for the subtypes vary: myCAFs are defined sometimes as FAP1-high, sometimes ANTRX1+, and their subtype ECM-myCAFs are sometimes defined by FAP1, sometimes by ANTRX1 and sometimes by transcriptional signature score. Combined, these make reading the manuscript very tedious even for a person who is familiar with CAF heterogeneity, and leave the reader wonder why a particular definition is chosen for a particular analysis. The CAF definitions should be streamlined and used in a consistent and simple manner.
2. The manuscript does not provide sufficient evidence for the claims related to causal role

of ECM-myCAFs in CD8+ T lymphocyte re-infiltration after chemotherapy, such as “decrease of YAP1 in the ECM-myCAF cluster of FAP+ (CAF-S1) fibroblasts after chemotherapy promote CD8+ T lymphocyte re-infiltration”. The authors do show that i) chemotherapy-induced changes in FAP histology score and CD8+ T cell infiltration anti-correlate, and that ii) in a co-culture model, ECM-myCAFs dampen cytotoxic effectors of CD8+ T cells in a YAP1/CYR61 dependent manner but fail to evidence that ECM-myCAFs (however they are defined) have causal relationship with CD8+ T infiltration. Thus, the authors should either remove the claims of causal relationship between ECM-myCAF prevalence and CD8+ T infiltration, or, preferably, provide more direct support.

3. Spatial transcriptomic analysis is performed only in one patient; this is especially problematic in considering that the cancer type studied is highly heterogeneous in multiple levels. The analyses need to be expanded and current findings validated e.g. in publicly available HGSC spatial transcriptomics data sets.

Minor comments:

4. “Standard treatments also include targeted therapies, such as anti-angiogenic drugs and PARP inhibitors that improve progression-free and overall survival in platinum-resistant HGSOC patients 2-10.”

=> Inaccurate, PARPi do not improve PFS/OS in platinum resistant patients. Please re-formulate.

5. “Interestingly, the mesenchymal molecular subtype of HGSOC has been identified in all studies and is systematically associated with poor prognosis. Mesenchymal HGSOC exhibit tumor cells with mesenchymal features and enrichment in specific myofibroblastic CAF populations 25,30,40.”

=> The studies cited do not directly evidence that tumor cells in “mesenchymal HGSOC” would exhibit mesenchymal features. Please re-formulate and refer to single-cell analyses showing that the bulk-RNA based HGSOC subtypes mainly reflect the TME composition of

the tumors rather than features of the cancer cells themselves.

6. Four different CAF populations (referred to as CAF-S1 to -S4) have been recently identified in HGSOC by combining different CAF markers, including Fibroblast Activation Protein (FAP), Smooth Muscle- α Actin (SMA), Integrin β 1 (CD29) and Fibroblast Specific Protein 1 (FSP1)

=> Please also indicate gene names (e.g (SMA, encoded by ACTA2)) to bridge the types more closely to transcriptomic data.

7. "In contrast, the content in both CAF-S1 and CAF-S4 significantly decreased upon chemotherapy, while the proportion of normal-like CAF-S2 and CAF-S3 increased (Fig.1I)."

=> All the CAF markers shown in Fig 1G significantly decrease after chemo. Both of the states that decrease after chemo are defined by high marker expression (Fig 1H; CAF-S1, CAF-4), whereas the states that increase after chemo are based on low/below median marker expression (CAF-S2: CD-29, SMA below Q1; CAF-S3: CD29 between Q1 and median). The authors need to address the possibility that the general decrease in all studied CAF markers could cause the indicated shift in CAF subtypes as a possible confounder for the observed result.

8. "Interestingly, platinum-resistant patients with high residual CAF-S1 or CAF-S4 content after chemotherapy survived less and relapsed earlier than patients enriched in CAF-S2 or CAF-S3 after treatment (P-value = 0.01 for overall survival and P-value = 0.018 for disease-free survival by Kaplan-Meier test) (Supplementary Fig. S1A)."

=> Please test whether this relationship is dependent on the level of residual epithelial content after treatment e.g. by multivariate analysis.

9. Fig 1L; Please show the CAF proportions as separate/partially overlapping barplots / points, the current stacked format with error bars is incomprehensible.

10. “We next tested if there was a link between variations in CAF populations and TIL recruitment upon treatment. CD8+ T cell content tended to be anti-correlated with CAF-S1 fibroblasts before chemotherapy, but this tendency was not observed after chemotherapy (Supplementary Figure S2C, D).” => Too speculative as there is no statistical significance below 0.05; please reformulate.

11. Fig 2E: Are the statistical analyses corrected for multiple testing? If not, please perform the correction and indicate changes that are significant after multiple testing adjustment.

12. “Taken together, these findings suggest that the decrease in the ECM-myCAF cluster after chemotherapy might be instrumental for re-infiltration of CD8+ T lymphocytes after treatment.”

=> This section shows the shift from myCAFs, especially ECM-myCAFs, to iCAFs but does not address T cell infiltration at all. Please remove this sentence.

13. “In agreement, TEAD-target genes were significantly down-regulated in CAF-S1 and in the ECM-myCAF cluster in HGSOC after chemotherapy compared to treatment-naïve samples”

=> Please clearly describe how the TF regulome analysis was performed. Also indicate the source of TEAD target genes and add the list of genes used in further analyses e.g. as a supplementary table.

14. “We next sought to validate this observation by testing if the protein level of YAP1, one of the main TEAD regulator, was decreased in CAF-S1 after chemotherapy in the retrospective cohort of HGSOC patients (Fig. 4E, F).”

=> Please reformulate the sentence; YAP1 and its paralogue TAZ (encoded by WWTR1) are transcriptional activators that bind and promote the activity of the TEAD transcription factors. Please also justify in text why YAP1 knock-down was chosen over TAZ (proto-oncogenic status, expression in stroma, etc).

15. "Taken as a whole, these data suggest that YAP1 expression in ECM-myCAF might be a key negative regulator of CD8+ T cell cytotoxicity."

=> The data presented thus far in the paper indicate no causality, only negative association. Please reformulate the sentence.

16. Please clearly describe e.g. in supplementary tables the combinations of PBMCs and primary CAFs from different patients. For example, Fig 5 displays data from "n = 16 independent experiments, 3 CAF-S1 primary cell lines, 8 PBMC donors"; however, how CAFs from 3 patients are combined with PBMCs from 8 patients to obtain 16 measurements is not explained. Please also address e.g. by multivariate analysis to which extent the PBMC donor and to which extent the CAF donor affect the level of CAF induced suppression of the CD8+ cytotoxic markers.

17. In the Fig 5D to G subpanels it is not obvious that the statistical significance values displayed are from paired wilcoxon/t-tests. Please add e.g. a supplementary figure that shows each paired comparison by connecting the paired observation points by a line. It could also be useful to show the values in ECM-myCAF co-cultures as normalized to the respective PBMC control.

Reviewer #2 (Remarks to the Author): with expertise in cancer associated fibroblasts, ovarian cancer

Licaj et al carry out extensive analysis of HGSOC tissues using various methods (IHC, scRNAseq, spatial transcriptomics) to assess and quantify different CAF subtypes in HGSOC in response to standard-of-care chemotherapy. They show that chemotherapy leads to an overall decrease in epithelial cells and increase in stromal cells, as well as an increase in the number of CD8+ T-cells infiltrating the stroma. While the overall stroma increases, the abundance of certain CAF subtypes (CAF-S1 and CAF-S4) decrease, whereas the abundance of fibroblast types that can also be found in normal tissues (CAF-S2 and CAF-S3) increase. More specifically, CAFs that expressed high levels of FAP and ANTXR1 and can be classified as "ECM-myCAF" are the most significantly decreased upon chemotherapy treatment, and decrease of this population is correlated with increased CD8+ T cell infiltration. It was also

observed that ECM-myCAFs showed decreased expression of TEAD-target genes with chemotherapy treatment, leading the authors to hypothesize that YAP1/TEAD signaling mediates the exclusion of T-cells from tumor stroma. They carry out in vitro assays using isolated ECM-myCAFs co-cultured with isolated CD8+ T-cells to show that ECM-myCAF co-culture decreases expression of cytotoxicity markers and increases expression of PD-1 on CD8+ cells, and that this is reversed with YAP1 knockdown in ECM-myCAFs (and also with knockdown of YAP1/TEAD target gene CYR61). They therefore conclude that decreased YAP1 in the ECM-myCAF cluster of FAP+ (CAF-S1) fibroblasts after chemotherapy promotes CD8+ T cell re-infiltration, suggesting that targeting YAP1 in combination with chemo may favor re-infiltration of CD8+ T cells in patients and improve outcomes.

The manuscript presents a substantial body of work and a very nice analysis of the dynamics of CAF populations present in HGSOC before and after chemotherapy. While this aspect of the paper is very robust, showing consistent changes in CAF populations in response to chemo, the portions of the paper focusing on the functional roles of CAF subtypes (in particular the ECM-myCAF population) with respect to mediating CD8+ T-cell infiltration are not well supported. The vast majority of the data linking CAF populations to CD8+ T-cell infiltration are correlative associations. The data are not sufficient to support the conclusion that a YAP1/TEAD-dependent pathway in ECM-myCAF is a mediator of cytotoxic T cell exclusion. The only functional experiment done, shown Figure 5, shows that co-culture of CD8+ T cells with ECM-myCAFs leads to a reduction in expression of markers of cytotoxicity, which is a different phenomenon and does not address the hypothesis that ECM-myCAFs inhibit T-cell infiltration. In addition, the spatial transcriptomics results indicated that ECM-myCAFs were spatially segregated from CD8+ T-cells, thus the ability of ECM-myCAFs to reduce expression of cytotoxicity markers in CD8+ T cells in a co-culture setting is interesting, it is unclear how relevant this is to the observed changes in T-cell infiltration seen with chemotherapy treatment in vivo. Finally, as mentioned by the authors in the discussion, the role of YAP1 in remodeling ECM production and tumor stiffening and thus creating a barrier to T cell infiltration has been previously described. Overall, the data provided do not support the conclusions that ECM-myCAFs are in addition directly affecting anti-tumor immunity via a YAP1 signaling pathway.

In addition to these major concerns, some other issues/questions relating to individual experiments are as follows:

In Figure 1, epithelial content was compared before and after chemo using morphology/H&E and also EpCAM IHC. However, while the epithelial content was shown to decrease based on H&E, it was not significantly decreased by EPCAM staining. This seems to be a discrepancy and should be explained.

It is clear from data in Figure 1 (e.g. Figure 1G) that the proportion of the various CAF subtypes is quite variable from patient-to-patient. For example, the FAP H-score ranges from 0 to 400. In addition, the patients within the cohort were made up of 21 “sensitive” and 14 “resistant” patients (based on relapse status within 6 months of treatment). It would seem that if CAFs are affecting chemotherapy response, there may be different CAF content between sensitive and resistant patients. Was this assessed? This would also be of interest relative to those few whose markers changed in the opposite direction (e.g. in about a quarter of the samples the FAP and CD29 expression increased substantially rather than decreased after chemo). Were these responses correlated in any way to overall therapy responsiveness?

Supplementary Figure 1 shows survival data only for the 14 patients classified as resistant. Why were these patients selected out for this analysis? Were there any patients in the chemosensitive group that had high residual CAF-S1 or CAF-S4?

In Figure 2G,H, graphs show a negative correlation between the proportion of FAP-high CAF-S1 and CD8/CD3 cells. However, there is no figure or statistical analysis of the flow cytometry data with respect to numbers present and if they were statistically different between treatment naïve and chemo-treated samples.

In Figure 3 and 4 two samples from the same patient (one taken before and one after chemotherapy) are analyzed by spatial transcriptomics. While the results are quite nice and show that spatial transcriptomics can be used to identify and characterize CAF subtypes and their spatial relationship to other cell types (e.g. T cells), a single sample is insufficient to

draw conclusions, especially given the known patient-to-patient heterogeneity with respect to pretreatment CAF population abundances.

In Figure 3, 5 untreated and 7 treated patient samples were analyzed by scRNAseq. While I appreciate the difficulty of obtaining fresh samples for scRNAseq, there is a concern that these are not matched before/after samples, and there are so few in number. There is such variation between patients in terms of the CAF populations present at baseline, making it a concern as to whether differences are truly due to chemo, or simply random sampling error. Additional data sets publicly available (one HGSOc and one breast cancer) were also analyzed but it is not described how many samples were in the treated vs untreated groups and/or whether any of the samples were paired.

In Figure 4 it was observed that there was a decrease in TEAD target genes in CAF-S1 and in ECM-myCAF after chemo, both in scRNAseq data and in samples analyzed by spatial transcriptomics. In Figure 4C (right side) there appears to be virtually no ECM-myCAFs present, which doesn't correspond to the ECM-myCAF shown as being ~15% of CAF-S1 in this sample in Figure 3K. There is a concern that with so few cells of the subtype in question present, whether it's possible to reliably profile the TEAD target genes in this sample. Indeed, the same is true for the scRNAseq data – only percentages are given of each cell type after chemo, but if the ECM-myCAF population is greatly decreased (e.g. in Figure 1F it goes down to a very small proportion of cells), how many cells are actually being analyzed for expression of TEAD target genes?

Figure 4A shows that TEAD transcription factor activity is decreased in the CAF-S1 population after chemotherapy and Figure 4B shows TEAD target genes are down in CAF-S1 and in ECM-myCAF after chemotherapy. It would be useful to see the same analyses of the other CAF populations to confirm that this is indeed unique to the ECM-myCAFs as described. Differential expression of YAP1/TEAD signaling in the ECM-myCAF population has not been shown.

It seems that the YAP1-IHC, including nuclear staining, was scored by eye. Based on Figure 4I, this would be very difficult scoring to do. It would be useful to know: how many

individuals scored the samples? Were the scorers blinded to the chemo treatment status of the samples? How well did individual scores correlate with each other? Ideally a more unbiased scoring approach (e.g. using image analysis software) would be used.

Why assume that the decreased expression of certain genes/pathways within the ECM-myCAF are mediating effects (e.g. T-cell infiltration), rather than just the overall decrease in the number of these cells? Indeed, for most of the measurements shown in Figure 4, one cannot distinguish between these two possibilities (i.e. how can a decrease in expression specifically within the ECM-myCAF populations be distinguished from just an overall decrease in ECM-myCAFs that endogenously express these genes?) This is only shown in Figure 4B (right panel), but as mentioned above, there is concern with respect to not knowing how many cells were actually analyzed in this comparison, especially in the after-chemotherapy group.

No data are shown to demonstrate that TEAD target genes or YAP1 are differentially expressed in ECM-myCAFs. It would be of interest to see the distribution of expression among the CAF subtypes present.

In Figure 5C, only the post-chemo-treated sample is used to make the point that differentiated CD8+ cells are spatially segregated away from ECM-myCAFs. Showing this in a single tissue section is insufficient to demonstrate that this is a general phenomenon. Also, the ECM-myCAF distribution here looks different from that shown in Figure 4C (right panel).

Reviewer #3 (Remarks to the Author): with expertise in cancer associated fibroblasts, chemotherapy, TME

In this work, authors have reported that FAP+ CAF-S1 myofibroblasts decrease following chemotherapy in patients with HGSOc. In particular, they observe a reduction of ANTXR1-expressing CAFs, which express nuclear YAP1. Chemotherapy therefore results in an overall decrease of YAP1-expressing CAFs, which might act on CD8+ T cell infiltration and activation through secretion of CYR61.

This study includes data from immunohistochemistry, flow cytometry, single cell RNA

sequencing and spatial transcriptomic analyses from three HGSOc cohorts. However, throughout the manuscript there are severe quantification and technical inconsistencies, which are particularly concerning for Figures 1 and 2. Addressing such issues is crucial to sustain the authors' messages and conclusions.

Major concerns

1) Relative abundance of stroma and epithelium

Based on the results displayed in figures 1A-D, authors claim a reduction of tumor epithelium after chemotherapy in matched samples. Additionally, flow cytometry analyses in Figure 1K show a decreased percentage of EPCAM+ epithelial cells after chemotherapy (in unmatched samples). Yet, the percentage of EPCAM+ cells showed in Figure 1B, D measured in matched samples before and after therapy remains similar indicating no decrease of tumor epithelium, which contradicts the data in Figure 1C and 1K. How do these results fit together to demonstrate an epithelial reduction?

In this line, if EPCAM+ cells percentages remain constant in matched patient samples before/after chemotherapy, it follows that the percentage of stroma should also remain constant. Instead, authors claim that it is increased based on morphological assessment. On the other hand, the example depicted in supplementary Figure 1B seems to show a large reduction in the amount of CAFs in patients treated with chemotherapy, which contradicts the increase % of total CAFs displayed in Figure 1K. Could authors clarify these discrepant data? Please provide % for each cell subset in supplementary Figure S1B (and in Figure 1J). Of note, cell subsets have to be selected equally in the two analyzed groups for proper comparison of flow cytometry profiles. In the flow cytometry analyses depicted in supplementary Figure 1B and Figure 1K, the gating strategy is altered between "treatment naïve" and "after chemotherapy" samples, particularly in CD45+ vs EPCAM plots. This fact might pose a relevant bias towards the quantification in Figure 1K. How this variation in selecting areas is impacting the results of the comparison between "treatment naïve" and "after chemotherapy" samples?

If this stromal increase is not fully demonstrated, authors cannot claim that the increased infiltration reported in Figure 2 is dependent on an overall increase of the stroma.

In this same direction, samples used in Supplementary Figure 3 shows what it seems to be a similar stromal content before and after chemotherapy. Could authors provide with a percentage of stroma and epithelium from the spatial transcriptomic analysis? How does it fit with all the histological and flow cytometry quantifications depicted in Figure 1?

2) IHC analysis and H-scores

In Figure 1, please provide the rationale behind performing an H-score quantification for EPCAM. Is EPCAM intensity informative? This is not addressed in the manuscript.

Remarkably, there are several patients in Figure 1D center that experience a decrease in EPCAM staining intensity. Does this imply a transformation such as EMT? Is EPCAM staining intensity also affected in flow cytometry analyses?

H-score evaluates an overall staining intensity. Hence, in figure 1G, the H-score is not informative of the actual relative abundance of a certain CAF subpopulation or their localization within the tumor. This could explain some inconsistencies seen between IHC and flow cytometry analyses. For instance, total CAF-S1 does not vary between “treatment naïve” and “after chemotherapy” in Figure 1L, which argues against the decrease reported in the manuscript for Figure 1G. While indeed ECM-myCAFs seem to be drastically reduced in treated samples, many other CAF-S1 subtypes are increasing. This overall increase seems to compensate for the ECM-myCAF decrease, which could explain Figure 1L. Please clarify these points.

Also, it is impossible to ascertain whether T cells are more abundant in, for instance, regions enriched with CAF-S2/S3 from the data presented in Figure 3. Overall, could authors provide with CAF subpopulations abundance and T cell density data? In particular, it would be interesting to match T cell infiltration for every CAF subpopulations. In this sense, how are CAF-S2/S3/S4 represented in the spatial transcriptomic data? Could authors show how the percentage of all 4 main CAF subpopulations is varying after chemotherapy? It would also be important to contrast these CAF subpopulations with the location of T cells.

For IHC analyses, there are powerful open-source softwares that allow accurate quantification of tumor areas and cells with positive DAB staining. Given the aforementioned concerns, authors should reanalyze the abundance of EPCAM+ cells, stromal compartment and T cell infiltration more accurately, using IHC analysis softwares

such as QuPath.

3) T cell stromal and epithelial infiltration

Assuming that the density of T cells remains invariable within the stroma before and after chemotherapy, this will inherently mean that any relative decrease in CAF-S1/S4 and/or increase in CAF-S2/S3 won't affect infiltration. Authors make a poor connection between this effect and the actual proportion of CAF subpopulations, and any correlation depicted in Figure 2 does not fully demonstrate causality. As mentioned in comment 2), it would be interesting to know more about the relative abundance of CAF subpopulations to verify their influence on T cell infiltration. On the other hand, the only relevant effect is an increased infiltration of T cells within the epithelium. This could result directly from the effect of chemotherapy on T cells, or indirectly through the exposure of neoantigens released from dying tumor cells. In this direction, authors should provide with evidence that an increased infiltration is indeed mediated by shifts between CAF subpopulations and not tumor-lymphocyte interactions that are independent from CAFs.

There are concerns on how the authors reach to negative correlations between CAF-S1 and T cells in Figures 2F-H. First, in Supplementary Figures 2C-E, no correlations have been observed between these CAFs and T cells, which suggest that CAF-S1 has no role in modulating T cell infiltration. This makes any conclusion from Figure 2F very difficult, and suggest that FAP and T cell decrease are actually two independent events instead of a cause-effect. Please clarify. Secondly, merging points from "treatment naïve" and "after chemotherapy" samples in the same XY graph is not an accurate approach to find correlations in Figure 2H. Indeed, treatment naïve samples seem to exert the whole statistical effect, while there doesn't seem to be any correlation between %CD8/CD3 and CAF-S1 in samples after chemotherapy. In contrast, authors do segregate samples before and after chemotherapy in different XY plots for Figures 4G and 4H, where they show statistically significant correlations. Therefore, it is important to provide with accurate percentages of CAF-S1 and T cells in both groups separately, and analyze how different they are before and after chemotherapy, and with correlations in both groups separately. Please provide correlation analysis for %CD8/CD3 and CAF subsets in "treatment naïve" samples only. Please provide correlation analysis for %CD8/CD3 and CAF subsets in "after

chemotherapy” samples only. As mentioned above, cell population selection in flow cytometry should be consistent between samples to achieve accurate comparative analysis. Yet again, selection areas differ between “treatment naïve” and “after chemotherapy” samples in Figure 2G. How this variation in selecting areas is impacting the results of the comparison between “treatment naïve” and “after chemotherapy” samples?

4)YAP1 and CYR61

In the 4th results section, authors conclude that the decrease of nuclear YAP1 in the stroma is associated with a reduction in ECM-myCAFs. Yet, the 5th section starts by claiming that YAP1 levels decrease significantly in ECM-myCAFs following chemotherapy. While the latter statement contradicts the former, data depicted in Figure 4 does not show decrease of YAP1 in ECM-myCAFs. Instead, this effect is calculated by H-scores over the total stromal content. Therefore, it is not clear whether YAP1 reduction is following the loss of these CAFs or if it is being also decreased in the remaining CAFs. Authors should provide with immunofluorescence pictures depicting colocalization of YAP1 and ANTXR1 in CAF-, and quantify ANTXR1+YAP1+ and ANTXR1+YAP1- CAFs before and after chemotherapy. Also, what is the significance of YAP1 reduction in tumor epithelial cells seen in Figure 4J? Since the authors demonstrate that such reduction does not affect T cell activation in Figure 5B, could it have any other effect that has not been investigated? From the picture (left, magnification) displayed in Figure 4I, nuclear YAP negative cells counted as epithelial cells could be intraepithelial immune cells instead. Please, perform this analysis taking into account the potential presence of YAP negative intraepithelial immune cells. Finally, concluding that CYR61 has a “key” role is too strong in view of the results from the only one functional experiment (Figure 6G). Either authors should provide with more functional evidence or lighten this statement.

Additional comments

- Data listed in Data availability statement are not available. Please provide updated links allowing reviewer access.
- Authors should delimitate the stroma and epithelium in all IHC pictures from Figure 1. In

particular, it would help to add representative stromal percentages for Figures 1A and 1B.

-Since serial sections for FAP, CD29, SMA and FSP1 stainings are used, authors should provide with images for every marker in same tumor regions as a representation of the decision tree depicted in Figure 1H (for one CAF-S subset in Figure 1F). In supplementary Figures, please provide these representative stainings for the three other CAF-S subsets.

-Authors should provide with representative population percentages in all their flow cytometry analyses.

-In Figure 1F, CD29 and FSP1 seem to be staining both stromal and epithelial cells, and a reduced expression can be observed in both compartments after chemotherapy. What is the significance of CD29 and FSP1 expression in the epithelium?

-Figure 1F is a composite of screenshots (The upper right grey square gives it away). Please provide with higher quality images.

-Authors mention the existence of “platinum resistant patients with high residual CAF-S1 or CAF-S4 content after chemotherapy”, which is also mentioned throughout the discussion. To make this claim, please provide with GSEA data comparing responder patients versus non-responders to test whether CAF-S1/4 are indeed enriched in non-responders.

-According to the methods section, five representative areas (0.105 mm²) per slide were used for T cell quantification. What is the stromal/epithelial content of said areas? How representative are they of the intratumoral variability?

-Please provide with percentages in representative images from Figure 2A. Also, please point the magnified region depicted in the lower-right miniatures.

-In Figure 2G it seems that samples after chemotherapy include much more counted events. For the sake of comparison, authors should limit to a same number of total T cells in their analyses.

-How can authors explain the lack of correlation between T cells or YAP1 and aSMA? In theory, since aSMA stains both CAF-S1 and CAF-S4 populations, the supposed correlation between CAF-S1 and T cells or YAP1 should suffice to see a similar, but probably milder, effect when using aSMA.

-It is interesting to note that a large proportion of samples in Figure 2F tend to focus around a delta H-score of 0 and above, which would imply that many tumors do not only experience no CAF alterations, but FAP can be increased after chemotherapy. Can this effect be connected with overall responses to chemotherapy?

-Since all cells from the scRNAseq analysis from Figure 3F are CAF-S1, what is the relative percentage of this subpopulation within the whole tumor microenvironment, as detected by flow cytometry?

-What does Residual or Relapse mean in Figures 4C and D, respectively? If the sample after chemotherapy is that of a relapsing patient, we could conclude that they are resistant to chemotherapy. Therefore, it follows that CAF-S1 enrichment does not determine resistance to chemotherapy.

-In Figure 5C, while there are clear regions devoid of ECM-myCAFs with increased CD8 intensity, it is not clear whether they are completely excluded from ECM-myCAFs. Could authors focus on particular areas of interest? Also, and since this is the post-chemotherapy sample, would this effect also be observed in the pre-chemotherapy sample? Authors should also include the map for YAP1 expression to reinforce their claim of CD8 T cells being excluded from YAP1+ CAFs.

-FACS gating strategies in Supplementary Figure 4E: Authors should provide with a control of CAF-S4 to demonstrate that FAP and CD29 levels are truly concordant with those of CAF-S1.

Reviewer #4 (Remarks to the Author): with expertise in cancer associated fibroblasts, transcriptomics

In this study the authors present a comprehensive analysis of the role of the stromal compartment in responding to chemotherapy and T cell infiltration. Notably, the authors use patient samples, single cell and spatial methods to infer a subset of CAF-S1, ECM-myCAF coupled with inhibition of YAP1 signaling pathway as a better therapeutic strategy for HGSOc. Particularly, restraining the expression of YAP1 target gene, CYR61 restored the percentage of Gzmb+ CD8+ T lymphocytes. While the study has significant impact and scope, it will benefit from addressing the following comments.

Comments:

1. Figure panels 3F-H with single cell classification of various CAF clusters are not distinct cell types, rather appear as artificial clusters. In order to distinguish which CAF related program is being up regulated, I recommend using topic modeling methods such as cNMF (Kotliar et

al., eLife 2019) to assess changes in gene expression programs pre- and post- treatment and clearly demonstrate the behaviour of the ECM my CAF program.

2. Can the authors comment on the ANTXR1- iCAF sub population of CAF-S1 cluster in relation to chemotherapy? Are consistent levels of these sub populations associated with positive or negative outcomes in HGSOC and other fibrotic solid cancers?

POINT-BY-POINT RESPONSES TO REVIEWERS' COMMENTS

We first would like to thank the four reviewers for their positive evaluation of our work, and for considering our paper suitable for publication in *Nature Communications*, pending modifications. We have now included in the new version of the text the modifications in apparent for addressing their concerns. All these modifications improved our manuscript, and we thank you for all your suggestions. For better visualization of the discussion below, initial comments of the reviewers are indicated in italic and our answers in normal style. We also highlighted below in blue the modifications inserted in the new version of the manuscript and corresponding pages.

Reviewer #1 (Remarks to the Author): with expertise in single-cell transcriptomics, ovarian cancer

Licaj et al. study the effects of chemotherapy in high-grade serous ovarian cancer (HGSC) on cancer associated fibroblasts (CAFs), and the potential effects of these changes on HGSC patients' prognosis and immune response. The authors inspect chemotherapy-induced changes in CAF subtypes and their effects on the immune response by 1) immunohistochemistry (IHC) in a retrospective cohort of 35 patients with paired specimens, 2) by FACS and single-cell RNA-sequencing (scRNA-seq) in prospective cohorts of 20 or 12 non-paired samples, respectively, 3) by spatial transcriptomics analysis of one patient prior and after chemotherapy, and 4) using published scRNA-seq data, complemented with 5) analysis of CD8+ T cell gene expression in HGSC-derived CAF co-cultures. While it is very important to address the effects of chemotherapy to CAFs and their potential immunomodulatory consequences, and the authors do find interesting associations between changes in the immune cells and CAFs, the manuscript is not entirely convincing in its current form.

We thank the Reviewer for the positive evaluation of our work, in particular for considering the various aspects of the experimental analyses we have provided in our paper. We have now considered all concerns underlined by the Reviewer. We thank the Reviewer for his/her recommendations, which have improved our manuscript.

Major comments:

1. The manuscript is difficult to follow, as the authors shift between different CAF subtype levels: first CAF-S1 to -S4, and later either 2 (myCAF/iCAF) or 8 CAF-S1 subtypes, wherein the latter levels are quite mixed from Fig. 3 onwards. More importantly, the markers and definitions used for the subtypes vary: myCAFs are defined sometimes as FAP1-high, sometimes ANTRX1+, and their subtype ECM-myCAFs are sometimes defined by FAP1, sometimes by ANTRX1 and sometimes by transcriptional signature score. Combined, these make reading the manuscript very tedious even for a person who is familiar with CAF heterogeneity, and leave the reader wonder why a particular definition is chosen for a particular analysis. The CAF definitions should be streamlined and used in a consistent and simple manner.

We thank the Reviewer for this mention. As recommended, we have now better defined the different CAF populations all along the new manuscript, in particular in the **introduction p3-4**. We have now indicated in more details the distinct markers characterizing these different populations. Briefly, the CAF-S1 population is defined as FAP+ SMA+ CAF, and the CAF-S4 population as FAP- SMA+ CAF. Among the CAF-S1 (or FAP+ SMA+ CAF) population, we can distinguish ANTXR1+ and ANTXR1- CAF-S1. ANTXR1+ CAF-S1 are positive for the ANTXR1 (ANTXR cell adhesion molecule 1) marker and enriched in myofibroblasts (myCAF), as ANTXR1+ CAF-S1 express high levels of FAP and SMA (FAP^{High} SMA^{High}). In contrast, ANTXR1- CAF-S1, which are negative for the ANTXR1 marker, express low-to-medium levels of FAP and SMA (FAP^{Low-Med} SMA^{Low}) but are still positive for these two markers and are mainly inflammatory (iCAF). Among the ANTXR1+ CAF-S1 (or myCAF), we uncover that the ECM-myCAF cluster is the most abundant one in tumors before treatment and the most affected by chemotherapy, as now shown by using complementary technics (IHC, scRNAseq, spatial transcriptomics). As recommended by the Reviewer, we have now better described these different CAF populations and CAF-S1 (FAP+ SMA+) clusters in the new version of the Text, in particular in the Introduction **p3-4**.

p3-4: Introduction: “Four different CAF populations (referred to as CAF-S1 to -S4) have been recently identified in HGSC by combining different CAF markers, including Fibroblast Activation Protein (FAP), Actin Alpha 2 Smooth Muscle (ACTA2/SMA), Integrin β 1 (ITGB1/CD29) and Fibroblast Specific Protein 1 (FSP1) ⁴⁰. The myofibroblastic CAF-S1 (FAP^{Pos} CD29^{Med} SMA^{Pos} FSP1^{Med-High}) and the pericyte-like CAF-S4 (FAP^{Neg} CD29^{High} SMA^{High} FSP1^{Med}) are strictly detected in tumors, but not in healthy tissues, and are enriched in mesenchymal HGSC ⁴⁰..... Consistent with the pro-metastatic and immunosuppressive functions of the CAF-S1 population, single cell data of CAF-S1 from cancer patients recently highlighted that this population is composed of 8 distinct cellular clusters ⁸². Indeed, among the CAF-S1 (or FAP+ SMA+ CAF) population, we distinguish ANTXR1+ and ANTXR1- CAF-S1. ANTXR1+ CAF-S1 are positive for the ANTXR cell adhesion molecule 1 (ANTXR1) marker and enriched in myofibroblasts (myCAF), as ANTXR1+ CAF-S1 express high levels of FAP and SMA (FAP^{High} SMA^{High}). In contrast, ANTXR1- CAF-S1, which are negative for the ANTXR1 marker, are positive but express

low to medium levels of FAP and SMA (FAP^{Low-Med} SMA^{Low}) and are mainly inflammatory (iCAF). Among the ANTXR1⁺ CAF-S1, the ECM-myCAF cluster is the most abundant one in tumors before treatment and is associated with primary resistance to immunotherapy, while the ANTXR1⁻ CAF-S1 clusters are not⁸².”

2. *The manuscript does not provide sufficient evidence for the claims related to causal role of ECM-myCAFs in CD8⁺ T lymphocyte re-infiltration after chemotherapy, such as “decrease of YAP1 in the ECM-myCAF cluster of FAP⁺ (CAF-S1) fibroblasts after chemotherapy promote CD8⁺ T lymphocyte re-infiltration”. The authors do show that i) chemotherapy-induced changes in FAP histology score and CD8⁺ T cell infiltration anti-correlate, and that ii) in a co-culture model, ECM-myCAFs dampen cytotoxic effectors of CD8⁺ T cells in a YAP1/CYR61 dependent manner but fail to evidence that ECM-myCAFs (however they are defined) have causal relationship with CD8⁺ T infiltration. Thus, the authors should either remove the claims of causal relationship between ECM-myCAF prevalence and CD8⁺ T infiltration, or, preferably, provide more direct support.*

As requested, in the new version of our manuscript, we have now removed the mention of causal relationship between ECM-myCAF and CD8⁺ T cell re-infiltration all along the Text in the new version of the manuscript. We have thus softened our conclusions by eliminating the notion of “cause” and “T cell re-infiltration”.

In addition to these text modifications softening our conclusions, we have now also provided additive experiments confirming the link between ECM-myCAF abundance and reduced CD8⁺ T cell cytotoxicity. We now provide new data by performing functional assays showing that ECM-myCAF dampen cytotoxic effectors of CD8⁺ T lymphocytes. Indeed, we now show that the Detox-iCAF cluster (which belongs to the iCAF clusters of the CAF-S1 population) stimulates the migration of CD8⁺ T lymphocytes. Furthermore, ECM-myCAF directly decreased CD8⁺ T lymphocyte cytotoxicity by increasing PD-1 and reducing Granzyme, Perforin and IFN- γ , while Detox-iCAF are not able to do so. Importantly, we now also demonstrate that CD8⁺ T lymphocytes, which have been co-cultured with ECM-myCAF, show reduced capacity to kill cancer cells. Finally, we show that all ECM-myCAF-mediated effects are significantly affected when YAP1 expression is silenced in ECM-myCAF, thereby demonstrating that these effects are dependent of the YAP1 co-transcription factor. These new functional assays are shown in the **New Fig. 6** and **new Supplementary Fig. S7** and described in the Results' section **p14-p15**.

p14-15: Results: “We next aimed to investigate the role of YAP1 in ECM-myCAF on CD8⁺ T lymphocytes by performing functional assays. We sought to compare the impact of YAP1 silencing in ECM-myCAF on CD8⁺ T cell cytotoxic activity, and used Detox-iCAF as control. To do so, we first isolated Detox-iCAF and ECM-myCAF primary fibroblasts from HGSOc patient samples and evaluated their impact on CD8⁺ T lymphocytes isolated from peripheral blood mononuclear cells (PBMC) of healthy donors. We first verified the identity of these fibroblasts in culture by flow cytometry using specific markers (**Supplementary Figure S6A**) and by bulk RNAseq (**Supplementary Figure S6B-D**) and validated that they corresponded to the ECM-myCAF and Detox-iCAF clusters, respectively. We then analyzed the impact of these two CAF-S1 clusters on CD8⁺ T cell cytotoxicity, considering both markers and activity. We found that, upon co-culture, ECM-myCAF significantly increased the percentage of PD-1⁺ CD8⁺ T lymphocytes, while Detox-iCAF did not (**Fig. 6A** and **Supplementary Figure S7A**). Moreover, this increase in PD-1⁺ CD8⁺ T cells by ECM-myCAF was concomitant to the reduced percentages of granzyme B⁺, perforin⁺ and IFN- γ ⁺ CD8⁺ T lymphocytes (**Fig. 6B-D** and **Supplementary Figure S7A**). Here again, the impact on the percentages of CD8 T cells positive for cytotoxic markers was specific of the ECM-myCAF fibroblasts and not detected with Detox-iCAF (**Fig. 6B-D** and **Supplementary Figure S7A**). Interestingly, we confirmed that ECM-myCAF but not Detox-iCAF reduced CD8⁺ T cell cytotoxic activity (**Fig. 6E**). Indeed, we observed that CD8⁺ T lymphocytes showed reduced capacity to kill cancer cells after co-culture with ECM-myCAF but not with Detox-iCAF (**Fig. 6E**), thereby confirming ECM-myCAF-mediated immunosuppression on CD8⁺ T lymphocyte cytotoxicity. Interestingly, YAP1 silencing (**Supplementary Figure S7B** for siRNA efficacy) in ECM-myCAF reversed ECM-myCAF-mediated effects on cytotoxic CD8⁺ T cells, while it had no impact in Detox-iCAF (**Fig. 6A-E**). Finally, we tested if these CAF-S1 clusters could modulate CD8⁺ T cell migration by performing transwell assay (**Fig. 6F**). Interestingly, Detox-iCAF enhanced CD8⁺ T cell migration, while ECM-myCAF fibroblasts did not (**Fig. 6F**). Altogether, these observations show complementary roles of the two CAF-S1 clusters on CD8 T lymphocytes: while Detox-iCAF attracts CD8⁺ T lymphocytes, ECM-myCAF dampen their cytotoxic identity and functions through a YAP-1 dependent mechanism. “

p44: Corresponding legend of Fig. 6: “**(A) Up**, Representative flow cytometry plots showing CD8 and PD-1 protein levels in control condition (CD8⁺ T cells alone) (-) or co-cultured with Detox-iCAF or ECM-myCAF primary fibroblasts transfected with an untargeted siRNA (siCtrl) or with two different siRNA targeting YAP1 (siYAP1(1), siYAP1(2)). The population of interest (CD8⁺ PD-1⁺) is represented in red and the isotype control in black. **Bottom**, Bar plots showing the % of PD-1⁺ T cells among CD8⁺ T lymphocytes alone or in presence of Detox-iCAF or ECM-myCAF (**Left**) and transfected with siCtrl or siYAP1 (**Middle and Right**). Data are mean \pm SEM (n = 5). P-values from paired Wilcoxon test. **(B-D)** Same as **(A)** for Granzyme B⁺ (**B**), Perforin⁺ (**C**) and IFN- γ ⁺ (**D**) CD8⁺ T lymphocytes. P-values from paired Student t-test. **(E) Left**, Representative flow cytometry

plots showing CAO3 cell death after 24h of incubation with CD8⁺ T lymphocytes pre-incubated with Detox-iCAF or ECM-myCAF primary fibroblasts transfected with an untargeted siRNA (siCtrl) or with two different siRNA targeting YAP1 (siYAP1(1), siYAP1(2)). **Right**, Bar plots showing the % of cancer cell death after incubation with CD8⁺ T lymphocytes. Data are mean \pm SEM (n = 5). P-values from paired Student t-test. **(F)** Bar plots showing the % of migration of CD8⁺ T lymphocytes after 24h of transwell co-culture with Detox-iCAF or ECM-myCAF primary fibroblasts transfected with an untargeted siRNA (siCtrl) or with two different siRNA targeting YAP1 (siYAP1(1), siYAP1(2)). Data are mean \pm SEM (n = 8). P-values from unpaired Student t-test.”

3. *Spatial transcriptomic analysis is performed only in one patient; this is especially problematic in considering that the cancer type studied is highly heterogeneous in multiple levels. The analyses need to be expanded and current findings validated e.g. in publicly available HGSC spatial transcriptomics data sets.*

We thank the Reviewer for this comment. We do agree that spatial transcriptomic data from one patient might not be sufficient. As requested, we have now provided spatial transcriptomics data from a higher number of HGSC patients, and we reached now a total of 10 HGSC patients analyzed with this technology. We would like to emphasize that, at the time of this review, there is no publicly available spatial transcriptomics data after chemotherapy in HGSC patients; data are only available at baseline. We thus generated new data and performed new spatial transcriptomic analysis on additive HGSC naïve of treatment and after chemotherapy for a total of 10 HGSC patients. By adding these new analyses, we validated our initial conclusions and observed after chemotherapy a general increase in the stromal content, a reduced FAP (CAF-S1) and ANTXR1 (ECM-myCAF) expression, a decreased YAP-1/TEAD-dependent signaling pathway and an increased CD8⁺ T cell content. These new spatial transcriptomic data have now been included in the new version of the manuscript in the **New Fig. 3, 4, 5, Sup Fig. 4**, and corresponding description in the Results section **p11-13**, and their corresponding legends **p43-45**.

p11, Results Fig. 3: “Finally, we aimed to define the spatial localization and contribution of CAF-S1 clusters in HGSC at time of diagnosis and in residual disease. To do so, we used the Visium technology and compared spatial transcriptomic data (see Methods #*Spatial Transcriptomics*) from 10 different HGSC samples collected at baseline and after chemotherapy (**Fig. 3N-P**). We first performed pathological annotations to distinguish epithelial and stromal compartments in these HGSC sections (**Supplementary Fig. S4D**). We confirmed that the amount of stroma detected, as well as the number of CD3⁺ and CD8⁺ T lymphocytes, after chemotherapy were higher than in treatment-naïve (**Supplementary Fig. S4E, F**), confirming observations obtained by IHC and flow cytometry and validating that samples selected for spatial transcriptomic as representative). In order to extract the signals from those compartments, we evaluated the abundance of the different CAF-S1 clusters in each spot within the sections by applying the deconvolution method cell2location¹⁰³, using the matrix of reference cell types from the HGSC cellular atlas (**Supplementary Fig. S4A, B**) as input. By this way, we could explore the spatial localization of each CAF-S1 cluster in the 10 HGSC collected and compared the baseline and residual states. We first confirmed within these tissue sections that ANTXR1 expression level and the content in ANTXR1⁺ CAF-S1 were significantly reduced after chemotherapy compared to treatment-naïve section (**Fig. 3O**). In these sections, we identified the different CAF-S1 clusters and confirmed that, among them, the proportion of the ANTXR1⁺ ECM-myCAF cluster was this one, which accumulated before treatment and decreased the most after chemotherapy (**Fig. 3P**). We also observed that the content in the Detox-iCAF cluster increased after treatment (**Fig. 3P**). Taken as a whole, these data highlight the relevance of the decrease in the ANTXR1⁺ ECM-myCAF cluster content after chemotherapy in various cancer types by using several complementary approaches.”

p43: Corresponding legend of Fig. 3: “(N) Representative images of ANTXR1 expression in HGSC Visium sections at time of diagnosis (**Up**, Treatment-naïve) and in residual disease (**Bottom**, After chemotherapy). (O) Violin plot of ANTXR1 expression (**Left**) and barplot of the mean proportion of deconvoluted ANTXR1⁺ CAF-S1 (black) and ANTXR1⁻ CAF-S1 (grey) among CAF-S1 (**Right**) in stromal spots of 10 HGSC Visium sections analyzed at time of diagnosis (N = 4, Treatment naïve) and in residual disease (N = 6, After chemotherapy). n = 37 643 total spots (n = 13 438 Treatment naïve; n = 24 205 after chemotherapy). P-values from Wilcoxon test. (P) % of each deconvoluted CAF-S1 cluster among total CAF-S1 in the 10 HGSC Visium sections analyzed before (N = 4, Treatment-naïve) and after chemotherapy (N = 6). P-values from Wilcoxon test. In panels with error bars, errors bars indicate SEM.”

p12: Results Fig. 4: “Consistent with these observations, spatial transcriptomic data also showed a significant down-regulation of TEAD-target genes after chemotherapy, in particular in areas enriched in ECM-myCAF (**Fig. 4C, D**).”

p43-44: Corresponding legend of Fig. 4: “(C) Representative images showing the expression score of TEAD-target genes (**Left**) and the abundance of deconvoluted ECM-myCAF per spot (**Right**) in HGSC Visium sections analyzed at time of diagnosis (**Up**, Treatment-naïve) and in residual disease (**Bottom**, After chemotherapy). At least one ECM-myCAF per spot is identified by deconvolution in 2 410 spots out of 24 205 spots analyzed after

treatment (average abundance of ECM-myCAF per spot = 2.3; average abundance of all cells per spot = 10). n = 37 643 total spots (13 438 Treatment naïve; 24 205 after chemotherapy). (D) Violin plot showing expression score of TEAD-target genes in stromal spots of spatial transcriptomic data. N = 10 sections (4 Treatment-naïve; 6 after chemotherapy). P-value from Wilcoxon test. Same number of spots as in (C).”

p13: Results Fig. 5: “To get an overview of the spatial organization of CD8⁺ T lymphocytes and ECM-myCAF, we first took advantage of spatial transcriptomic data and cellular HGSOC atlas we built (**Supplementary Fig. S4A, B**). We mapped the localization and the abundance of each cell population by performing deconvolution using the Cell2location algorithm¹⁰³. We applied non-negative matrix factorization (NMF) analysis to identify patterns of co-localization of cell types and states and found that the ECM-myCAF cluster spatially segregated away from CD8⁺ T lymphocytes (**Fig. 5 A-C**). We confirmed this observation by evaluating the distances of CD8⁺ T lymphocytes to the closest ECM-myCAF-enriched or -depleted spots (**Fig. 5D**).”

p44-45: Corresponding legend of Fig. 5: “(A) Representative images showing abundance of deconvoluted ECM-myCAF (**Right**) and differentiated CD8⁺ T lymphocytes (**Left**) per spot in HGSOC Visium sections analyzed at time of diagnosis (**Up**, Treatment-naïve) and in residual disease (**Bottom**, After chemotherapy). The bottom right section shows 2 distinct pathological responses after treatment, one non-responding with a high proportion of residual ECM-myCAF (Left part of the section) and one responding with a reduced ECM-myCAF content (Right part of the section). At least one CD8⁺ T lymphocyte per spot is identified by deconvolution in 5172 spots out of 24 205 spots analyzed after treatment (average abundance of CD8⁺ T cells per spot = 0.67; average abundance of all cells per spot = 11.3). n = 37 643 total spots (n = 13 438 Treatment naïve; n = 24 205 after chemotherapy). (B) Heatmap of the non-negative matrix factorization computed on the deconvolution output using 10 factors on 10 HGSOC sections (4 Treatment-naïve; 6 after chemotherapy). Colors and sizes of circles indicate the scaled cell type abundance. (C) Representative images showing the spatial distribution and score intensity of ECM-myCAF- and CD8⁺ T cell-enriched factors (Factor 6 and Factor 7, respectively) on a HGSOC sample after treatment. (D) Distribution of the closest distances between CD8-enriched spots and ECM-myCAF-enriched or ECM-myCAF-depleted spots (n = 10) in a radius of 1mm.”

Minor comments:

4. “Standard treatments also include targeted therapies, such as anti-angiogenic drugs and PARP inhibitors that improve progression-free and overall survival in platinum-resistant HGSOC patients 2-10.”

=> Inaccurate, PARPi do not improve PFS/OS in platinum resistant patients. Please re-formulate.

As recommended, we have modified the text accordingly (**p3**), and eliminate the mention that the Reviewer accurately highlighted as inappropriate. We thank the Reviewer for the careful reading of our manuscript.

p3: Introduction: “Patients with advanced HGSOC receive a combination of platinum- and taxane-based chemotherapy prior to surgery¹. Standard treatments also include targeted therapies, such as anti-angiogenic drugs and PARP inhibitors^{2,3,4,5,6,7,8,9,10}. However, despite all these treatments, more than 70% of patients still relapse.”

5. “Interestingly, the mesenchymal molecular subtype of HGSOC has been identified in all studies and is systematically associated with poor prognosis. Mesenchymal HGSOC exhibit tumor cells with mesenchymal features and enrichment in specific myofibroblastic CAF populations 25,30,40.”

=> The studies cited do not directly evidence that tumor cells in “mesenchymal HGSOC” would exhibit mesenchymal features. Please re-formulate and refer to single-cell analyses showing that the bulk-RNA based HGSOC subtypes mainly reflect the TME composition of the tumors rather than features of the cancer cells themselves.

We thank the Reviewer for this suggestion, which highlights the importance of stromal heterogeneity in HGSOC. We have now modified the text and added a sentence to include this important point, as recommended, **p3**.

p3: Introduction: “Interestingly, the mesenchymal molecular subtype of HGSOC has been identified in all studies and is systematically associated with poor prognosis. Mesenchymal HGSOC exhibit tumor cells with mesenchymal features and enrichment in specific myofibroblastic CAF populations^{25,30,40,41}. In particular, single cell RNA sequencing analyses (scRNAseq) show that the mesenchymal subtype of HGSOC reflects the abundance of fibroblasts rather than distinct subsets of malignant cells^{42,43,44,45,46}.”

6. Four different CAF populations (referred to as CAF-S1 to -S4) have been recently identified in HGSOC by combining different CAF markers, including Fibroblast Activation Protein (FAP), Smooth Muscle- α Actin (SMA), Integrin b1 (CD29) and Fibroblast Specific Protein 1 (FSP1)

=> Please also indicate gene names (e.g (SMA, encoded by ACTA2)) to bridge the types more closely to transcriptomic data.

We thank the Reviewer for the careful reading of our manuscript. We have now indicated the official name of each CAF marker, when different of the usual name, as requested, and modified the text accordingly, **p3**.

p3: Introduction: “Four different CAF populations (referred to as CAF-S1 to -S4) have been recently identified in HGSOV by combining different CAF markers, including Fibroblast Activation Protein (FAP), Actin Alpha 2 Smooth Muscle (ACTA2/SMA), Integrin β 1 (ITGB1/CD29) and Fibroblast Specific Protein 1 (FSP1) ⁴⁰.”

7. “In contrast, the content in both CAF-S1 and CAF-S4 significantly decreased upon chemotherapy, while the proportion of normal-like CAF-S2 and CAF-S3 increased (Fig. 1I).”

=> All the CAF markers shown in Fig 1G significantly decrease after chemo. Both of the states that decrease after chemo are defined by high marker expression (Fig 1H; CAF-S1, CAF-4), whereas the states that increase after chemo are based on low/below median marker expression (CAF-S2: CD-29, SMA below Q1; CAF-S3: CD29 between Q1 and median). The authors need to address the possibility that the general decrease in all studied CAF markers could cause the indicated shift in CAF subtypes as a possible confounder for the observed result.

We do agree with the Reviewer that these markers characterize activated CAF, and that their global decrease (evaluated by H-scores combining % of stained cells multiplied by intensity of the staining) is consistent with a shift from activated CAF to less activated / normal-like CAF. Indeed, as we have previously published in breast and ovarian cancer (Costa, *Cancer Cell*, 2018; Givel, *Nat. Commun*, 2018), high expression of these markers characterizes activated CAF and their differential protein levels enable us to identify the CAF-S1 and CAF-S4 populations compared to CAF-S2 and CAF-S3. At that time, we established a decision tree algorithm based on the relative protein levels of these different CAF markers, which allows us to differentiate these different populations and is reminded in our manuscript (Fig. 1H, and in the Methods section). Still, we do agree with the Reviewer that the general decrease of these activated CAF markers (such as FAP, SMA, CD29/ β 1-integrin, FSP1) following chemotherapy could be linked to a shift from activated CAF (CAF-S1/CAF-S4) into CAF-S2 or CAF-S3. As recommended by the Reviewer, we have now modified the description of our data to answer to this comment in the Results section, **p6-7**.

p6-7: Results: “We evaluated the histological score (H-score, combining intensity of the staining and percentages of stained cells) for each aforementioned CAF marker in the stroma (Fig. 1F, G). The H-scores of the different CAF markers tested, including FAP, SMA, CD29 and FSP1, significantly decreased in CAF upon chemotherapy (Fig. 1F, G), thereby suggesting that chemotherapy might promote a shift from activated CAF (CAF-S1/CAF-S4) to less activated CAF or normal-like fibroblasts. To verify this hypothesis, we applied a decision tree algorithm (Fig. 1H) (see Methods, #Decision tree algorithm for prediction of CAF population identity and our previous publications for further details ^{40, 80, 82}) to determine the global enrichment in each CAF population per tumor before and after treatment (Fig. 1I).”

8. “Interestingly, platinum-resistant patients with high residual CAF-S1 or CAF-S4 content after chemotherapy survived less and relapsed earlier than patients enriched in CAF-S2 or CAF-S3 after treatment (P-value = 0.01 for overall survival and P-value = 0.018 for disease-free survival by Kaplan-Meier test) (Supplementary Fig. S1A).”

=> Please test whether this relationship is dependent on the level of residual epithelial content after treatment e.g. by multivariate analysis.

As requested, we have now tested if the impact of the content in CAF populations was independent of the residual content in cancer cells after treatment by applying multivariate Cox regression analysis. The hazard ratio for overall survival (OS) and disease-free survival (DFS) was 5.9 (95%CI [1.1-31.3], p=0.038) and 4.9 (95%CI [1.1-21.6], p=0.037), respectively, in patients enriched in CAF-S1 or CAF-S4 compared to those enriched in CAF-S2 or CAF-S3, after considering the level of residual epithelial content following treatment in multivariate analysis. By this way, we validated that the impact observed with CAF populations is independent of the residual epithelial content after treatment. We thank the Reviewer for this suggestion, which improves our manuscript. This multivariate analysis is now included in the **Supplementary Fig. S1B** and described **p7** and in the corresponding legend, **p47**.

p7: Results: “Importantly, multivariate Cox regression analysis showed that this effect was independent on the level of residual epithelial content after treatment (**Supplementary Fig. S1B**), thereby highlighting the interest of variations of these CAF populations after treatment.”

p47: Corresponding legend of Sup Fig. S1B: “(B) Multivariate Cox regression analysis for OS (left) and DFS (right) considering enrichment of CAF populations and residual epithelium content after chemotherapy.”

9. Fig 1L; Please show the CAF proportions as separate/partially overlapping barplots / points, the current stacked format with error bars is incomprehensible.

We are sorry for the lack of clarity. As requested, we have now completed the information given on the variation of the 4 different CAF populations analyzed by FACS after chemotherapy by showing separated plots analyzing the activated CAF-S4 (FAP- SMA+ CAF) and CAF-S1 (FAP+ SMA+ CAF) populations before and after chemotherapy in more details. Indeed, we now provide new quantitative data on the variations of the content in CAF-S4 among CAF and iCAF (FAP^{Low-Med} CAF) and myCAF (FAP^{High} CAF) among the CAF-S1 population, as the CAF-S4 population and the myCAF subset of the CAF-S4 population are the most reduced by chemotherapy. As recommended, we added these new representations in the **new Fig. 1L**.

p7-8: Results: “While the global CAF content increased, we observed variations in CAF-S1 and CAF-S4 content. Indeed, the proportion of the CAF-S4 population was reduced in treated samples (**Fig. 1L**). In addition, flow cytometry analysis enabled us to detect that, among the CAF-S1 population, the content in FAP^{High} CAF-S1 (enriched in myCAF clusters) was the most reduced after chemotherapy (**Fig. 1J, L**). Taken as a whole, these data show that the abundance of both CAF-S4 and FAP^{High} CAF-S1 populations significantly decreases after chemotherapy in HGSOC patients.”

p41: Corresponding legend of Fig. 1L: “(L) Left, % of CAF populations: (FAP^{High} CAF-S1 / myCAF: dark red; FAP^{Low-Med} CAF-S1 / iCAF: light red; CAF-S2: orange; CAF-S3: green; CAF-S4: blue) among total CAF in treatment-naïve HGSOC and after chemotherapy (N = 20). P-value from Chi-square test. Right, Bar plot showing the % of FAP^{High} (myCAF) and FAP^{Low-Med} (iCAF) among CAF-S1 before (treatment-naïve) and after chemotherapy. (N = 20). P-values from Mann-Whitney test. Data are means \pm SEM.”

10. “We next tested if there was a link between variations in CAF populations and TIL recruitment upon treatment. CD8⁺ T cell content tended to be anti-correlated with CAF-S1 fibroblasts before chemotherapy, but this tendency was not observed after chemotherapy (Supplementary Figure S2C, D).”

=> Too speculative as there is no statistical significance below 0.05; please reformulate.

As recommended by the Reviewer, we have now reformulated the sentence in the text, to describe more accurately our data. This change has been made **p8-9** in the new version of the text.

p8-9: Results: ‘We next wondered whether TIL density after chemotherapy could be linked to the extent of CAF-S1 or CAF-S4 decrease. To compare the variations of each cellular population after *versus* before chemotherapy in each patient, we established a delta score (Δ) calculated as followed: content of the studied population after chemotherapy *minus* (-) content of the same population before chemotherapy (see also Methods #Establishment of a delta-score measuring variations of each population by chemotherapy). We analyzed the variations of CD8⁺ TILs (assessed by the Δ -number of CD8⁺ TILs per surface unit) and CAF-S1 (evaluated by the Δ -score of FAP, a CAF-S1 specific marker) in paired HGSOC patients (**Fig. 2E, F** and **Supplementary Fig. S2C**). Interestingly, this analysis showed an anti-correlation between CD8⁺ TILs and CAF-S1, suggesting that the more CAF-S1 decrease after chemotherapy, the more CD8⁺ TIL density increases in the tumor and thus highlighting the importance of the extent of CAF-S1 variation upon treatment. In contrast, we found no association between the proportion of TILs and the overall decrease in myofibroblastic CAF populations, *i.e* when considering both CAF-S1 and CAF-S4 together (evaluated by the Δ -score of the SMA marker, a common marker of these two myofibroblastic populations) (**Supplementary Fig. S2D, E**). This result showed that the increase in CD8⁺ TIL density after treatment is specifically anti-correlated with the CAF-S1 content in HGSOC.”

11. Fig 2E: Are the statistical analyses corrected for multiple testing? If not, please perform the correction and indicate changes that are significant after multiple testing adjustment.

We thank the Reviewer for the careful reading of the manuscript, and we are sorry for the missing information. The correction for multiple testing was not initially applied in the statistical analyses performed in **Fig. 2E** in the first version of the manuscript. As requested by the Reviewer, we have now also performed these analyses by using the Benjamini & Hochberg correction method. When applying this multiple testing, the correlation between Δ H-score SMA and Δ H-score FAP reaches significance ($p = 0.05$), and the anti-correlation between Δ Nb CD8⁺ and Δ H-score FAP tend to give the same results without reaching but approaching significance ($p = 0.08$). We believe that these relationships are of interest and noteworthy, and that multiple test adjustment is not strictly mandatory in this context. Still, as recommended by the Reviewer, we have now performed these statistical analyses corrected for multiple testing and added the p-values after correction for multiple testing in the **new Fig. 2E** and its **corresponding legend, p41-42**.

p41-42: Corresponding legend of Fig. 2E: “(E) Correlation matrix between variations (after / before chemotherapy) of FAP, SMA, FSP1 and CD29 histological scores (H-scores) and the number of CD3⁺, CD8⁺ and FOXP3⁺ TILs per mm² in HGSOC (N = 35). Variations are assessed by a delta score (Δ) calculated as: Variable (H-score or TILs/mm²) after chemotherapy - Variable before chemotherapy. Correlations with p-values < 0.1 (Spearman test after Benjamini & Hochberg correction for multiple testing) are shown. Positive correlations are

in red and negative correlations in blue (N = 35). Square sizes are proportional to P values and color intensities to the correlation coefficients (each P-value is specified in each square).”

12. “Taken together, these findings suggest that the decrease in the ECM-myCAF cluster after chemotherapy might be instrumental for reinfiltration of CD8+ T lymphocytes after treatment.”

=> This section shows the shift from myCAFs, especially ECM-myCAFs, to iCAFs but does not address T cell infiltration at all. Please remove this sentence.

As recommended by the Reviewer, this sentence has now been removed from the text, **p11**.

13. “In agreement, TEAD-target genes were significantly down-regulated in CAF-S1 and in the ECM-myCAF cluster in HGSOc after chemotherapy compared to treatment-naïve samples”

=> Please clearly describe how the TF regulome analysis was performed. Also indicate the source of TEAD target genes and add the list of genes used in further analyses e.g. as a supplementary table.

We are sorry for the missing information in our first version. As recommended, we have now better described the analysis of transcription factors activities in the **Results** and **Methods**’ sections. We also added the list of TEAD target genes considered in the **new Supplementary Table S2**. Inference of transcription factor activity from the gene expression of their target (regulon) was assessed by using VIPER v1.32 and DoRothEA v1.10 R packages. In brief, DoRothEA algorithm constructs a network of TF-gene interactions based on prior knowledge from publicly available databases and literature. To increase the accuracy of the results, only Regulons with a high confidence level (A, B and C) were included in the analysis. We thank the Reviewer for the careful reading of our manuscript. As recommended, all these precisions have now been added in the text, **p12**, and in the Methods section **p26**. We also added the list of TEAD target genes considered in the **new Supplementary Table S2**.

p12: Results: “To this end, we inferred transcription factor activity (TF) from the expression of their gene targets using the DoRothEA algorithm. We first observed that one of the most differential regulon modulated in CAF-S1 after chemotherapy was composed of TEAD-target genes (Fig. 4A, B and Supplementary Table 2).”

p30: Methods sections: “DoRothEA analyses: Inference of transcription factor (TF) activity from the gene expression of their target (regulon) was assessed by using VIPER v1.32 and DoRothEA v1.10 R packages. In brief, the DoRothEA algorithm constructs a network of TF-gene interactions based on prior knowledge from publicly available databases and literature. To increase the accuracy of the results, only Regulons with a high confidence level (A, B and C) were included in the analysis.”

14. “We next sought to validate this observation by testing if the protein level of YAP1, one of the main TEAD regulator, was decreased in CAF-S1 after chemotherapy in the retrospective cohort of HGSOc patients (Fig. 4E, F).”

=> Please reformulate the sentence: YAP1 and its paralogue TAZ (encoded by WWTR1) are transcriptional activators that bind and promote the activity of the TEAD transcription factors. Please also justify in text why YAP1 knock-down was chosen over TAZ (proto-oncogenic status, expression in stroma, etc).

We thank the Reviewer for this mention and advice. We have now reformulated the sentence, as requested, in the new version of the Text, **p 12**. Moreover, as recommended, we have now explained that we have initially dedicated our study on YAP1 because YAP1 is highly expressed in CAF-S1 fibroblasts, in particular in ECM-myCAF. Moreover, by performing multiple analyses and functional assays, we hope the Reviewer will agree that we validated that YAP1 is an important player in ECM-myCAF-mediated effects. Still, to follow the Reviewer’s recommendation, we have also indicated in the Discussion that it might be interesting to analyze TAZ function in a follow-up study.

p12: Results: “YAP1 and its paralogue TAZ (encoded by WWTR1) are transcriptional co-activators that bind and promote the activity of the TEAD transcription factors. Expression of some YAP1/TEAD-target genes was highly detected in CAF-S1 fibroblasts ^{40, 80}, in particular in ECM-myCAF (**Supplementary Fig. S5A**), suggesting that YAP1 could be a key regulator of TEAD transcription factors in CAF-S1 in HGSOc. We thus next sought to validate this hypothesis by testing if the YAP1 protein level.....”

p17: Discussion: “In conclusion, although it might be also interesting to investigate the function of TAZ after chemotherapy, our study suggests that YAP1 might be...”

15. “Taken as a whole, these data suggest that YAP1 expression in ECM-myCAF might be a key negative regulator of CD8+ T cell cytotoxicity.”

=> The data presented thus far in the paper indicate no causality, only negative association. Please reformulate the sentence.

We have now reformulated this sentence and replaced it in the new version of the Text, **p14**, as recommended by the Reviewer.

p14: Results: “Altogether, these data suggest that YAP1 in ANTXR1⁺ CAF, enriched in ECM-myCAF, spatially segregate away from CD8⁺ T lymphocytes in HGSOc.”

16. Please clearly describe e.g. in supplementary tables the combinations of PBMCs and primary CAFs from different patients. For example, Fig 5 displays data from “n = 16 independent experiments, 3 CAF-S1 primary cell lines, 8 PBMC donors”; however, how CAFs from 3 patients are combined with PBMCs from 8 patients to obtain 16 measurements is not explained. Please also address e.g. by multivariate analysis to which extent the PBMC donor and to which extent the CAF donor affect the level of CAF induced suppression of the CD8⁺ cytotoxic markers.

We apologize for the lack of clarity in this part of our manuscript. We have now re-considered all these indications within the text, and modified the text accordingly. For instance, concerning the functional assays, when we indicated in the first version: “n = 5 independent experiments, 3 CAF-S1 primary cell lines, 3 PBMC donors”, this means that 2 CAF-S1 primary cells were analyzed with 2 different PBMC from healthy donors (thus making 4 independent experiments), and a 3rd CAF-S1 primary cell line was analyzed with a 3rd PBMC donor (5th independent experiment). Here, we would like to emphasize that all experiments in this Figure (**new Fig. 6, previous Fig. 5**) have been re-performed and complemented to include the comparison with the Detox-iCAF. To avoid any misunderstanding, we have now indicated the number of independent experiments without additive precision in the **New Fig. 6 legend, p45-46**.

17. In the Fig 5D to G subpanels it is not obvious that the statistical significance values displayed are from paired wilcoxon/t-tests. Please add e.g. a supplementary figure that shows each paired comparison by connecting the paired observation points by a line. It could also be useful to show the values in ECM-myCAF co-cultures as normalized to the respective PBMC control.

We thank the Reviewer for this suggestion. As requested, we have now provided all graphs providing paired statistical tests in the **New Fig. 6** and unpaired tests in the new **Supplementary Fig. S7A**, as requested by the Reviewer.

Reviewer #2 (Remarks to the Author): with expertise in cancer associated fibroblasts, ovarian cancer

Licaj et al carry out extensive analysis of HGSOc tissues using various methods (IHC, scRNAseq, spatial transcriptomics) to assess and quantify different CAF subtypes in HGSOc in response to standard-of-care chemotherapy. They show that chemotherapy leads to an overall decrease in epithelial cells and increase in stromal cells, as well as an increase in the number of CD8⁺ T-cells infiltrating the stroma. While the overall stroma increases, the abundance of certain CAF subtypes (CAF-S1 and CAF-S4) decrease, whereas the abundance of fibroblast types that can also be found in normal tissues (CAF-S2 and CAF-S3) increase. More specifically, CAFs that expressed high levels of FAP and ANTXR1 and can be classified as “ECM-myCAF” are the most significantly decreased upon chemotherapy treatment, and decrease of this population is correlated with increased CD8⁺ T cell infiltration. It was also observed that ECM-myCAFs showed decreased expression of TEAD-target genes with chemotherapy treatment, leading the authors to hypothesize that YAP1/TEAD signaling mediates the exclusion of T-cells from tumor stroma. They carry out in vitro assays using isolated ECM-myCAFs co-cultured with isolated CD8⁺ T-cells to show that ECM-myCAF co-culture decreases expression of cytotoxicity markers and increases expression of PD-1 on CD8⁺ cells, and that this is reversed with YAP1 knockdown in ECM-myCAFs (and also with knockdown of YAP1/TEAD target gene CYR61). They therefore conclude that decreased YAP1 in the ECM-myCAF cluster of FAP⁺ (CAF-S1) fibroblasts after chemotherapy promotes CD8⁺ T cell re-infiltration, suggesting that targeting YAP1 in combination with chemo may favor re-infiltration of CD8⁺ T cells in patients and improve outcomes.

The manuscript presents a substantial body of work and a very nice analysis of the dynamics of CAF populations present in HGSOc before and after chemotherapy. While this aspect of the paper is very robust, showing consistent changes in CAF populations in response to chemo, the portions of the paper focusing on the functional roles of CAF subtypes (in particular the ECM-myCAF population) with respect to mediating CD8⁺ T-cell infiltration are not well supported. The vast majority of the data linking CAF populations to CD8⁺ T-cell infiltration are correlative associations. The data are not sufficient to support the conclusion that a YAP1/TEAD-dependent pathway in ECM-myCAF is a mediator of cytotoxic T cell exclusion. The only functional experiment done, shown Figure 5, shows that co-culture now of CD8⁺ T cells with ECM-myCAFs leads to a reduction in expression of markers of cytotoxicity, which is a different phenomenon and does not address the hypothesis that ECM-myCAFs inhibit T-cell infiltration.

In addition, the spatial transcriptomics results indicated that ECM-myCAFs were spatially segregated from CD8+ T-cells, thus the ability of ECM-myCAFs to reduce expression of cytotoxicity markers in CD8+ T cells in a co-culture setting is interesting, it is unclear how relevant this is to the observed changes in T-cell infiltration seen with chemotherapy treatment in vivo. Finally, as mentioned by the authors in the discussion, the role of YAP1 in remodeling ECM production and tumor stiffening and thus creating a barrier to T cell infiltration has been previously described. Overall, the data provided do not support the conclusions that ECM-myCAFs are in addition directly affecting anti-tumor immunity via a YAP1 signaling pathway.

We first would like to thank the Reviewer for his/her support of our work, and for considering that our study represents a substantial body of work and a very nice analysis of CAF populations in HGSOc before and after chemotherapy.

Regarding the notion of CD8+ T cell infiltration versus CD8+ T cell content, we do agree with the Reviewer. Indeed, although it has been previously shown that CAF and ECM accumulation can be a physical barrier in mouse models, this is not possible to directly prove in patients the impact of ECM-myCAF on CD8+ T cell re-infiltration upon chemotherapy. Thus, we do agree with the Reviewer regarding the notion of CD8+ T cell “re-infiltration” compared to CD8+ T cell content in patients and we modified the text accordingly **all along the new version** of the manuscript.

Still, we sought to go a step further in the role of the different CAF populations on CD8+ T cell content, and have complemented our study at several levels. By using functional assays, we have now shown that CD8+ T lymphocyte migration and CD8+ T cell cytotoxicity are regulated by 2 different FAP+ CAF (CAF-S1) clusters. We now demonstrate that the Detox-iCAF cluster (which belongs to the iCAF clusters of the CAF-S1 population but is ANTXR1-) stimulates the migration of CD8+ T lymphocytes. In contrast, ECM-myCAF directly decrease CD8+ T lymphocyte cytotoxicity by reducing the secretion of Granzyme, Perforin and IFN γ , and by increasing the percentages of PD-1+ CD8+ T lymphocytes, while Detox-iCAF are unable to do so. Importantly, we now also demonstrate that CD8+ T lymphocytes, which have been co-cultured with ECM-myCAF, show reduced capacity to kill cancer cells. Finally, all ECM-myCAF-mediated effects tested by functional assays and listed above are significantly reduced when YAP1 expression is silenced in ECM-myCAF, thereby demonstrating that these effects are dependent of the YAP1 co-transcription factor. All these data are consistent with the content in these cells in HGSOc patients. These new functional assays are shown in the **New Fig. 6** and **new Supplementary Fig. S7** and described in the Results’ section **p14-15**.

p14-15: Results: “We next aimed to investigate the role of YAP1 in ECM-myCAF on CD8+ T lymphocytes by performing functional assays. We sought to compare the impact of YAP1 silencing in ECM-myCAF on CD8+ T cell cytotoxic activity, and used Detox-iCAF as control. To do so, we first isolated Detox-iCAF and ECM-myCAF primary fibroblasts from HGSOc patient samples and evaluated their impact on CD8+ T lymphocytes isolated from peripheral blood mononuclear cells (PBMC) of healthy donors. We first verified the identity of these fibroblasts in culture by flow cytometry using specific markers (**Supplementary Figure S6A**) and by bulk RNAseq (**Supplementary Figure S6B-D**) and validated that they corresponded to the ECM-myCAF and Detox-iCAF clusters, respectively. We then analyzed the impact of these two CAF-S1 clusters on CD8+ T cell cytotoxicity, considering both markers and activity. We found that, upon co-culture, ECM-myCAF significantly increased the percentage of PD-1+ CD8+ T lymphocytes, while Detox-iCAF did not (**Fig. 6A** and **Supplementary Figure S7A**). Moreover, this increase in PD-1+ CD8+ T cells by ECM-myCAF was concomitant to the reduced percentages of granzyme B+, perforin+ and IFN- γ + CD8+ T lymphocytes (**Fig. 6B-D** and **Supplementary Figure S7A**). Here again, the impact on the percentages of CD8 T cells positive for cytotoxic markers was specific of the ECM-myCAF fibroblasts and not detected with Detox-iCAF (**Fig. 6B-D** and **Supplementary Figure S7A**). Interestingly, we confirmed that ECM-myCAF but not Detox-iCAF reduced CD8+ T cell cytotoxic activity (**Fig. 6E**). Indeed, we observed that CD8+ T lymphocytes showed reduced capacity to kill cancer cells after co-culture with ECM-myCAF but not with Detox-iCAF (**Fig. 6E**), thereby confirming ECM-myCAF-mediated immunosuppression on CD8+ T lymphocyte cytotoxicity. Interestingly, YAP1 silencing (**Supplementary Figure S7B** for siRNA efficacy) in ECM-myCAF reversed ECM-myCAF-mediated effects on cytotoxic CD8+ T cells, while it had no impact in Detox-iCAF (**Fig. 6A-E**). Finally, we tested if these CAF-S1 clusters could modulate CD8+ T cell migration by performing transwell assay (**Fig. 6F**). Interestingly, Detox-iCAF enhanced CD8+ T cell migration, while ECM-myCAF fibroblasts did not (**Fig. 6F**). Altogether, these observations show complementary roles of the two CAF-S1 clusters on CD8 T lymphocytes: while Detox-iCAF attracts CD8+ T lymphocytes, ECM-myCAF dampen their cytotoxic identity and functions through a YAP-1 dependent mechanism.”

p44: Corresponding legend of Fig. 6: “(A) Up, Representative flow cytometry plots showing CD8 and PD-1 protein levels in control condition (CD8+ T cells alone) (-) or co-cultured with Detox-iCAF or ECM-myCAF primary fibroblasts transfected with an untargeted siRNA (siCtrl) or with two different siRNA targeting YAP1 (siYAP1(1), siYAP1(2)). The population of interest (CD8+ PD-1+) is represented in red and the isotype control in

black. **Bottom**, Bar plots showing the % of PD-1⁺ T cells among CD8⁺ T lymphocytes alone or in presence of Detox-iCAF or ECM-myCAF (**Left**) and transfected with siCtrl or siYAP1 (**Middle and Right**). Data are mean \pm SEM (n = 5). P-values from paired Wilcoxon test. (**B-D**) Same as (**A**) for Granzyme B⁺ (**B**), Perforin⁺ (**C**) and IFN- γ ⁺ (**D**) CD8⁺ T lymphocytes. P-values from paired Student t-test. (**E**) **Left**, Representative flow cytometry plots showing CAOV3 cell death after 24h of incubation with CD8⁺ T lymphocytes pre-incubated with Detox-iCAF or ECM-myCAF primary fibroblasts transfected with an untargeted siRNA (siCtrl) or with two different siRNA targeting YAP1 (siYAP1(1), siYAP1(2)). **Right**, Bar plots showing the % of cancer cell death after incubation with CD8⁺ T lymphocytes. Data are mean \pm SEM (n = 5). P-values from paired Student t-test. (**F**) Bar plots showing the % of migration of CD8⁺ T lymphocytes after 24h of transwell co-culture with Detox-iCAF or ECM-myCAF primary fibroblasts transfected with an untargeted siRNA (siCtrl) or with two different siRNA targeting YAP1 (siYAP1(1), siYAP1(2)). Data are mean \pm SEM (n = 8). P-values from unpaired Student t-test.”

In addition to these major concerns, some other issues/questions relating to individual experiments are as follows: In Figure 1, epithelial content was compared before and after chemo using morphology/H&E and also EpCAM IHC. However, while the epithelial content was shown to decrease based on H&E, it was not significantly decreased by EPCAM staining. This seems to be a discrepancy and should be explained.

We are sorry for the lack of clarity in the first version of our manuscript. Indeed, as expected after treatment, we show that the global epithelial content -assessed by pathologists based on morphological criteria- decreases after chemotherapy in most HGSOC patients. We also wondered whether this global epithelium decrease could be associated with a lack of epithelial features. To test this hypothesis, we performed immunohistochemistry staining of EPCAM, a well-known epithelial marker. Most epithelial cancer cells were EPCAM⁺ before treatment and the intensity of the staining remained high after chemotherapy as shown in **Fig. 1D**, indicating that residual epithelial cells after chemotherapy expressed EPCAM at the same levels as before treatment. Thus, after chemotherapy, we observed a global decrease in the epithelial content, but residual cancer cells maintain their epithelial features, evaluated by EPCAM staining. We do agree that the description of these data in the first version of the text was misleading, and we modified the text accordingly, **p6**.

p6: Results: “As expected, most HGSOC exhibited a lower epithelial content after chemotherapy compared to the corresponding samples at time of diagnosis (**Fig. 1A, C**). We next wondered whether this global decrease in the epithelium content could be associated with a lack of epithelial features by performing immunohistochemistry (IHC) staining of EPCAM, a well-known epithelial marker (**Fig. 1B, D**). Although a small number of patients exhibited a decrease in EPCAM staining, epithelial cancer cells were EPCAM⁺ before treatment and the intensity of the staining remained high after chemotherapy in most patients (**Fig. 1D**), indicating that residual epithelial cells after chemotherapy expressed EPCAM at the same levels as before treatment.”

p40: Corresponding legend of Fig. 1: “(**C**) Percentage (%) of epithelium (determined by morphological analysis on HES staining) in HGSOC before and after chemotherapy (N = 35 patients, n = 70 matched samples). Data are shown using paired (Left, P-value from paired Wilcoxon test) and unpaired (Right, P-value from Mann-Whitney test) statistical analyses. (**D**) Intensity of EPCAM staining in epithelial cells, before and after chemotherapy, ranging from 0 to 4 (N = 35). Paired (Left, P-value from paired Wilcoxon test) and unpaired (Right, P-value from Mann-Whitney test) analyses.”

It is clear from data in Figure 1 (e.g. Figure 1G) that the proportion of the various CAF subtypes is quite variable from patient-to-patient. For example, the FAP H-score ranges from 0 to 400. In addition, the patients within the cohort were made up of 21 “sensitive” and 14 “resistant” patients (based on relapse status within 6 months of treatment). It would seem that if CAFs are affecting chemotherapy response, there may be different CAF content between sensitive and resistant patients. Was this assessed? This would also be of interest relative to those few whose markers changed in the opposite direction (e.g. in about a quarter of the samples the FAP and CD29 expression increased substantially rather than decreased after chemo). Were these responses correlated in any way to overall therapy responsiveness?

We thank the reviewer for this comment. We do agree with this comment and found interesting that the proportions of CAF populations before/after treatment are different from one patient to another, and we do agree that investigating the predictive value of CAF composition is a very interesting question. To address this question, we have analyzed the data from the 35 patients of the Retrospective Curie 1 cohort before chemotherapy. However, we didn’t observe any significant association between response to chemotherapy and CAF population content at time of diagnosis. We do agree with the Reviewer that this lack of predictive value at diagnosis is an important information, and to be totally transparent, we have now added this information in the new version of text **p7**.

p7: Results: “Although CAF populations at diagnosis did not predict survival, residual CAF populations after chemotherapy were significantly associated with response to chemotherapy.”

In addition, as we observed the importance of maintenance of CAF-S1/CAF-S4 populations after chemotherapy in platinum resistant patients (**Supplementary Fig. 1A**), we would like to emphasize that we have now also tested if the impact of the content in CAF populations was independent of the residual content in cancer cells after treatment by applying multivariate Cox regression analysis. The hazard ratio for overall survival (OS) and disease-free survival (DFS) was 5.9 (95%CI [1.1-31.3], p=0.038) and 4.9 (95%CI [1.1-21.6], p=0.037), respectively, in patients enriched in CAF-S1 or CAF-S4 compared to those enriched in CAF-S2 or CAF-S3, after considering the level of residual epithelial content following treatment in multivariate analysis. By this way, we validated that the impact observed with CAF populations is independent of the residual epithelial content after treatment. This multivariate analysis is now included in the **Supplementary Fig. S1B** and described **p7** and in the corresponding legend, **p47**.

p7: Results: “Indeed, platinum-resistant patients with high residual CAF-S1 or CAF-S4 content after chemotherapy survived less and relapsed earlier than patients enriched in CAF-S2 or CAF-S3 after treatment (P-value = 0.01 for overall survival and P-value = 0.018 for disease-free survival by Kaplan-Meier test) (**Supplementary Fig. S1A**). Importantly, multivariate Cox regression analysis showed that this effect was independent on the level of residual epithelial content after treatment (**Supplementary Fig. S1B**), thereby highlighting the interest of variations of these CAF populations after treatment.”

p47: Corresponding legend of Sup Fig. S1B: “(B) Multivariate Cox regression analysis for OS (left) and DFS (right) considering enrichment of CAF populations and residual epithelium content after chemotherapy.”

Furthermore, as we observed a switch in the composition of specific CAF-S1 clusters in HGSOC after chemotherapy (with a significant reduction of the ECM-myCAF cluster and an increase in the Detox-iCAF cluster, **Fig. 3**), we sought to test if this switch could be associated with response to treatment. To do so, we now analyzed an independent cohort of HGSOC patients for whom we had access to a long-term clinical follow up and we performed bulk RNA sequencing. By using a deconvolution method to assess the composition of CAF-S1 clusters in each patient before and after treatment, we validated the decrease in ECM-myCAF content and the increase of Detox-iCAF after treatment in this cohort. Interestingly, we found a significant association between the content in these clusters after chemotherapy and the chemotherapy response. Indeed, we observed a lower proportion of ECM-myCAF and a higher proportion of Detox-iCAF after chemotherapy in responder patients compared to non-responder patients (p = 0.024 and p = 0.015, respectively). We have now integrated these results in **New Fig. 3K, L** and in the new version of the manuscript, **p10-11**.

p10-11, Results: “We then sought to validate these findings by studying bulk RNA-seq data from an independent cohort of ovarian cancer patients (containing 45 samples before treatment and 25 after chemotherapy, see **Table 1**, retrospective SCANDARE Curie 2 cohort). For this cohort of patients, long-term clinical follow-up and information on treatment responses of patients were available. Cellular composition of each sample -before and after chemotherapy- was inferred by using BayesPrism method¹⁰² based on a high-resolution HGSOC cellular atlas that we built and annotated from both Curie and publicly available scRNAseq datasets (**Supplementary Fig. S4A, B**). This HGSOC atlas was composed of 49 909 cells and 24 different cell types and states, including CAF-S1 clusters, thereby constituting a comprehensive HGSOC cellular landscape (**Supplementary Fig. S4A, B**). In this independent cohort, we confirmed that the proportion of CD8⁺ T lymphocytes increased after treatment (**Fig. 3J**). Importantly, we also validated the decrease in ECM-myCAF content and increase in Detox-iCAF after chemotherapy (**Fig. 3K**). Moreover, thanks to the long-term clinical follow-up available in the SCANDARE cohort, we were able to highlight that the decrease in the ECM-myCAF content and the increase in the Detox-iCAF proportion following chemotherapy were mainly detected in responder patients but not in non-responders (**Fig. 3L** and **Supplementary Fig. S4C**). Moreover, the increase of CD8 T cell density was also detected in responder patients (**Fig. 3M**), showing that variations of these CAF-S1 clusters are concomitant to increased CD8⁺ T cell content after chemotherapy. Taken as a whole, these data confirmed the clinical interest of the variations of these CAF-S1 clusters upon treatment.”

p43, Corresponding Legend Fig. 3J-M: “(J) Proportion of CD8⁺ T lymphocytes in the retrospective SCANDARE Curie 2 cohort before (treatment-naïve) and after chemotherapy (N = 70 samples: 45 treatment-naïve, 25 after chemotherapy). P-value from Mann-Whitney test. (K) Same as (J) for the proportion of ECM-myCAF (**Left**) and Detox-iCAF (**Right**) clusters among CAF-S1. (L, M) Same as (J, K) according to response to chemotherapy for ECM-myCAF (**L, Left**), Detox-iCAF (**L, Right**) and CD8⁺ T cells (**M**). P-values from Mann-Whitney test.”

Supplementary Figure 1 shows survival data only for the 14 patients classified as resistant. Why were these patients selected out for this analysis? Were there any patients in the chemosensitive group that had high residual CAF-S1 or CAF-S4?

We thank the reviewer for raising this point. We focused on chemotherapy-resistant patients because they are the most relevant population from a clinical perspective. Indeed, our objective was to identify whether the composition

of CAF could be associated with early relapse. This is indicated **p7**. In addition, as mentioned above, we have now validated that the impact observed with CAF populations is independent of the residual epithelial content after treatment. This multivariate analysis is now included in the **Supplementary Figure 1B**, and described **p7**.

p7: Results: “Indeed, platinum-resistant patients with high residual CAF-S1 or CAF-S4 content after chemotherapy survived less and relapsed earlier than patients enriched in CAF-S2 or CAF-S3 after treatment (P-value = 0.01 for overall survival and P-value = 0.018 for disease-free survival by Kaplan-Meier test) (**Supplementary Fig. S1A**). Importantly, multivariate Cox regression analysis showed that this effect was independent on the level of residual epithelial content after treatment (**Supplementary Fig. S1B**), thereby highlighting the interest of variations of these CAF populations after treatment.”

p47: Corresponding legend of Sup Fig. S1B: “**(B)** Multivariate Cox regression analysis for OS (left) and DFS (right) considering enrichment of CAF populations and residual epithelium content after chemotherapy.”

In Figure 2G,H, graphs show a negative correlation between the proportion of FAP-high CAF-S1 and CD8/CD3 cells. However, there is no figure or statistical analysis of the flow cytometry data with respect to numbers present and if they were statistically different between treatment naïve and chemo-treated samples.

As requested, we have now included a plot showing the percentage of CD8⁺ T lymphocytes among total CD3⁺ T cells detected by flow cytometry in HGSOC patients before and after chemotherapy (**Fig. 2H**). When we consider all patients (without distinguishing patients enriched in FAP^{High} CAF-S1 from those enriched in FAP^{Low-Med} CAF-S1), we did not detect any significant difference between treatment-naïve and chemo-treated samples. This is only when we focused our analysis on FAP^{High} CAF-S1 that we could detect the anti-correlation between the % of CAF-S1 and the number of CD8⁺ T lymphocytes among CD3⁺ T cells. In contrast, there is no significant correlation with FAP^{Low-Med} CAF-S1 (**Fig. 2J**) and CAF-S4 (**Supplementary Fig. S2F**). These data are now shown in **Fig. 2H-J** and **Supplementary Fig. S2F** and described **p9**.

p9: Results: “Finally, we sought to confirm this observation by flow cytometry analysis of untreated and treated HGSOC (Prospective cohort 1, **Table 1**) (**Fig. 2 G-J**), although these unpaired samples did not allow us to compare cellular variations before / after chemotherapy per patient. When we considered all patients (without distinguishing patients enriched in FAP^{High} CAF-S1 from those enriched in FAP^{Low-Med} CAF-S1), we did not detect any significant difference between treatment-naïve and chemo-treated samples (**Fig. 2H**). However, when we focused our analysis on FAP^{High} CAF-S1 thanks to the sensitivity of the flow cytometry method, we detected an anti-correlation between the percentage of CAF-S1 among total CAF and the number of CD8⁺ T lymphocytes among CD3⁺ T cells (**Fig. 2I**). But, there was no link in any way with FAP^{Low-Med} CAF-S1 (**Fig. 2J**) and CAF-S4 (**Supplementary Fig. S2F**).”

p42, Corresponding legend of Fig. 2: “**(H)** % of CD3⁺ T lymphocytes among CD45⁺ cells (**Left**) and % CD8⁺ T lymphocytes among CD3⁺ T cells (**Right**) assessed by flow cytometry from HGSOC samples collected before and after chemotherapy (Prospective cohort 1). N = 16 patients. P-values from unpaired Student t-test. **(I, J)** Correlations plots with linear regression lines between % of CD8⁺ among CD3⁺ TILs and % of FAP^{High} CAF-S1 **(I)** and FAP^{Low-Med} CAF-S1 **(J)** among CAF, before chemotherapy (treatment-naïve, **Left**), after chemotherapy (**Middle**) or before and after chemotherapy (Circles: treatment-naïve; Triangles: after chemotherapy) (**Right**). Data are from flow cytometry (Prospective cohort 1). N = 16 patients. P values from Spearman correlation test.”

As mentioned above, we also analyzed an independent cohort of HGSOC patients for whom we had access to a long-term clinical follow up and we performed bulk RNA sequencing. In this independent cohort, we confirmed that the proportion of CD8⁺ T lymphocytes increased after treatment (**Fig. 3J**), together with the decrease in ECM-myCAF content and the increase in Detox-iCAF after chemotherapy (**Fig. 3K**). We have now integrated these results in **New Fig. 3K, L** and described them, **p11**.

p11, Results: “We then sought to validate these findings by studying bulk RNA-seq data from an independent cohort of ovarian cancer patients.....In this independent cohort, we confirmed that the proportion of CD8⁺ T lymphocytes increased after treatment (**Fig. 3J**). Importantly, we also validated the decrease in ECM-myCAF content and increase in Detox-iCAF after chemotherapy (**Fig. 3K**).”

p43, Corresponding Legend Fig. 3J-M: “**(J)** Proportion of CD8⁺ T lymphocytes in the retrospective SCANDARE Curie 2 cohort before (treatment-naïve) and after chemotherapy (N = 70 samples: 45 treatment-naïve, 25 after chemotherapy). P-value from Mann-Whitney test. **(K)** Same as **(J)** for the proportion of ECM-myCAF (**Left**) and Detox-iCAF (**Right**) clusters among CAF-S1. **(L, M)** Same as **(J, K)** according to response to chemotherapy for ECM-myCAF (**L, Left**), Detox-iCAF (**L, Right**) and CD8⁺ T cells (**M**). P-values from Mann-Whitney test.”

Finally, spatial transcriptomic analyses from 10 HGSOC patients show the same results and confirmed these observations. These data are now inserted in the **New Fig. 3N-P** and **New Supplementary Fig. S4**, and described **p11-12**.

p11, Results Fig. 3: “Finally, we aimed to define the spatial localization and contribution of CAF-S1 clusters in HGSOC at time of diagnosis and in residual disease. To do so, we used the Visium technology and compared spatial transcriptomic data (see Methods #*Spatial Transcriptomics*) from 10 different HGSOC samples collected at baseline and after chemotherapy (Fig. 3N-P). We first performed pathological annotations to distinguish epithelial and stromal compartments in these HGSOC sections (Supplementary Fig. S4D). We confirmed that the amount of stroma detected, as well as the number of CD3⁺ and CD8⁺ T lymphocytes, after chemotherapy were higher than in treatment-naïve (Supplementary Fig. S4E, F), confirming observations obtained by IHC and flow cytometry and validating that samples selected for spatial transcriptomic as representative). In order to extract the signals from those compartments, we evaluated the abundance of the different CAF-S1 clusters in each spot within the sections by applying the deconvolution method cell2location¹⁰³, using the matrix of reference cell types from the HGSOC cellular atlas (Supplementary Fig. S4A, B) as input. By this way, we could explore the spatial localization of each CAF-S1 cluster in the 10 HGSOC collected and compared the baseline and residual states. We first confirmed within these tissue sections that ANTXR1 expression level and the content in ANTXR1⁺ CAF-S1 were significantly reduced after chemotherapy compared to treatment-naïve section (Fig. 3O). In these sections, we identified the different CAF-S1 clusters and confirmed that, among them, the proportion of the ANTXR1⁺ ECM-myCAF cluster was this one, which accumulated before treatment and decreased the most after chemotherapy (Fig. 3P). We also observed that the content in the Detox-iCAF cluster increased after treatment (Fig. 3P). Taken as a whole, these data highlight the relevance of the decrease in the ANTXR1⁺ ECM-myCAF cluster content after chemotherapy in various cancer types by using several complementary approaches.”

p43: Corresponding legend of Fig. 3: “(N) Representative images of ANTXR1 expression in HGSOC Visium sections at time of diagnosis (Up, Treatment-naïve) and in residual disease (Bottom, After chemotherapy). (O) Violin plot of ANTXR1 expression (Left) and barplot of the mean proportion of deconvoluted ANTXR1⁺ CAF-S1 (black) and ANTXR1⁻ CAF-S1 (grey) among CAF-S1 (Right) in stromal spots of 10 HGSOC Visium sections analyzed at time of diagnosis (N = 4, Treatment naïve) and in residual disease (N = 6, After chemotherapy). n = 37 643 total spots (n = 13 438 Treatment naïve; n = 24 205 after chemotherapy). P-values from Wilcoxon test. (P) % of each deconvoluted CAF-S1 cluster among total CAF-S1 in the 10 HGSOC Visium sections analyzed before (N = 4, Treatment-naïve) and after chemotherapy (N = 6). P-values from Wilcoxon test. In panels with error bars, errors bars indicate SEM.”

p48: Corresponding legend of Sup Fig. S4: “Supplementary Figure S4: HGSOC cellular atlas for deconvolution of bulk RNAseq and spatial data: (A) UMAP of 49 909 cells from HGSOC, which encompass 24 different cell types and states and compose a comprehensive HGSOC cellular atlas. (B) Dotplot of expression values of specific genes in the 24 cell types and states of the HGSOC atlas. Colors highlight average expression of each gene across all cells within a cell type. Circle sizes represent the % of cells expressing a given gene.(D) Top, H&E staining of the 10 spatial transcriptomic sections analyzed at baseline (Left, Treatment-naïve) and in residual disease (Right, After chemotherapy). Bottom, Pathological annotations of the sections distinguishing spots covering the epithelial compartment (red) and the stromal compartment (blue). (E) % of stroma (determined by morphological analysis on HES staining) in HGSOC sections analyzed by spatial transcriptomics. P-value from Welch t-test. (F) Number of CD3⁺ (Left) and CD8⁺ (Right) T lymphocytes identified per spot after deconvolution in the spatial transcriptomic data before and after treatment. At least one CD3⁺ T lymphocyte per spot is identified by deconvolution in 16 194 spots out of 37 643 spots (average abundance of CD3⁺ or CD8⁺ per spot = 2.16 (CD3⁺) and 0.52 (CD8⁺); average abundance of all cells per spot = 10.2). n = 37 643 total spots (13 438 Treatment-naïve; 24 205 after chemotherapy. P-values from Mann-Whitney test.”

In Figure 3 and 4 two samples from the same patient (one taken before and one after chemotherapy) are analyzed by spatial transcriptomics. While the results are quite nice and show that spatial transcriptomics can be used to identify and characterize CAF subtypes and their spatial relationship to other cell types (e.g. T cells), a single sample is insufficient to draw conclusions, especially given the known patient-to-patient heterogeneity with respect to pretreatment CAF population abundances.

We thank the Reviewer for this comment. We do agree that spatial transcriptomic data from one patient might not be sufficient. As requested, we have now provided spatial transcriptomics data from a higher number of HGSOC patients, and we reached now a total of 10 HGSOC patients analyzed with this technology. We would like to emphasize that, at the time of this review, there is no publicly available spatial transcriptomics data after chemotherapy in HGSOC patients; data are only available at baseline. We thus generated new data and performed new spatial transcriptomic analysis on additive HGSOC naïve of treatment and after chemotherapy for a total of 10 HGSOC patients. By adding these new analyses, we validated our initial conclusions and observed after chemotherapy a general increase in the stromal content, a reduced FAP (CAF-S1) and ANTXR1 (ECM-myCAF) expression, a decreased YAP-1/TEAD-dependent signaling pathway and an increased CD8⁺ T cell content. These new spatial transcriptomic data have now been included in the new version of the manuscript in the **New Fig. 3**,

4, 5, Sup Fig. 4, and corresponding description in the Results section p11-13, and their corresponding legends p43-45.

p11, Results Fig. 3: “Finally, we aimed to define the spatial localization and contribution of CAF-S1 clusters in HGSOC at time of diagnosis and in residual disease. To do so, we used the Visium technology and compared spatial transcriptomic data (see Methods #Spatial Transcriptomics) from 10 different HGSOC samples collected at baseline and after chemotherapy (Fig. 3N-P). We first performed pathological annotations to distinguish epithelial and stromal compartments in these HGSOC sections (Supplementary Fig. S4D). We confirmed that the amount of stroma detected, as well as the number of CD3⁺ and CD8⁺ T lymphocytes, after chemotherapy were higher than in treatment-naïve (Supplementary Fig. S4E, F), confirming observations obtained by IHC and flow cytometry and validating that samples selected for spatial transcriptomic as representative). In order to extract the signals from those compartments, we evaluated the abundance of the different CAF-S1 clusters in each spot within the sections by applying the deconvolution method cell2location¹⁰³, using the matrix of reference cell types from the HGSOC cellular atlas (Supplementary Fig. S4A, B) as input. By this way, we could explore the spatial localization of each CAF-S1 cluster in the 10 HGSOC collected and compared the baseline and residual states. We first confirmed within these tissue sections that ANTXR1 expression level and the content in ANTXR1⁺ CAF-S1 were significantly reduced after chemotherapy compared to treatment-naïve section (Fig. 3O). In these sections, we identified the different CAF-S1 clusters and confirmed that, among them, the proportion of the ANTXR1⁺ ECM-myCAF cluster was this one, which accumulated before treatment and decreased the most after chemotherapy (Fig. 3P). We also observed that the content in the Detox-iCAF cluster increased after treatment (Fig. 3P). Taken as a whole, these data highlight the relevance of the decrease in the ANTXR1⁺ ECM-myCAF cluster content after chemotherapy in various cancer types by using several complementary approaches.”

p43: Corresponding legend of Fig. 3: “(N) Representative images of ANTXR1 expression in HGSOC Visium sections at time of diagnosis (Up, Treatment-naïve) and in residual disease (Bottom, After chemotherapy). (O) Violin plot of ANTXR1 expression (Left) and barplot of the mean proportion of deconvoluted ANTXR1⁺ CAF-S1 (black) and ANTXR1⁻ CAF-S1 (grey) among CAF-S1 (Right) in stromal spots of 10 HGSOC Visium sections analyzed at time of diagnosis (N = 4, Treatment naïve) and in residual disease (N = 6, After chemotherapy). n = 37 643 total spots (n = 13 438 Treatment naïve; n = 24 205 after chemotherapy). P-values from Wilcoxon test. (P) % of each deconvoluted CAF-S1 cluster among total CAF-S1 in the 10 HGSOC Visium sections analyzed before (N = 4, Treatment-naïve) and after chemotherapy (N = 6). P-values from Wilcoxon test. In panels with error bars, errors bars indicate SEM.”

p12: Results Fig. 4: “Consistent with these observations, spatial transcriptomic data also showed a significant down-regulation of TEAD-target genes after chemotherapy, in particular in areas enriched in ECM-myCAF (Fig. 4C, D).”

p43-44: Corresponding legend of Fig. 4: “(C) Representative images showing the expression score of TEAD-target genes (Left) and the abundance of deconvoluted ECM-myCAF per spot (Right) in HGSOC Visium sections analyzed at time of diagnosis (Up, Treatment-naïve) and in residual disease (Bottom, After chemotherapy). At least one ECM-myCAF per spot is identified by deconvolution in 2 410 spots out of 24 205 spots analyzed after treatment (average abundance of ECM-myCAF per spot = 2.3; average abundance of all cells per spot = 10). n = 37 643 total spots (13 438 Treatment naïve; 24 205 after chemotherapy). (D) Violin plot showing expression score of TEAD-target genes in stromal spots of spatial transcriptomic data. N = 10 sections (4 Treatment-naïve; 6 after chemotherapy). P-value from Wilcoxon test. Same number of spots as in (C).”

p13: Results Fig. 5: “To get an overview of the spatial organization of CD8⁺ T lymphocytes and ECM-myCAF, we first took advantage of spatial transcriptomic data and cellular HGSOC atlas we built (Supplementary Fig. S4A, B). We mapped the localization and the abundance of each cell population by performing deconvolution using the Cell2location algorithm¹⁰³. We applied non-negative matrix factorization (NMF) analysis to identify patterns of co-localization of cell types and states and found that the ECM-myCAF cluster spatially segregated away from CD8⁺ T lymphocytes (Fig. 5A-C). We confirmed this observation by evaluating the distances of CD8⁺ T lymphocytes to the closest ECM-myCAF-enriched or -depleted spots (Fig. 5D).”

p44-45: Corresponding legend of Fig. 5: “(A) Representative images showing abundance of deconvoluted ECM-myCAF (Right) and differentiated CD8⁺ T lymphocytes (Left) per spot in HGSOC Visium sections analyzed at time of diagnosis (Up, Treatment-naïve) and in residual disease (Bottom, After chemotherapy). The bottom right section shows 2 distinct pathological responses after treatment, one non-responding with a high proportion of residual ECM-myCAF (Left part of the section) and one responding with a reduced ECM-myCAF content (Right part of the section). At least one CD8⁺ T lymphocyte per spot is identified by deconvolution in 5172 spots out of 24 205 spots analyzed after treatment (average abundance of CD8⁺ T cells per spot = 0.67; average abundance of all cells per spot = 11.3). n = 37 643 total spots (n = 13 438 Treatment naïve; n = 24 205 after chemotherapy). (B) Heatmap of the non-negative matrix factorization computed on the deconvolution output using 10 factors on 10 HGSOC sections (4 Treatment-naïve; 6 after chemotherapy). Colors and sizes of circles indicate the scaled cell

type abundance. (C) Representative images showing the spatial distribution and score intensity of ECM-myCAF- and CD8⁺ T cell-enriched factors (Factor 6 and Factor 7, respectively) on a HGSOC sample after treatment. (D) Distribution of the closest distances between CD8-enriched spots and ECM-myCAF-enriched or ECM-myCAF-depleted spots (n = 10) in a radius of 1mm.”

In Figure 3, 5 untreated and 7 treated patient samples were analyzed by scRNAseq. While I appreciate the difficulty of obtaining fresh samples for scRNAseq, there is a concern that these are not matched before/after samples, and there are so few in number. There is such variation between patients in terms of the CAF populations present at baseline, making it a concern as to whether differences are truly due to chemo, or simply random sampling error. Additional data sets publicly available (one HGSOC and one breast cancer) were also analyzed but it is not described how many samples were in the treated vs untreated groups and/or whether any of the samples were paired.

We thank the Reviewer for appreciating the difficulty to get access to fresh HGSOC patient samples at baseline and in residual disease to perform scRNAseq. In order to reduce the random sampling error mentioned by the Reviewer, we have now validated our conclusions in several cohorts of patients, including publicly available scRNAseq data, for which we have now included information on the number of patients analyzed and the number of cells in the new version of the text. By this way, we first confirmed our observations in an independent cohort of HGSOC patients from the Helsinki hospital⁹⁹. From the Helsinki cohort, 7.964 cells (isolated from 11 paired patients at baseline and residual disease) have been annotated as stromal cells by the authors⁹⁹ and 5.658 CAF-S1 were selected based on both FAP expression and CAF-S1 gene signature (1.773 CAF-S1 at baseline and 3885 CAF-S1 after chemotherapy) and conserved for further analysis. From our newly generated HGSOC Curie dataset, 5.618 CAF-S1 were isolated from 12 patients (5 patients at baseline and 7 after chemotherapy). We obtained exactly the same conclusions from the analyses of these 2 independent HGSOC cohorts, thereby strongly confirming the reduction in the amount of ECM-myCAF and the increase in Detox-iCAF content in residual disease samples. Moreover, we validated these conclusions by analyzing 7.959 CAF-S1 isolated from 42 patients with breast cancer (31 at baseline and 11 after chemotherapy). We thank the Reviewer for the careful reading of our paper. We have now added all these information from the 3 independent cohorts analyzed in the **Methods** section (§*Single cell RNA sequencing(scRNAseq) data processing*) **p28**, as well as in the legend of the **New Fig. 3 p41-42 and in Supplementary Fig. 3 p46-47**.

p9 and 10: Results Fig. 3: “.... We first observed that ANTXR1 expression, as well as the percentage of ANTXR1⁺ CAF-S1, were significantly decreased in treated HGSOC compared to treatment-naïve samples (**Fig. 3E, Left**), thereby confirming, at single cell level, the data obtained by IHC and flow cytometry from HGSOC. We also took advantage of the publicly available scRNAseq data from CAF isolated from untreated and chemotherapy-treated samples from an independent HGSOC cohort of patients treated at Helsinki’s hospital⁹⁹ and in breast cancer (BC) patients¹⁰⁰ (**Fig. 3F, G**). By this way, we confirmed that ANTXR1 expression and the proportion of ANTXR1⁺ CAF-S1 were reduced following chemotherapy in an independent cohort of HGSOC patients (Helsinki cohort), as well as in BC (**Fig. 3F, G**), strengthening the validity of our observations both in ovarian and breast cancer.

... To identify the CAF-S1 clusters modulated by chemotherapy, we used the reference-based approach from Seurat to predict and annotate CAF-S1 clusters in scRNAseq data (**Fig. 3H, I and Supplementary Fig. S3A-C**). By this way, we observed high prediction scores of all CAF-S1 clusters in the two HGSOC independent cohorts (Curie and Helsinki cohorts) as well as in BC (**Supplementary Fig. S3A-C, Bottom**)..... Among the different CAF-S1 clusters identified, the ANTXR1⁺ ECM-myCAF was the most abundant in treatment-naïve HGSOC (in both Curie and Helsinki cohorts) and in BC (**Fig. 3H, I and Supplementary Fig. S3E**). Interestingly, the ECM-myCAF was also the CAF-S1 cluster, which decreased the most -compared to all other CAF-S1 clusters- after chemotherapy in HGSOC and in BC (**Fig. 3H, I and Supplementary Fig. S3E**). As the number of CAF analyzed after neoadjuvant chemotherapy in BC was much lower than from treatment-naïve samples (**Supplementary Fig. S3E**), we validated all these observations after down-sampling of treatment-naïve CAF-S1 cells (**Supplementary Fig. S3F**).”

p42: Corresponding legend of Fig. 3: “(E-G) Violin plots of ANTXR1 expression (**Left**) and barplots of the percentage (%) of ANTXR1⁺ CAF-S1 among total CAF-S1 (**Right**) in treatment-naïve patients and after chemotherapy in HGSOC Curie cohort (Prospective cohort 2) (**E**), HGSOC Helsinki cohort (**F**) and breast cancer (BC) cohort (**G**). Data from scRNAseq: n = 5 618 (**E**), 5 658 (**F**) and 7 959 (**G**) CAF-S1 fibroblasts, from N = 12 HGSOC patients, including 5 treatment-naïve and 7 after chemotherapy (**E**), N = 11 paired (before and after treatment) HGSOC patients (**F**), and N = 42 BC patients, including 31 treatment-naïve and 11 after chemotherapy (**G**). P-values from Wilcoxon test (**E-G, Left**) and from Fischer’s Exact test (**E-G, Right**). (**H Left**, UMAP of 5 618 CAF-S1 cells from HGSOC Curie cohort colored by treatment status (**Up**) or by CAF-S1 clusters predicted using label transfer (**Bottom**). N = 12 HGSOC patients, including 5 treatment-naïve and 7 after chemotherapy.

Right, Proportion of CAF-S1 clusters among CAF-S1. P-value from Fisher's exact test. **(I)** Same as in **(H)** for 5 658 CAF-S1 cells from HGSOE Helsinki cohort (N = 11 paired HGSOE patients)."

p47-48: Corresponding legend of Sup. Fig. 3: "(A-C) UMAP showing patient distribution (**Top**) and prediction scores (**Bottom**) of CAF-S1 clusters assessed by label transfer in scRNAseq from HGSOE Curie cohort (A), HGSOE Helsinki cohort (B) and BC cohort (C)..... (E) UMAP of 7 959 CAF-S1 from the BC cohort (N = 42 patients including 31 treatment-naïve and 11 after chemotherapy) colored by treatment status (**Left**) or by CAF-S1 clusters predicted using label transfer (**Middle**). **Right**, Proportion of CAF-S1 clusters among CAF-S1. P-value from Fisher's exact test. **(F-H)** Impact of down-sampling of CAF-S1 in scRNAseq data from the BC cohort. **(F)** UMAP showing CAF-S1 cluster distribution predicted by label transfer. **(G)** Violin plot of ANTXR1 expression (**Left**) and barplot of the % of ANTXR1⁺ CAF-S1 among total CAF-S1 (**Right**) in treatment-naïve patients and after chemotherapy. P-values from Wilcoxon (**Left**) and Fisher's Exact test (**Right**). **(H)** Proportion of CAF-S1 clusters among CAF-S1 in treatment-naïve and chemotherapy-treated samples. P-value from Fisher's Exact test."

In Figure 4, it was observed that there was a decrease in TEAD target genes in CAF-S1 and in ECM-myCAF after chemo, both in scRNAseq data and in samples analyzed by spatial transcriptomics. In Figure 4C (right side) there appears to be virtually no ECM-myCAFs present, which doesn't correspond to the ECM-myCAF shown as being ~15% of CAF-S1 in this sample in Figure 3K. There is a concern that with so few cells of the subtype in question present, whether it's possible to reliably profile the TEAD target genes in this sample. Indeed, the same is true for the scRNAseq data – only percentages are given of each cell type after chemo, but if the ECM-myCAF population is greatly decreased (e.g. in Figure 1F it goes down to a very small proportion of cells), how many cells are actually being analyzed for expression of TEAD target genes?

We agree that there is indeed a decrease of the proportion of ECM-myCAF after chemotherapy, as observed in scRNAseq and spatial transcriptomic data. Consistent with the decrease in the proportion of ECM-myCAF in residual samples, we observed less ECM-myCAF in scRNAseq and less spots in spatial transcriptomic containing ECM-myCAF. To address the point of the Reviewer, we have now increased the number of spatial transcriptomic experiment up to 10 different samples, including 6 samples analyzed after chemotherapy. This now covers 2410 out of 24205 spots after treatment, where at least one ECM-myCAF per spot is identified by deconvolution. The average abundance of ECM-myCAF per spot in these 2410 spots is 2.3, with an average number of total cells detected per spot of 10 (as expected from the technology design). All these information are now given in the **legend of Fig. 4**. Please note that for spatial transcriptomic analysis, cell2location deconvolution estimates the cell type abundance, where non-zero values indicate the number of cells identified from the transcriptomic data, which can be incomplete due to difference in-transcriptomic activity of the different cell types or cells not entirely captured under a spot. Still, cell2location authors showed that it correlates with the real number of cells in each spot¹⁰³, thereby reinforcing data obtained with this algorithm. As requested, we have now included these precisions (average number of ECM-myCAF, and total cells per spot detected, as well as number of spots analyzed) in the Fig. legend of the **New Fig. 4**. We have also included these information for the differentiated CD8⁺ T lymphocytes in the legend of the **New Fig. 5**.

p43: Legend Fig. 4C: "(C) Representative images showing the expression score of TEAD-target genes (**Left**) and the abundance of deconvoluted ECM-myCAF per spot (**Right**) in HGSOE Visium sections analyzed at time of diagnosis (**Up**, Treatment-naïve) and in residual disease (**Bottom**, After chemotherapy). At least one ECM-myCAF per spot is identified by deconvolution in 2 410 spots out of 24 205 spots analyzed after treatment (average abundance of ECM-myCAF per spot = 2.3; average abundance of all cells per spot = 10). n = 37 643 total spots (13 438 Treatment naïve; 24 205 after chemotherapy)."

p44: Legend Fig. 5: "(A) Representative images showing abundance of deconvoluted ECM-myCAF (**Right**) and differentiated CD8⁺ T lymphocytes (**Left**) per spot in HGSOE Visium sections analyzed at time of diagnosis (**Up**, Treatment-naïve) and in residual disease (**Bottom**, After chemotherapy). The bottom right section shows 2 distinct pathological responses after treatment, one non-responding with a high proportion of residual ECM-myCAF (Left part of the section) and one responding with a reduced ECM-myCAF content (Right part of the section). At least one CD8⁺ T lymphocyte per spot is identified by deconvolution in 5172 spots out of 24 205 spots analyzed after treatment (average abundance of CD8⁺ T cells per spot = 0.67; average abundance of all cells per spot = 11.3). n = 37 643 total spots (n = 13 438 Treatment naïve; n = 24 205 after chemotherapy)."

Figure 4A shows that TEAD transcription factor activity is decreased in the CAF-S1 population after chemotherapy and Figure 4B shows TEAD target genes are down in CAF-S1 and in ECM-myCAF after chemotherapy. It would be useful to see the same analyses of the other CAF populations to confirm that this is indeed unique to the ECM-myCAFs as described. Differential expression of YAP1/TEAD signaling in the ECM-myCAF population has not been shown.

We have previously published that some TEAD-target genes are up-regulated in CAF-S1 compared to CAF-S4 in HGSOC (Givel, *Nature Communications*, 2018, Supplementary Data 1 listing genes specifically up-regulated in CAF-S1 compared to CAF-S4⁴⁰). As recommended by the Reviewer, we have now compared the expression of TEAD-target genes in ECM-myCAF clusters and compared it to all the other CAF-S1 clusters the different populations analyzed. We confirmed that expression of TEAD-target genes is significantly higher in ECM-myCAF compared to the other CAF-S1 clusters. As recommended by the Reviewer, we have now added this comparison in the **Supplementary Fig. S5A** and provided better explanation of our rationale. We thank the Reviewer for this suggestion. These data have now been added in the new version of the text, **p12**.

p12: Results: “We already observed that several TEAD-target genes were up-regulated in CAF-S1 compared to CAF-S4 in HGSOC⁴⁰. Moreover, we found that TEAD-target genes were more highly expressed in ECM-myCAF than in all other CAF-S1 clusters (**Supplementary Fig. S5A**).”

It seems that the YAP1-IHC, including nuclear staining, was scored by eye. Based on Figure 4I, this would be very difficult scoring to do. It would be useful to know: how many individuals scored the samples? Were the scorers blinded to the chemo treatment status of the samples? How well did individual scores correlate with each other? Ideally a more unbiased scoring approach (e.g. using image analysis software) would be used.

Evaluation of YAP1 nuclear staining has been evaluated by a biologist and validated by a pathologist in a non-blinded manner. We followed the same protocol for the evaluation of histological scores of all CAF markers, as well as T cell content, in this manuscript. All quantifications gave very consistent results independently of the person. These information are now provided in the **Methods** section, **#Quantification of IHC staining, p26**. In addition, as requested, we validated the method by performing a comparison between the manual counting and automatized detection of CD8⁺ T lymphocytes and found a highly significant correlation. We could not include this data in the paper due to space constraint, but we provide it below (**Supplementary Fig. for the Reviewer**).”

Legend of Sup Fig. for the Reviewer: Correlation plot between number of CD8⁺ T lymphocytes / mm² evaluated by manual counting and by the QuPath positive cell detection tool. N = 10 HGSOC (5 treatment naïve; 5 after chemotherapy). P value from Spearman correlation test.

p26: Methods, end of the paragraph #Quantification of IHC staining: “The evaluation of histological scores of all CAF markers and the quantifications of immune cells were carried out in a non-blinded manner by two independent researchers, including a pathologist. All quantifications gave very consistent results regardless of the person.”

In addition, we have now performed colocalization analyses between YAP1 and ANTXR1, marker of ECM-myCAF, as well as YAP1, ANTXR1 and CD8. These new results are now provided in the **New Fig. 4M-O** and in the **New Fig. 5I-N**. The corresponding method is now described in the **Methods** section (**#In silico analysis of colocalization of IHC staining on serial sections**), **p27**.

p27: Methods: “**#In silico analysis of colocalization of IHC staining on serial sections.** Regions of interest from either serial IHC stained for ANTXR1 and YAP1 or serial IHC stained for both ANTXR1/CD8 and YAP1 were registered using the Fiji plugin BUNwarpJ by manually identifying common features. Registered images were imported in QuPath and epithelial regions were masked by training a pixel classification model as implemented in QuPath. The stromal region was then segmented using Cell Detection on the Hematoxylin channel with default parameters. After quantifying the DAB staining intensity in the first image (YAP1), the cell detections were then transferred to the other registered image to quantify -in the same cell- the subsequent staining using the DAB channel (for ANTXR1), or DAB and Fast Red for ANTXR1 and CD8 co-staining respectively. Measurements were then exported and analyzed in R to plot YAP1 intensity in ANTXR1⁺ CAF and the proportion of YAP1^{+/}-ANTXR1⁺ CAF, or exported in python for colocalization analysis. Cell mean DAB staining intensity of ANTXR1

and YAP1 were used to separate the segmented cells into YAP1⁺ and ANTXR1⁺ cells. Threshold for positivity was assessed using the Single Measurement classifier tool in QuPath. Co-occurrence probability of clusters was computed using the function `gr.co-occurrence` from the Squidpy package¹⁰⁴ and plotted with the function `pl.co-occurrence` according to the CD8⁺ T cell cluster, other cell types than the one of interest were removed from the plot after computing the co-occurrence probability ratio for a simplified presentation of the results. Neighborhood enrichment analysis was conducted after building a connectivity matrix with the function `gr.spatial_neighbors`, computed with `gr.nhood_enrichment` and plotted with `pl.nhood_enrichment` from the Squidpy package, with default parameters. Merged staining of ANTXR1 and YAP1 was performed by deconvoluting the DAB staining of the registered region of interest (ROI) with the Color deconvolution method integrated in Fiji¹⁴⁷. ANTXR1 and YAP1 were colored in red and green, respectively, and the lookup table was inverted in Fiji to highlight the staining. The images were then merge.”

Why assume that the decreased expression of certain genes/pathways within the ECM-myCAF are mediating effects (e.g. T-cell infiltration), rather than just the overall decrease in the number of these cells? Indeed, for most of the measurements shown in Figure 4, one cannot distinguish between these two possibilities (i.e. how can a decrease in expression specifically within the ECM-myCAF populations be distinguished from just an overall decrease in ECM-myCAFs that endogenously express these genes?) This is only shown in Figure 4B (right panel), but as mentioned above, there is concern with respect to not knowing how many cells were actually analyzed in this comparison, especially in the after-chemotherapy group.

We thank the Reviewer for this interesting comment, which points out that some of our results and corresponding methods were perhaps not precisely enough described in the first version of our manuscript. Indeed, we do agree with the Reviewer that the decrease in ECM-myCAF content can induce a global reduced expression of TEAD-target genes, as these genes are highly expressed in ECM-myCAF. We have thus modified the text accordingly to mention the concomitant reduced expression of TEAD-target genes and ECM-myCAF content and highlighting the two possibilities (loss of ECM-myCAF upon treatment / reduced expression of TEAD-target genes in residual ECM-myCAF upon chemotherapy), **p13**. Moreover, as requested by the Reviewers, we have now indicated the precise number of cells analyzed in **Fig. 4B** in the new version of the Text, **p42**. Similarly, the numbers of cells or spots analyzed all along our manuscript have now been indicated in the corresponding legends, as requested.

p12-13: Results: “Moreover, the decrease in nuclear YAP1 staining in stroma following chemotherapy was correlated with the one of FAP and ANTXR1 (**Fig 4K, L**), but not with SMA (**Supplementary Figure S5E**). These observations suggested that the reduced expression of YAP1 and TEAD-target genes after chemotherapy was concomitant to the decrease in the content of ANTXR1⁺ CAF, marker of ECM-myCAF, suggesting that the loss of these CAF after treatment may explain -at least in part- the reduced YAP1/TEAD-signaling pathway. In addition, we also considered that chemotherapy might also reduce YAP1 protein levels in the residual ANTXR1⁺ CAF after treatment. To test this hypothesis, we analyzed the co-localization of YAP1 and ANTXR1 staining in CAF before and after chemotherapy (**Fig. 4M**). We found that the residual ANTXR1⁺ CAF after treatment displayed a decreased YAP1 staining compared to ANTXR1⁺ CAF before treatment (**Fig. 4M, N**). Indeed, while the majority of ANTXR1⁺ CAF were YAP1⁺ before treatment, the residual ANTXR1⁺ CAF were predominantly YAP1^{-Low} (**Fig. 4O**). Taken as a whole, our results indicate that the reduction of YAP1 staining after chemotherapy was mainly due to the concomitant loss of the ECM-myCAF population, on the one hand, but also to the reduction of YAP1 staining in these residual CAF-S1 on the other.”

p43: Legend of Fig. 4B: “(B) Violin plots showing expression of TEAD-target genes before and after chemotherapy in CAF-S1 (**Left**, n = 1 968 Treatment-naïve, n = 3 650 After chemotherapy) and in ECM-myCAF (**Right**, n = 1 099 Treatment-naïve, n = 107 After chemotherapy). Data are from N = 12 HGSOC patients, including 5 treatment-naïve and 7 after chemotherapy. P-values from Mann Whitney test.”

No data are shown to demonstrate that TEAD target genes or YAP1 are differentially expressed in ECM-myCAFs. It would be of interest to see the distribution of expression among the CAF subtypes present.

As indicated above, we have now analyzed the expression of TEAD-target genes in ECM-myCAF compared to all the other CAF-S1 clusters, as requested by the Reviewer. These data have been added in the new version of the **Supplementary Fig. S5A**, and corresponding text. This was indeed very useful and confirm that TEAD-transcription factor activity is particularly high in these cells.

p12: Results: “We already observed that several TEAD-target genes were up-regulated in CAF-S1 compared to CAF-S4 in HGSOC⁴⁰. Moreover, we found that TEAD-target genes were more highly expressed in ECM-myCAF than in all other CAF-S1 clusters (**Supplementary Fig. S5A**).”

In Figure 5C, only the post-chemo-treated sample is used to make the point that differentiated CD8+ cells are spatially segregated away from ECM-myCAFs. Showing this in a single tissue section is insufficient to demonstrate

that this is a general phenomenon. Also, the ECM-myCAF distribution here looks different from that shown in Figure 4C (right panel).

We have now provided many additional analyses to demonstrate that differentiated CD8⁺ cells are spatially segregated away from ECM-myCAF, which are now described in the **New Fig. 5, p13**. These new data first include spatial transcriptomic analysis of ECM-myCAF and differentiated CD8⁺ T lymphocytes based on 10 different HGSOC sections (4 before and 6 after treatment) (**Fig. 5A-D**). In addition, we have now performed CD8 and ANTXR1 co-staining by IHC and histological analysis using an unsupervised approach with analysis of spatial co-occurrence probability between ECM-myCAF and CD8⁺ T cells (**Fig. 5E-H**). Finally, based on the IHC results from the retrospective cohort, we next observed that YAP1 protein levels in the stroma was negatively correlated with the number of CD8⁺ T lymphocytes in HGSOC after chemotherapy, and that the YAP1⁺ ANTXR1⁺ CAF spatially segregate away from CD8⁺ T lymphocytes in HGSOC (**Fig. 5I-N**). All these data are now described in the **New Fig. 5, p13-14**, and its corresponding legend, **p44**.

p13-14: Results: “We mapped the localization and the abundance of each cell population by performing deconvolution using the Cell2location algorithm¹⁰³. We applied non-negative matrix factorization (NMF) analysis to identify patterns of co-localization of cell types and states and found that the ECM-myCAF cluster spatially segregated away from CD8⁺ T lymphocytes (**Fig. 5 A-C**). We confirmed this observation by evaluating the distances of CD8⁺ T lymphocytes to the closest ECM-myCAF-enriched or -depleted spots (**Fig. 5D**). To gain deeper insights into the spatial organization of CD8⁺ T lymphocytes and ANTXR1⁺ CAF at the protein level and at single cell resolution, we performed histological analysis of ANTXR1, CD8 and pan-cytokeratin co-staining in HGSOC (**Fig. 5E**). After cell segmentation, we were able to classify each cell type (ANTXR1⁺ and ANTXR1⁻ CAF, CD8⁺ T lymphocytes and pan-cytokeratin⁺ cancer cells) based on intensities of their specific markers using an unsupervised approach (**Fig. 5F, G**) (See also Methods #Pan-cytokeratin, CD8 and ANTXR1 staining and colocalization analysis). We next compared the spatial co-localization of ANTXR1⁺ and ANTXR1⁻ CAF with CD8⁺ T lymphocytes by applying a co-occurrence method implemented in Squidpy¹⁰⁴, which computes the probability of detecting a cluster of interest (here either ANTXR1⁺ or ANTXR1⁻ CAF) depending on the presence of another cluster (here CD8⁺ T lymphocytes) within an increasing radius. This spatial co-occurrence analysis revealed that ANTXR1⁺ CAF were spatially distant from CD8⁺ T lymphocytes and did not colocalize together, while ANTXR1⁻ CAF were more likely to colocalize with CD8⁺ T cells (**Fig. 5H**). We confirmed this spatial segregation between ANTXR1⁺ CAF and CD8⁺ T lymphocytes by performing IHC co-staining of ANTXR1 and CD8 in the retrospective Curie 1 cohort of HGSOC (**Fig. 5I, J**).

We then hypothesized that the YAP1⁺ ANTXR1⁺ CAF population could play a role in CD8⁺ T cell exclusion. Based on the IHC results from the retrospective Curie 1 cohort, we observed that YAP1 protein levels in the stroma was negatively correlated with the number of CD8⁺ T lymphocytes in HGSOC after chemotherapy (**Fig. 5K**). In contrast, there was no link between CD8⁺ T cell content and YAP1 staining in the epithelium (**Fig. 5L**), suggesting that YAP1 in the stroma but not in the epithelium might affect CD8⁺ T cell density. We thus next performed serial staining of YAP1, ANTXR1 and CD8 and integrated YAP1 in the spatial co-occurrence analysis. We found that the segregation observed between ANTXR1⁺ CAF and CD8⁺ T lymphocytes (shown in **Fig. 5J**) was mainly associated with YAP1⁺ ANTXR1⁺ CAF and that the neighborhood of YAP1⁺ ANTXR1⁺ CAF was specifically depleted of CD8⁺ T lymphocytes (**Fig. 5M, N**). Altogether, these data suggest that YAP1 in ANTXR1⁺ CAF, enriched in ECM-myCAF, spatially segregate away from CD8⁺ T lymphocytes in HGSOC.”

p44: Corresponding legend of Fig. 5: “(A) Representative images showing abundance of deconvoluted ECM-myCAF (**Right**) and differentiated CD8⁺ T lymphocytes (**Left**) per spot in HGSOC Visium sections analyzed at time of diagnosis (**Up**, Treatment-naïve) and in residual disease (**Bottom**, After chemotherapy). The bottom right section shows 2 distinct pathological responses after treatment, one non-responding with a high proportion of residual ECM-myCAF (Left part of the section) and one responding with a reduced ECM-myCAF content (Right part of the section). At least one CD8⁺ T lymphocyte per spot is identified by deconvolution in 5172 spots out of 24 205 spots analyzed after treatment (average abundance of CD8⁺ T cells per spot = 0.67; average abundance of all cells per spot = 11.3). n = 37 643 total spots (n = 13 438 Treatment naïve; n = 24 205 after chemotherapy). (B) Heatmap of the non-negative matrix factorization computed on the deconvolution output using 10 factors on 10 HGSOC sections (4 Treatment-naïve; 6 after chemotherapy). Colors and sizes of circles indicate the scaled cell type abundance. (C) Representative images showing the spatial distribution and score intensity of ECM-myCAF- and CD8⁺ T cell-enriched factors (Factor 6 and Factor 7, respectively) on a HGSOC sample after treatment. (D) Distribution of the closest distances between CD8-enriched spots and ECM-myCAF-enriched or ECM-myCAF-depleted spots (n = 10) in a radius of 1mm. (E) Co-staining of Pan-cytokeratin (blue), ANTXR1 (green) and CD8 (red) in a HGSOC section at baseline. Scale bar = 1mm. (F) Cell segmentation of the HGSOC section shown in (E). Colors represent unsupervised clustering results identifying cancer cells (blue), CD8⁺ T lymphocytes (red), ANTXR1⁺ CAF (green) and ANTXR1⁻ CAF (black). The black box shows the area considered for the analysis of co-occurrence between the different cell types analyzed in (H). Scale bar = 1mm. (G) Same as in (F) for the analysis shown in (H). Scale bar = 1mm. (H) CD8⁺ T lymphocyte co-occurrence probability ratio (*i.e.* co-

occurrence probability to observe a cell type of interest with a CD8⁺ T lymphocyte reported to the probability to find the cell type of interest, computed at increasing distance) with CD8⁺ T lymphocytes, ANTXR1⁺ CAF and ANTXR1⁻ CAF in the ROI showed in (G). (I) Representative views of ANTXR1 (brown) and CD8 (red) IHC co-staining on paired HGSOC sections before and after chemotherapy. Stars show areas enriched in ANTXR1^{High} CAF with almost no CD8 detected (mainly observed before treatment) and arrows highlight ANTXR1^{Low-Neg} zones with high CD8⁺ T cell density (mainly observed after chemotherapy). Scale bars, 50 μ m. (J) **Left**, Segmentation of ANTXR1⁺ CAF, ANTXR1⁻ CAF and CD8⁺ T lymphocytes in the stroma of a residual sample. Scale bar = 50 μ m. **Right**, CD8⁺ T lymphocytes co-occurrence probability ratio with CD8⁺ T lymphocytes, ANTXR1⁺ CAF and ANTXR1⁻ CAF in the section showed in Left. (K) Correlation plot with linear regression line between H-score of YAP1 in the stroma and the number of total CD8⁺ T lymphocytes in HGSOC after chemotherapy. Each dot represents one tumor (N = 35). P value from Spearman correlation test. (L) Same as in (K) between H-score of YAP1 in the epithelium and the number of total CD8⁺ T lymphocytes in HGSOC after chemotherapy. (M) Representative views of ANTXR1 (brown) and CD8 (red) IHC co-staining on the same sections, together with YAP1 (brown) stained on serial sections in paired (before and after chemotherapy) HGSOC. Top, Arrows show YAP1 nuclear staining in ANTXR1^{High} areas with almost no CD8 T cell detected. Bottom, Arrowheads show high CD8 T lymphocyte density in ANTXR1^{Low} and YAP1^{Low} areas. Scale bars, 50 μ m. (N) **Left**, Co-occurrence probability ratio between CD8⁺ T lymphocytes and residual YAP1⁺ ANTXR1⁺ CAF in HGSOC after treatment. **Right**, Neighbors enrichment score computed on the spatial connectivity graph between CD8⁺ T lymphocytes, YAP1⁺ ANTXR1⁺ CAF and other CAF.”

Reviewer #3 (Remarks to the Author): with expertise in cancer associated fibroblasts, chemotherapy, TME

In this work, authors have reported that FAP+ CAF-S1 myofibroblasts decrease following chemotherapy in patients with HGSOC. In particular, they observe a reduction of ANTXR1-expressing CAFs, which express nuclear YAP1. Chemotherapy therefore results in an overall decrease of YAP1-expressing CAFs, which might act on CD8+ T cell infiltration and activation through secretion of CYR61.

This study includes data from immunohistochemistry, flow cytometry, single cell RNA sequencing and spatial transcriptomic analyses from three HGSOC cohorts. However, throughout the manuscript there are severe quantification and technical inconsistencies, which are particularly concerning for Figures 1 and 2. Addressing such issues is crucial to sustain the authors' messages and conclusions.

We thank the Reviewer for the careful reading of our paper. We have now answered to all the concerns and done our best to clarify some points, which were unclear in the initial version of the paper.

Major concerns

1) Relative abundance of stroma and epithelium

Based on the results displayed in figures 1A-D, authors claim a reduction of tumor epithelium after chemotherapy in matched samples. Additionally, flow cytometry analyses in Figure 1K show a decreased percentage of EPCAM+ epithelial cells after chemotherapy (in unmatched samples). Yet, the percentage of EPCAM+ cells showed in Figure 1B, D measured in matched samples before and after therapy remains similar indicating no decrease of tumor epithelium, which contradicts the data in Figure 1C and 1K. How do these results fit together to demonstrate an epithelial reduction?

In this line, if EPCAM+ cells percentages remain constant in matched patient samples before/after chemotherapy, it follows that the percentage of stroma should also remain constant. Instead, authors claim that it is increased based on morphological assessment.

On the other hand, the example depicted in supplementary Figure 1B seems to show a large reduction in the amount of CAFs in patients treated with chemotherapy, which contradicts the increase % of total CAFs displayed in Figure 1K. Could authors clarify these discrepant data? Please provide % for each cell subset in supplementary Figure S1B (and in Figure 1J). Of note, cell subsets have to be selected equally in the two analyzed groups for proper comparison of flow cytometry profiles. In the flow cytometry analyses depicted in supplementary Figure 1B and Figure 1K, the gating strategy is altered between “treatment naïve” and “after chemotherapy” samples, particularly in CD45+ vs EPCAM plots. This fact might pose a relevant bias towards the quantification in Figure 1K. How this variation in selecting areas is impacting the results of the comparison between “treatment naïve” and “after chemotherapy” samples? If this stromal increase is not fully demonstrated, authors cannot claim that the increased infiltration reported in Figure 2 is dependent on an overall increase of the stroma.

In this same direction, samples used in Supplementary Figure 3 shows what it seems to be a similar stromal content before and after chemotherapy. Could authors provide with a percentage of stroma and epithelium from the spatial transcriptomic analysis? How does it fit with all the histological and flow cytometry quantifications depicted in Figure 1?

We are sorry for the lack of clarity in the first version of our manuscript. In **Fig. 1**, by studying the retrospective cohort of paired samples, we show that the global epithelial content -assessed by pathologists based on morphological criteria- decreases after chemotherapy in most HGSOc patients, as expected after treatment. We also wondered whether this global decrease in epithelium content could be associated with a lack of epithelial features. To test this hypothesis, we performed immunohistochemistry staining of EPCAM, a well-known epithelial marker. Most epithelial cancer cells were EPCAM⁺ before treatment and the intensity of the staining remained high after chemotherapy in residual epithelial cells (**Fig. 1D**), indicating that residual epithelial cells after chemotherapy expressed EPCAM at the same levels as before treatment. Thus, after chemotherapy, we observed a global decrease in the epithelial content, but residual cancer cells maintain their epithelial features, evaluated by EPCAM staining. We do agree with the Reviewer that the description of these data in the first version of the text was misleading, and we are sorry for that. As requested, we modified the presentation of the results in the **New Fig. 1** and the text accordingly, **p6**.

p6: Results: “As expected, most HGSOc exhibited a lower epithelial content after chemotherapy compared to the corresponding samples at time of diagnosis (**Fig. 1A, C**). We next wondered whether this global decrease in the epithelium content could be associated with a lack of epithelial features by performing immunohistochemistry (IHC) staining of EPCAM, a well-known epithelial marker (**Fig. 1B, D**). Although a small number of patients exhibited a decrease in EPCAM staining, epithelial cancer cells were EPCAM⁺ before treatment and the intensity of the staining remained high after chemotherapy in most patients (**Fig. 1D**), indicating that residual epithelial cells after chemotherapy expressed EPCAM at the same levels as before treatment.”

p40: Corresponding legend of Fig. 1: “(C) Percentage (%) of epithelium (determined by morphological analysis on HES staining) in HGSOc before and after chemotherapy (N = 35 patients, n = 70 matched samples). Data are shown using paired (Left, P-value from paired Wilcoxon test) and unpaired (Right, P-value from Mann-Whitney test) statistical analyses. (D) Intensity of EPCAM staining in epithelial cells, before and after chemotherapy, ranging from 0 to 4 (N = 35). Paired (Left, P-value from paired Wilcoxon test) and unpaired (Right, P-value from Mann-Whitney test) analyses.”

These observations are consistent with the global reduction of epithelial content assessed by flow cytometry in a prospective cohort of patients (**Fig. 1K**). Indeed, as EPCAM intensity remains similar before and after treatment, measuring the percentages of EPCAM⁺ epithelial cells by flow cytometry enabled us to evaluate the global proportion of cancer cells in fresh samples by flow cytometry. We thus validated the decrease of the epithelial content after chemotherapy in the prospective cohort of HGSOc patients (**Fig. 1K**). Decrease of epithelial content after chemotherapy is thus validated in both the retrospective and prospective cohorts of HGSOc patients we analyzed. We are sorry if these results were not clearly described in the first version of the manuscript. And we thank the Reviewer for this comment, which improves the description of these data and leads to the modifications of the **Fig. 1** and the corresponding **Text**, **p7**.

p7: Results: “Among viable cells isolated from these fresh HGSOc, we identified epithelial (EPCAM⁺), hematopoietic (CD45⁺), endothelial cells (CD31⁺) and red blood cells (CD235a⁺) (**Supplementary Fig. S1C**). As shown above in the retrospective cohort, following chemotherapy, we first detected a decrease in the percentage of epithelial cells among viable cells, together with a concomitant increase in total CAF content (**Fig. 1K**).”

As requested, we have now shown in (**Fig. 1B**) representative images showing an increase in the global stromal content, as scored by both a biologist and a pathologist in most patients (as shown in **Fig. 1E**). Indeed, we observed a global increase in stromal content both by IHC and by flow cytometry analyses. There is no discrepancy in these data. Still, to avoid any misunderstanding, we have now introduced an additional representation of the quantifications (**Fig. 1C-E**) with a global analysis of the cohort using unpaired statistical tests for better visualization of the global variations observed.

p6: Results: “In contrast to the global reduction of the epithelium, the stromal content significantly increased after chemotherapy (**Fig. 1E**).”

Of note, we would like also to insist on the fact that the global increase in the stromal content after chemotherapy is not only detected by IHC and by flow cytometry, but also in the 10 spatial transcriptomic sections that we have now analyzed in the new version of the manuscript (**Supplementary Fig. 4D, E**).

p11: Results: “We first performed pathological annotations to distinguish epithelial and stromal compartments in these HGSOc sections (**Supplementary Fig. S4D**). We confirmed that the amount of stroma detected, as well as the number of CD3⁺ and CD8⁺ T lymphocytes, after chemotherapy were higher than in treatment-naïve (**Supplementary Fig. S4E, F**), confirming observations obtained by IHC and flow cytometry...”

In addition, as requested by the Reviewer, we have now also added the percentages of all populations detected in flow cytometry analyses in (**Fig. 1J**) and in (**Supplementary Fig. 1C**). Moreover, to address the Reviewer’s

comment in a more precise way, we have now provided another representative example of the flow cytometry analysis from HGSOC, where the increase in stroma after chemotherapy is more representative of the global increase detected in the global population. These new data have been included in the **New Supp. Fig. 1C**.

Each cell population analyzed by flow cytometry from “treatment-naïve” and “after chemotherapy” samples (**Fig. 1K**) has been analyzed by using the same gating strategy from one patient to the other, avoiding any bias in the analyses. Minor variations in the fluorescence intensity of CD45⁺ and EPCAM⁺ populations could be observed in acquisitions over the few years that were required to collect all samples of the Prospective cohort 1. We have now specified these points in the **Methods** section, **p24**.

p24: Methods: “#Flow cytometry on HGSOC samples, Characterization of CAF populations:Each cell population was selected using the same gating strategy for all the patients’ samples analyzed by flow cytometry. Minor variations in the fluorescence intensity of CD45⁺ and EPCAM⁺ populations could be observed in acquisitions over the few years that were required to collect all samples of the Prospective cohort 1.”

As requested, we have now provided spatial transcriptomics data from a higher number of HGSOC patients, and we reached now a total of 10 HGSOC patients analyzed with this technology. We would like to emphasize that, at the time of this review, there is no publicly available spatial transcriptomics data after chemotherapy in HGSOC patients; data are only available at baseline. We thus generated new data and performed new spatial transcriptomic analysis on additive HGSOC naïve of treatment and after chemotherapy for a total of 10 HGSOC patients. By adding these new analyses, we validated our initial conclusions and observed after chemotherapy a general increase in the stromal content, a reduced FAP (CAF-S1) and ANTXR1 (ECM-myCAF) expression, a decreased YAP-1/TEAD-dependent signaling pathway and an increased CD8⁺ T cell content. These new spatial transcriptomic data have now been included in the new version of the manuscript in the **New Fig. 3, 4, 5, Sup Fig. 4**, and corresponding description in the Results section **p11-13**, and their corresponding legends **p43-45**.

p11, Results Fig. 3: “Finally, we aimed to define the spatial localization and contribution of CAF-S1 clusters in HGSOC at time of diagnosis and in residual disease. To do so, we used the Visium technology and compared spatial transcriptomic data (see Methods #*Spatial Transcriptomics*) from 10 different HGSOC samples collected at baseline and after chemotherapy (**Fig. 3N-P**). We first performed pathological annotations to distinguish epithelial and stromal compartments in these HGSOC sections (**Supplementary Fig. S4D**). We confirmed that the amount of stroma detected, as well as the number of CD3⁺ and CD8⁺ T lymphocytes, after chemotherapy were higher than in treatment-naïve (**Supplementary Fig. S4E, F**), confirming observations obtained by IHC and flow cytometry and validating that samples selected for spatial transcriptomic as representative). In order to extract the signals from those compartments, we evaluated the abundance of the different CAF-S1 clusters in each spot within the sections by applying the deconvolution method cell2location¹⁰³, using the matrix of reference cell types from the HGSOC cellular atlas (**Supplementary Fig. S4A, B**) as input. By this way, we could explore the spatial localization of each CAF-S1 cluster in the 10 HGSOC collected and compared the baseline and residual states. We first confirmed within these tissue sections that ANTXR1 expression level and the content in ANTXR1⁺ CAF-S1 were significantly reduced after chemotherapy compared to treatment-naïve section (**Fig. 3O**). In these sections, we identified the different CAF-S1 clusters and confirmed that, among them, the proportion of the ANTXR1⁺ ECM-myCAF cluster was this one, which accumulated before treatment and decreased the most after chemotherapy (**Fig. 3P**). We also observed that the content in the Detox-iCAF cluster increased after treatment (**Fig. 3P**). Taken as a whole, these data highlight the relevance of the decrease in the ANTXR1⁺ ECM-myCAF cluster content after chemotherapy in various cancer types by using several complementary approaches.”

p43: Corresponding legend of Fig. 3: “(N) Representative images of ANTXR1 expression in HGSOC Visium sections at time of diagnosis (**Up**, Treatment-naïve) and in residual disease (**Bottom**, After chemotherapy). (O) Violin plot of ANTXR1 expression (**Left**) and barplot of the mean proportion of deconvoluted ANTXR1⁺ CAF-S1 (black) and ANTXR1⁻ CAF-S1 (grey) among CAF-S1 (**Right**) in stromal spots of 10 HGSOC Visium sections analyzed at time of diagnosis (N = 4, Treatment naïve) and in residual disease (N = 6, After chemotherapy). n = 37 643 total spots (n = 13 438 Treatment naïve; n = 24 205 after chemotherapy). P-values from Wilcoxon test. (P) % of each deconvoluted CAF-S1 cluster among total CAF-S1 in the 10 HGSOC Visium sections analyzed before (N = 4, Treatment-naïve) and after chemotherapy (N = 6). P-values from Wilcoxon test. In panels with error bars, errors bars indicate SEM.”

p12: Results Fig. 4: “Consistent with these observations, spatial transcriptomic data also showed a significant down-regulation of TEAD-target genes after chemotherapy, in particular in areas enriched in ECM-myCAF (**Fig. 4C, D**).”

p43-44: Corresponding legend of Fig. 4: “(C) Representative images showing the expression score of TEAD-target genes (**Left**) and the abundance of deconvoluted ECM-myCAF per spot (**Right**) in HGSOC Visium sections analyzed at time of diagnosis (**Up**, Treatment-naïve) and in residual disease (**Bottom**, After chemotherapy). At least one ECM-myCAF per spot is identified by deconvolution in 2 410 spots out of 24 205 spots analyzed after

treatment (average abundance of ECM-myCAF per spot = 2.3; average abundance of all cells per spot = 10). n = 37 643 total spots (13 438 Treatment naïve; 24 205 after chemotherapy). (D) Violin plot showing expression score of TEAD-target genes in stromal spots of spatial transcriptomic data. N = 10 sections (4 Treatment-naïve; 6 after chemotherapy). P-value from Wilcoxon test. Same number of spots as in (C).”

p13: Results Fig. 5: “To get an overview of the spatial organization of CD8⁺ T lymphocytes and ECM-myCAF, we first took advantage of spatial transcriptomic data and cellular HGSOC atlas we built (Supplementary Fig. S4A, B). We mapped the localization and the abundance of each cell population by performing deconvolution using the Cell2location algorithm¹⁰³. We applied non-negative matrix factorization (NMF) analysis to identify patterns of co-localization of cell types and states and found that the ECM-myCAF cluster spatially segregated away from CD8⁺ T lymphocytes (Fig. 5 A-C). We confirmed this observation by evaluating the distances of CD8⁺ T lymphocytes to the closest ECM-myCAF-enriched or -depleted spots (Fig. 5D).”

p44-45: Corresponding legend of Fig. 5: “(A) Representative images showing abundance of deconvoluted ECM-myCAF (Right) and differentiated CD8⁺ T lymphocytes (Left) per spot in HGSOC Visium sections analyzed at time of diagnosis (Up, Treatment-naïve) and in residual disease (Bottom, After chemotherapy). The bottom right section shows 2 distinct pathological responses after treatment, one non-responding with a high proportion of residual ECM-myCAF (Left part of the section) and one responding with a reduced ECM-myCAF content (Right part of the section). At least one CD8⁺ T lymphocyte per spot is identified by deconvolution in 5172 spots out of 24 205 spots analyzed after treatment (average abundance of CD8⁺ T cells per spot = 0.67; average abundance of all cells per spot = 11.3). n = 37 643 total spots (n = 13 438 Treatment naïve; n = 24 205 after chemotherapy). (B) Heatmap of the non-negative matrix factorization computed on the deconvolution output using 10 factors on 10 HGSOC sections (4 Treatment-naïve; 6 after chemotherapy). Colors and sizes of circles indicate the scaled cell type abundance. (C) Representative images showing the spatial distribution and score intensity of ECM-myCAF- and CD8⁺ T cell-enriched factors (Factor 6 and Factor 7, respectively) on a HGSOC sample after treatment. (D) Distribution of the closest distances between CD8-enriched spots and ECM-myCAF-enriched or ECM-myCAF-depleted spots (n = 10) in a radius of 1mm.”

2) IHC analysis and H-scores

In Figure 1, please provide the rationale behind performing an H-score quantification for EPCAM. Is EPCAM intensity informative? This is not addressed in the manuscript. Remarkably, there are several patients in Figure 1D center that experience a decrease in EPCAM staining intensity. Does this imply a transformation such as EMT? Is EPCAM staining intensity also affected in flow cytometry analyses?

H-score evaluates an overall staining intensity. Hence, in figure 1G, the H-score is not informative of the actual relative abundance of a certain CAF subpopulation or their localization within the tumor. This could explain some inconsistencies seen between IHC and flow cytometry analyses. For instance, total CAF-S1 does not vary between “treatment naïve” and “after chemotherapy” in Figure 1L, which argues against the decrease reported in the manuscript for Figure 1G. While indeed ECM-myCAFs seem to be drastically reduced in treated samples, many other CAF-S1 subtypes are increasing. This overall increase seems to compensate for the ECM-myCAF decrease, which could explain Figure 1L. Please clarify these points.

Also, it is impossible to ascertain whether T cells are more abundant in, for instance, regions enriched with CAF-S2/S3 from the data presented in Figure 3. Overall, could authors provide with CAF subpopulations abundance and T cell density data? In particular, it would be interesting to match T cell infiltration for every CAF subpopulations. In this sense, how are CAF-S2/S3/S4 represented in the spatial transcriptomic data? Could authors show how the percentage of all 4 main CAF subpopulations is varying after chemotherapy? It would also be important to contrast these CAF subpopulations with the location of T cells.

For IHC analyses, there are powerful open-source softwares that allow accurate quantification of tumor areas and cells with positive DAB staining. Given the aforementioned concerns, authors should reanalyze the abundance of EPCAM⁺ cells, stromal compartment and T cell infiltration more accurately, using IHC analysis softwares such as QuPath.

As mentioned above, the global epithelial content evaluated on morphological criteria decreases after chemotherapy. To evaluate if this global epithelium decrease was associated with a lack of epithelial features, we performed IHC staining of the epithelial marker EPCAM. Most epithelial cancer cells were EPCAM⁺ before treatment and the staining remained high after chemotherapy. As indicated by the Reviewer, even if a small number of patients exhibited a decrease of EPCAM staining intensity, most patients did not show variation in EPCAM staining intensity. This is why there is no statistical difference in EPCAM staining intensity before and after chemotherapy. We hope the Reviewer will agree that we cannot conclude on EMT process on that basis, and we establish our description and conclusion on the global population, as recommended. Still, we do agree that the description of these data in the first version of the text could be misleading, and we are sorry for the initial lack of clarity. To avoid any misunderstanding, we have now introduced an additional representation of the quantifications

(Fig. 1C-E) with a global analysis of the cohort using unpaired statistical tests for better visualization of the global variations observed. We have now modified the text accordingly, p6.

p6: Results: “We first evaluated the response to chemotherapy in each HGSOC patient by comparing epithelial and stromal content before and after chemotherapy (Fig. 1A-D). Tumor epithelium content was first assessed based on morphological criteria (Fig. 1A, C). As expected, most HGSOC exhibited a lower epithelial content after chemotherapy compared to the corresponding samples at time of diagnosis (Fig. 1A, C). We next wondered whether this global decrease in the epithelium content could be associated with a lack of epithelial features by performing immunohistochemistry (IHC) staining of EPCAM, a well-known epithelial marker (Fig. 1B, D). Although a small number of patients exhibited a decrease in EPCAM staining, epithelial cancer cells were EPCAM positive before treatment and remained high after chemotherapy in most patients (Fig. 1D), indicating that residual epithelial cells after chemotherapy expressed EPCAM at the same levels as before treatment.”

p40: Corresponding legend of Fig. 1: “(C) Percentage (%) of epithelium (determined by morphological analysis on HES staining) in HGSOC before and after chemotherapy (N = 35 patients, n = 70 matched samples). Data are shown using paired (Left, P-value from paired Wilcoxon test) and unpaired (Right, P-value from Mann-Whitney test) statistical analyses. (D) Intensity of EPCAM staining in epithelial cells, before and after chemotherapy, ranging from 0 to 4 (N = 35). Paired (Left, P-value from paired Wilcoxon test) and unpaired (Right, P-value from Mann-Whitney test) analyses.”

As described in the **Methods** section, H-scores correspond to the % of cells stained multiplied by the staining intensity. H-score evaluation of all CAF markers and the quantifications of immune cells were carried out in a non-blinded manner by two independent researchers, including a pathologist. As mentioned by the Reviewer, this is the combined analysis of several CAF markers, which enable identification of CAF populations. This is why we previously established the decision tree algorithm (Costa, *Cancer Cell*, 2018; Givel, *Nat. Commun.*, 2018), which is also provided in **Fig. 1H** and described in the **Methods** section, p26 (#Decision tree algorithm for prediction of CAF population identity). Thus, H-score of each individual CAF marker (shown in **Fig. 1G**) is not sufficient to identify CAF populations. Application of the decision tree on H-scores is required to identify each CAF population. These results are presented in **Fig. 1I**, therefore combining **Fig. 1H** and **Fig. 1G**, p6-7. Concerning flow cytometry, analysis of the intensity of FAP is much more sensitive, and flow cytometry enabled us to differentiate FAP^{High} and FAP^{Low-med} CAF-S1 populations, which was not possible by IHC. Thanks to this more sensitive method, we found that chemotherapy mainly reduced FAP^{High} CAF-S1, enriched in myCAF (**Fig. 1L**) and described p7. This observation is consistent with the fact that ECM-myCAF (FAP^{High} CAF-S1) is the most abundant CAF-S1 cluster in HGSOC before treatment and the most reduced after chemotherapy, as shown **Fig. 3**. We have now clarified these different points in the new version of the manuscript.

p7-8: Results: “We distinguished the four different CAF populations by flow cytometry in HGSOC prior to treatment (**Fig. 1J-L**), as observed by IHC. In addition, thanks to the sensitivity of the flow cytometry method, we were able to distinguish FAP^{Low-Med} from FAP^{High} CAF-S1, previously shown to characterize inflammatory (iCAF) and myofibroblastic (myCAF) clusters, respectively^{47, 50, 82}. As shown above in the retrospective cohort, following chemotherapy, we first detected a decrease in the percentage of epithelial cells among viable cells, together with a concomitant increase in total CAF content (**Fig. 1K**). While the global CAF content increased, we observed variations in CAF-S1 and CAF-S4 content. Indeed, the proportion of the CAF-S4 population was reduced in treated samples (**Fig. 1L**). In addition, flow cytometry analysis enabled us to detect that, among the CAF-S1 population, the content in FAP^{High} CAF-S1 (enriched in myCAF clusters) was the most reduced after chemotherapy (**Fig. 1J, L**). Taken as a whole, these data show that the abundance of both CAF-S4 and FAP^{High} CAF-S1 populations significantly decreases after chemotherapy in HGSOC patients.”

By analyzing breast and ovarian cancer sections, we have already published that the stroma enriched in CAF-S1 (compared to the other CAF-S2/3/4 populations) plays an important role in T cell infiltration before chemotherapy (Costa, *Cancer Cell*, 2018; Givel, *Nat Commun*, 2018; Kieffer, *Cancer Discovery*, 2020). Here, we complement these data and go a step further by analyzing in detail the impact of chemotherapy on the immunosuppressive CAF-S1 population, in particular on CAF-S1 clusters.

All IHC staining, analysis and quantifications have been performed in close collaborations and validated by pathologists, who are authors of this paper (Pr. A. Vincent-Salomon and Dr. L. Djerroudi), who are well-aware about the QuPath software. Still, as requested by the Reviewer, we have now compared the manual counting of the number of T lymphocytes / mm² by with the evaluation by QuPath positive cell detection tooQ. We validated that QuPath quantifications are highly correlated with manual quantifications in the different sections. This result is shown in the Sup **Fig. for Reviewer** below, which cannot be added in the manuscript due to space constraints.

Legend of Sup Fig. for the Reviewer: Correlation plot between number of CD8⁺ T lymphocytes / mm² evaluated by manual counting and by the QuPath positive cell detection tool. N = 10 HGSOC (5 treatment naïve; 5 after chemotherapy). P value from Spearman correlation test.

p26: Methods, end of the paragraph #Quantification of IHC staining: “The evaluation of histological scores of all CAF markers and the quantifications of immune cells were carried out in a non-blinded manner by two independent researchers, including a pathologist. All quantifications gave very consistent results regardless of the person.

3) T cell stromal and epithelial infiltration

Assuming that the density of T cells remains invariable within the stroma before and after chemotherapy, this will inherently mean that any relative decrease in CAF-S1/S4 and/or increase in CAF-S2/S3 won't affect infiltration. Authors make a poor connection between this effect and the actual proportion of CAF subpopulations, and any correlation depicted in Figure 2 does not fully demonstrate causality. As mentioned in comment 2), it would be interesting to know more about the relative abundance of CAF subpopulations to verify their influence on T cell infiltration.

On the other hand, the only relevant effect is an increased infiltration of T cells within the epithelium. This could result directly from the effect of chemotherapy on T cells, or indirectly through the exposure of neoantigens released from dying tumor cells. In this direction, authors should provide with evidence that an increased infiltration is indeed mediated by shifts between CAF subpopulations and not tumor-lymphocyte interactions that are independent from CAFs.

There are concerns on how the authors reach to negative correlations between CAF-S1 and T cells in Figures 2F-H. First, in Supplementary Figures 2C-E, no correlations has been observed between these CAFs and T cells, which suggest that CAF-S1 has no role in modulating T cell infiltration. This makes any conclusion from Figure 2F very difficult, and suggest that FAP and T cell decrease are actually two independent events instead of a cause-effect. Please clarify. Secondly, merging points from “treatment naïve” and “after chemotherapy” samples in the same XY graph is not an accurate approach to find correlations in Figure 2H. Indeed, treatment naïve samples seem to exert the whole statistical effect, while there doesn't seem to be any correlation between %CD8/CD3 and CAF-S1 in samples after chemotherapy. In contrast, authors do segregate samples before and after chemotherapy in different XY plots for Figures 4G and 4H, where they show statistically significant correlations. Therefore, it is important to provide with accurate percentages of CAF-S1 and T cells in both groups separately, and analyze how different they are before and after chemotherapy, and with correlations in both groups separately. Please provide correlation analysis for %CD8/CD3 and CAF subsets in “treatment naïve” samples only. Please provide correlation analysis for %CD8/CD3 and CAF subsets in “after chemotherapy” samples only. As mentioned above, cell population selection in flow cytometry should be consistent between samples to achieve accurate comparative analysis. Yet again, selection areas differ between “treatment naïve” and “after chemotherapy” samples in Figure 2G. How this variation in selecting areas is impacting the results of the comparison between “treatment naïve” and “after chemotherapy” samples?

When the Reviewer indicates that “the density of T cells remains invariable within the stroma before and after chemotherapy”, there is a misunderstanding. Indeed, there is a significant difference in the density of T cells (in particular CD3⁺ and CD8⁺ T lymphocytes) before and after chemotherapy, in particular in the stroma. This is shown Fig. 2C. In more details, we detected a higher proportion of T cells at the surface of stroma than epithelium both before and after treatment, consistent with the fact that T cell density is higher in stroma than in epithelium. Importantly, after chemotherapy, the T cell density increases (compared to before treatment) in both compartments

but it reaches its highest level in the stromal compartment after chemotherapy (Fig. 2C). This highest density at the surface of stroma after treatment is true for both CD3⁺ and CD8⁺ T lymphocytes (Fig. 2C). Thus, the density of T cells significantly increases after chemotherapy in the stroma.

As we also detected an increase in the proportion of stroma upon chemotherapy (Fig. 1), we tested if this increased stromal content could be linked -in any way- to the increased content in TILs after treatment, which could reinforce the role of stroma in T cell infiltration. To test this hypothesis, we normalized the number of TILs per unit surface of stroma (Fig 2D, Stroma). To do so, we divided T cell density by the proportion of stroma per patient. Interestingly, this normalization abrogates the difference of TIL density before/after treatment, thereby confirming that TIL infiltration after chemotherapy is associated with the increased stromal content. In other words, the more the stromal content increases after chemotherapy, the more T cells infiltrate the tumors. This underlines the importance of the stroma after chemotherapy, and justifies its characterization, in particular its link with CAF-S1. We have tried to re-write the text to better explain our different analyses, p8.

p8, Results: “We observed that CD3⁺ and CD8⁺ TILs significantly accumulated after chemotherapy in HGSOC, with the same tendency for FOXP3⁺ T cells but without reaching significance (Fig. 2B). CD3⁺, CD8⁺ and FOXP3⁺ TILs infiltrated more the stroma than the epithelium (Fig. 2C and Supplementary Fig. S2A). Importantly, the CD3⁺ and CD8⁺ T cell density reached their highest levels in the stromal compartment after chemotherapy (Fig. 2C), highlighting the potential importance of this compartment after treatment. As we detected an increase in the proportion of stroma upon chemotherapy (Fig. 1E), we tested if the increased content in TILs after treatment could be linked to this increased stromal content, which could reinforce the role of stroma in T cell density. To test this hypothesis, we normalized the number of TILs per unit surface of stroma (Fig 2D, Stroma). Interestingly, this normalization abrogated the difference of TIL density before/after treatment in the stroma (Fig. 2D and Supplementary Fig. S2B), showing that TIL density after chemotherapy is associated with the overall enrichment in stroma and underlying the importance of the stroma in this process.”

Regarding causality, we do agree with the Reviewer that correlations shown Fig. 2 do not demonstrate causality and we softened our conclusions all along the Text. However, we have now also provided additional experiments confirming the link between the abundance of the ECM-myCAF CAF-S1 cluster and reduced CD8⁺ T cell cytotoxicity. By using functional assays, we have now shown that CD8⁺ T lymphocyte migration and CD8⁺ T cell cytotoxicity are regulated by 2 different FAP⁺ CAF (CAF-S1) clusters. We now demonstrate that the Detox-iCAF cluster (which belongs to the iCAF clusters of the CAF-S1 population but is ANTXR1-) stimulates the migration of CD8⁺ T lymphocytes. In contrast, ECM-myCAF directly decrease CD8⁺ T lymphocyte cytotoxicity by reducing the secretion of Granzyme, Perforin and IFN γ , and by increasing the percentages of PD-1⁺ CD8⁺ T lymphocytes, while Detox-iCAF are unable to do so. Importantly, we now also demonstrate that CD8⁺ T lymphocytes, which have been co-cultured with ECM-myCAF, show reduced capacity to kill cancer cells. Finally, all ECM-myCAF-mediated effects tested by functional assays and listed above are significantly reduced when YAP1 expression is silenced in ECM-myCAF, thereby demonstrating that these effects are dependent of the YAP1 co-transcription factor. All these data are consistent with the content in these cells in HGSOC patients. These new functional assays are shown in the New Fig. 6 and new Supplementary Fig. S7 and described in the Results' section p14-15.

p14-15: Results: “We next aimed to investigate the role of YAP1 in ECM-myCAF on CD8⁺ T lymphocytes by performing functional assays. We sought to compare the impact of YAP1 silencing in ECM-myCAF on CD8⁺ T cell cytotoxic activity, and used Detox-iCAF as control. To do so, we first isolated Detox-iCAF and ECM-myCAF primary fibroblasts from HGSOC patient samples and evaluated their impact on CD8⁺ T lymphocytes isolated from peripheral blood mononuclear cells (PBMC) of healthy donors. We first verified the identity of these fibroblasts in culture by flow cytometry using specific markers (Supplementary Figure S6A) and by bulk RNAseq (Supplementary Figure S6B-D) and validated that they corresponded to the ECM-myCAF and Detox-iCAF clusters, respectively. We then analyzed the impact of these two CAF-S1 clusters on CD8⁺ T cell cytotoxicity, considering both markers and activity. We found that, upon co-culture, ECM-myCAF significantly increased the percentage of PD-1⁺ CD8⁺ T lymphocytes, while Detox-iCAF did not (Fig. 6A and Supplementary Figure S7A). Moreover, this increase in PD-1⁺ CD8⁺ T cells by ECM-myCAF was concomitant to the reduced percentages of granzyme B⁺, perforin⁺ and IFN- γ ⁺ CD8⁺ T lymphocytes (Fig. 6B-D and Supplementary Figure S7A). Here again, the impact on the percentages of CD8 T cells positive for cytotoxic markers was specific of the ECM-myCAF fibroblasts and not detected with Detox-iCAF (Fig. 6B-D and Supplementary Figure S7A). Interestingly, we confirmed that ECM-myCAF but not Detox-iCAF reduced CD8⁺ T cell cytotoxic activity (Fig. 6E). Indeed, we observed that CD8⁺ T lymphocytes showed reduced capacity to kill cancer cells after co-culture with ECM-myCAF but not with Detox-iCAF (Fig. 6E), thereby confirming ECM-myCAF-mediated immunosuppression on CD8⁺ T lymphocyte cytotoxicity. Interestingly, YAP1 silencing (Supplementary Figure S7B for siRNA efficacy) in ECM-myCAF reversed ECM-myCAF-mediated effects on cytotoxic CD8⁺ T cells, while it had no impact in Detox-iCAF (Fig. 6A-E). Finally, we tested if these CAF-S1 clusters could modulate CD8⁺ T cell migration by performing transwell assay (Fig. 6F). Interestingly, Detox-iCAF enhanced CD8⁺ T cell

migration, while ECM-myCAF fibroblasts did not (Fig. 6F). Altogether, these observations show complementary roles of the two CAF-S1 clusters on CD8 T lymphocytes: while Detox-iCAF attracts CD8⁺ T lymphocytes, ECM-myCAF dampen their cytotoxic identity and functions through a YAP-1 dependent mechanism. “

p44: Corresponding legend of Fig. 6: “(A) **Up**, Representative flow cytometry plots showing CD8 and PD-1 protein levels in control condition (CD8⁺ T cells alone) (-) or co-cultured with Detox-iCAF or ECM-myCAF primary fibroblasts transfected with an untargeted siRNA (siCtrl) or with two different siRNA targeting YAP1 (siYAP1(1), siYAP1(2)). The population of interest (CD8⁺ PD-1⁺) is represented in red and the isotype control in black. **Bottom**, Bar plots showing the % of PD-1⁺ T cells among CD8⁺ T lymphocytes alone or in presence of Detox-iCAF or ECM-myCAF (**Left**) and transfected with siCtrl or siYAP1 (**Middle and Right**). Data are mean ± SEM (n = 5). P-values from paired Wilcoxon test. (B-D) Same as (A) for Granzyme B⁺ (B), Perforin⁺ (C) and IFN-γ⁺ (D) CD8⁺ T lymphocytes. P-values from paired Student t-test. (E) **Left**, Representative flow cytometry plots showing CAOV3 cell death after 24h of incubation with CD8⁺ T lymphocytes pre-incubated with Detox-iCAF or ECM-myCAF primary fibroblasts transfected with an untargeted siRNA (siCtrl) or with two different siRNA targeting YAP1 (siYAP1(1), siYAP1(2)). **Right**, Bar plots showing the % of cancer cell death after incubation with CD8⁺ T lymphocytes. Data are mean ± SEM (n = 5). P-values from paired Student t-test. (F) Bar plots showing the % of migration of CD8⁺ T lymphocytes after 24h of transwell co-culture with Detox-iCAF or ECM-myCAF primary fibroblasts transfected with an untargeted siRNA (siCtrl) or with two different siRNA targeting YAP1 (siYAP1(1), siYAP1(2)). Data are mean ± SEM (n = 8). P-values from unpaired Student t-test.”

Negative correlation in Fig. 2F indicates the inverse relationship between the decrease in CAF-S1 content and the increase in CD8⁺ T cell density after chemotherapy. This analysis of paired HGSOc patients shows that when CAF-S1 content increases in patients after treatment, CD8⁺ T cell density decreases. Reciprocally, the more CAF-S1 content decreases upon chemotherapy, the more CD8⁺ T lymphocytes are detected. Our data in Fig. 2F and **Supplementary 2** thus indicate that the variation of the CAF-S1 content upon chemotherapy is associated with T cell density. These data are thus important observations in these paired patients, and we have thus tried to clarify this point in the new version of the Text, **p8-9**.

p8-9: Results: “We next wondered whether TIL density after chemotherapy could be linked to the extent of CAF-S1 or CAF-S4 decrease. To compare the variations of each cellular population after *versus* before chemotherapy in each patient, we established a delta score (Δ) calculated as followed: content of the studied population after chemotherapy *minus* (-) content of the same population before chemotherapy (see also Methods *#Establishment of a delta-score measuring variations of each population by chemotherapy*). We analyzed the variations of CD8⁺ TILs (assessed by the Δ -number of CD8⁺ TILs per surface unit) and CAF-S1 (evaluated by the Δ -score of FAP, a CAF-S1 specific marker) in paired HGSOc patients (Fig. 2E, F and **Supplementary Fig. S2C**). Interestingly, this analysis showed an anti-correlation between CD8⁺ TILs and CAF-S1, suggesting that the more CAF-S1 decrease after chemotherapy, the more CD8⁺ TIL density increases in the tumor and thus highlighting the importance of the extent of CAF-S1 variation upon treatment. In contrast, we found no association between the proportion of TILs and the overall decrease in myofibroblastic CAF populations, *i.e* when considering both CAF-S1 and CAF-S4 together (evaluated by the Δ -score of the SMA marker, a common marker of these two myofibroblastic populations) (**Supplementary Fig. S2D, E**). This result showed that the increase in CD8⁺ TIL density after treatment is specifically anti-correlated with the CAF-S1 content in HGSOc.”

As requested, in the prospective cohort of patients (unpaired patients), we now provide the correlations in treatment naïve and after treatment, separately (Fig. 2I and Fig. 2F, **Left and Middle**). Moreover, in Fig. 4G, H, we separated samples before and after chemotherapy, as done in Fig. 2A-D, as requested by the Reviewer. In Fig. 4, we analyzed the same retrospective cohort of paired patients as in Fig. 2. Thus, we provided data on T cell density and CAF-S1 content in Fig. 2 and clues on potential molecular mechanism in Fig. 4 based on the analysis of the same cohort. As we found (thanks to the sensitivity of flow cytometry analysis) that FAP^{High} CAF-S1 might be more involved in re-infiltration of CD8⁺ T cells after chemotherapy than FAP^{Low-Med} CAF-S1, we analyzed the marker ANTXR1 that we identified recently for this population (Kieffer, *Cancer Discovery*, 2020). Regarding quantifications, they have been performed in close collaboration with Pathologists. Fresh samples from the prospective cohorts were collected at time of surgery either from treatment-naïve or from chemotherapy-treated HGSOc patients, as part of routine standard of care. Surgeons and pathologists provided all samples dedicated to these analyses, and there was no further selection of specific areas by the biologists. After having been collected from pathologists, fresh tissues were digested, cells stained with antibody mix and analyzed. The same gating strategy was used for all patients. Thus, cell population selection in flow cytometry was consistent between samples to achieve accurate comparative analysis.

4) YAP1 and CYR61

In the 4th results section, authors conclude that the decrease of nuclear YAP1 in the stroma is associated with a reduction in ECM-myCAFs. Yet, the 5th section starts by claiming that YAP1 levels decrease significantly in ECM-

myCAFs following chemotherapy. While the latter statement contradicts the former, data depicted in Figure 4 does not show decrease of YAP1 in ECM-myCAFs. Instead, this effect is calculated by H-scores over the total stromal content. Therefore, it is not clear whether YAP1 reduction is following the loss of these CAFs or if it is being also decreased in the remaining CAFs. Authors should provide with immunofluorescence pictures depicting colocalization of YAP1 and ANTXR1 in CAF-, and quantify ANTXR1+YAP1+ and ANTXR1+YAP1- CAFs before and after chemotherapy.

Also, what is the significance of YAP1 reduction in tumor epithelial cells seen in Figure 4J? Since the authors demonstrate that such reduction does not affect T cell activation in Figure 5B, could it have any other effect that has not been investigated? From the picture (left, magnification) displayed in Figure 4I, nuclear YAP negative cells counted as epithelial cells could be intraepithelial immune cells instead. Please, perform this analysis taking into account the potential presence of YAP negative intraepithelial immune cells.

Finally, concluding that CYR61 has a “key” role is too strong in view of the results from the only one functional experiment (Figure 6G). Either authors should provide with more functional evidence or lighten this statement.

As requested, we have now analyzed two series of serial IHC sections, one from sample before chemotherapy and the other from sample after chemotherapy, both stained for ANTXR1 and YAP1. Following alignment of the sections, we isolated and segmented the stromal region, enabling us to quantify the levels of both ANTXR1 and YAP1 in each stromal cells. Through this analysis, we were thus able to quantify the level of YAP1 in ANTXR1+ CAF and compare this level before and after chemotherapy. Our results revealed that after chemotherapy, YAP1 staining decreases with loss of ANTXR1+ CAF and is also reduced in residual ANTXR1+ CAF. These new data have now been included in the new version of the manuscript **p13**, in the new **Fig. 4** and in the **Methods** section **p30**. Regarding YAP1 staining in epithelial cells, Pathologists have defined them as real staining in epithelial cells and have been able to differentiate YAP1- epithelial cells from YAP1- intraepithelial immune cells. Finally, as requested, we have now performed additional functional assays to provide more evidence about the role of CYR61 in the immunosuppressive activity of ECM-myCAF. We demonstrate that CD8+ T lymphocytes, which have been co-cultured with ECM-myCAF silenced for CYR61 restored cytotoxic activity with an increase secretion of Granzyme, Perforin and IFN γ , and a decrease PD-1+ CD8+ T cells. Importantly, these CD8+ T cells showed also an increase capacity to kill cancer cells. These new functional assays are shown in the **Supplementary Fig. 7C-F** and described **p15**, with a conclusion which has been lightened, as requested by the Reviewer.

p13: Results: “As YAP1 localization in the nucleus is key for its interaction with TEAD transcription factors, we next analyzed YAP1 accumulation in nuclei and found that YAP1 nuclear staining was significantly reduced after chemotherapy in stroma and in cancer cells (**Fig 4I, J**). Moreover, the decrease in nuclear YAP1 staining in stroma following chemotherapy was correlated with the one of FAP and ANTXR1 (**Fig 4K, L**), but not with SMA (**Supplementary Figure S5E**). These observations suggested that the reduced expression of YAP1 and TEAD-target genes after chemotherapy was concomitant to the decrease in the content of ANTXR1+ CAF, marker of ECM-myCAF, suggesting that the loss of these CAF after treatment may explain -at least in part- the reduced YAP1/TEAD-signaling pathway. In addition, we also considered that chemotherapy might also reduce YAP1 protein levels in the residual ANTXR1+ CAF after treatment. To test this hypothesis, we analyzed the colocalization of YAP1 and ANTXR1 staining in CAF before and after chemotherapy (**Fig. 4M**). We found that the residual ANTXR1+ CAF after treatment displayed a decreased YAP1 staining compared to ANTXR1+ CAF before treatment (**Fig. 4M, N**). Indeed, while the majority of ANTXR1+ CAF were YAP1+ before treatment, the residual ANTXR1+ CAF were predominantly YAP1^{-Low} (**Fig. 4O**). Taken as a whole, our results indicate that the reduction of YAP1 staining after chemotherapy was mainly due to the concomitant loss of the ECM-myCAF population, on the one hand, but also to the reduction of YAP1 staining in these residual CAF-S1 on the other.”

p27: Methods: “**#In silico analysis of colocalization of IHC staining on serial sections.** Regions of interest from either serial IHC stained for ANTXR1 and YAP1 or serial IHC stained for both ANTXR1/CD8 and YAP1 were registered using the Fiji plugin BUnwarpJ by manually identifying common features. Registered images were imported in QuPath and epithelial regions were masked by training a pixel classification model as implemented in QuPath. The stromal region was then segmented using Cell Detection on the Hematoxylin channel with default parameters. After quantifying the DAB staining intensity in the first image (YAP1), the cell detections were then transferred to the other registered image to quantify -in the same cell- the subsequent staining using the DAB channel (for ANTXR1), or DAB and Fast Red for ANTXR1 and CD8 co-staining respectively. Measurements were then exported and analyzed in R to plot YAP1 intensity in ANTXR1+ CAF and the proportion of YAP1^{+/-} ANTXR1+ CAF, or exported in python for colocalization analysis. Cell mean DAB staining intensity of ANTXR1 and YAP1 were used to separate the segmented cells into YAP1^{+/-} and ANTXR1^{+/-} cells. Threshold for positivity was assessed using the Single Measurement classifier tool in QuPath. Co-occurrence probability of clusters was computed using the function `gr.co-occurrence` from the Squidpy package¹⁰⁴ and plotted with the function `pl.co-occurrence` according to the CD8+ T cell cluster, other cell types than the one of interest were removed from the plot after computing the co-occurrence probability ratio for a simplified presentation of the results. Neighborhood enrichment analysis was conducted after building a connectivity matrix with the function `gr.spatial_neighbors`,

computed with `gr.nhood_enrichment` and plotted with `pl.nhood_enrichment` from the Squidpy package, with default parameters. Merged staining of ANTXR1 and YAP1 was performed by deconvoluting the DAB staining of the registered region of interest (ROI) with the Color deconvolution method integrated in Fiji¹⁴⁷. ANTXR1 and YAP1 were colored in red and green, respectively, and the lookup table was inverted in Fiji to highlight the staining. The images were then merge.”

p15: Results: “Among well-established YAP1-TEAD-target genes, CYR61 (Cysteine-rich angiogenic inducer 61, also called CCN1) was highly expressed by CAF-S1, and more specifically by the ECM-myCAF cluster (**Supplementary Fig. S7C**). As this gene encodes a secreted protein, we considered that CYR61 could be instrumental for ECM-myCAF to act on CD8⁺ T lymphocytes. We thus silenced CYR61 in ECM-myCAF primary fibroblasts (**Supplementary Fig. S7D** for siRNA efficacy) and found that CYR61 silencing in ECM-myCAF restored the percentages of granzyme B⁺, perforin⁺ and IFN- γ ⁺ CD8⁺ T lymphocytes and reduced the proportion of PD-1⁺ CD8⁺ T cells (**Supplementary Fig. S7E**). Consistent with these observations, CD8⁺ T lymphocytes, which have been co-cultured with ECM-myCAF silenced for CYR61, showed an increased capacity to kill cancer cells (**Supplementary Fig. S7F**), suggesting that the secreted factor CYR61, a well-known YAP1-target gene, might mediate ECM-myCAF immunosuppressive activity on CD8⁺ T cells. Taken as a whole, these data suggest that the down-regulation of YAP1 and downstream TEAD-signaling pathway observed in ECM-myCAF after chemotherapy might be key in CD8⁺ T cell enrichment in HGSOC patients.”

Additional comments

-Data listed in Data availability statement are not available. Please provide updated links allowing reviewer access.

As requested, private links with data from the study are now available for the Reviewer:

- scRNAseq data: <https://figshare.com/s/6064a9aa9123c30b78ea>
- Spatial transcriptomic data: <https://figshare.com/s/ab1e9b598339713a63a9>.

Final DOI links indicated in the “Data availability” section will be active for readers after complete acceptance of the paper.

p20: “Data availability: Raw single cell sequencing data, spatial barcode locations and related counts matrices are available from the European Genome-Phenome Archive platform (<https://ega-archive.org>) under accession EGAS00001007032 and EGAS00001007031, respectively. Processed scRNAseq data are available on Figshare under the DOI: 10.6084/m9.figshare.22147166. Processed spatial transcriptomic data are available on Figshare under the DOI: 10.6084/m9.figshare.22147103. Count data from scRNAseq dataset from HGSOC before and after chemotherapy was retrieved from Gene Expression Omnibus with accession code GSE165897⁹⁹. Publicly available scRNAseq dataset from breast cancer before and after chemotherapy were retrieved from <https://lambrechtslab.sites.vib.be/en/single-cell>¹⁰⁰. Publicly available scRNAseq dataset from treatment-naïve HGSOC patients were recovered from <https://lambrechtslab.sites.vib.be/en/high-grade-serous-tubo-ovarian-cancer-refined-single-cell-rna-sequencing-specific-cell-subtypes>⁴⁵.”

p20: “Code availability: Codes used for this study are available on Figshare under the DOI: 10.6084/m9.figshare.24271369.”

-Authors should delimitate the stroma and epithelium in all IHC pictures from Figure 1. In particular, it would help to add representative stromal percentages for Figures 1A and 1B.

As requested, we have now added indications on stromal and epithelial areas in all IHC pictures, as well as in the spatial transcriptomic data.

-Since serial sections for FAP, CD29, SMA and FSP1 stainings are used, authors should provide with images for every marker in same tumor regions as a representation of the decision tree depicted in Figure 1H (for one CAF-S subset in Figure 1F). In supplementary Figures, please provide these representative stainings for the three other CAF-S subsets.

We have now provided additive images for every markers from the same tumor regions in **Fig. 1F**. Representative staining of the other CAF subsets have already been shown in previous publications (Costa, *Cancer Cell*, 2018; Givel, *Nat Commun*, 2018; Pelon, *Nat Commun*, 2020; Bonneau, *Breast Cancer Res.*, 2020). Our publication is now focused on CAF-S1 and its clusters, their variations upon treatment and their relationships with CD8⁺ T cell density. We have thus dedicated the time of this Review to provide more data and analyses on these specific populations, providing by this way an innovative and deeper study of the immunosuppressive CAF-S1 population and its related clusters.

-Authors should provide with representative population percentages in all their flow cytometry analyses.

We have now provided representative population percentages in flow cytometry analyses, as requested.

-In Figure 1F, CD29 and FSP1 seem to be staining both stromal and epithelial cells, and a reduced expression can be observed in both compartments after chemotherapy. What is the significance of CD29 and FSP1 expression in the epithelium?

We do agree with the Reviewer. Indeed, CD29/Integrin β 1 and FSP1 stained stromal cells but also epithelial cells. We have previously studied and shown these staining in our most recent studies, in which we characterized these markers and analyzed CAF populations in details (Costa, *Cancer Cell*, 2018; Givel, *Nat Commun*, 2018; Pelon, *Nat Commun*, 2020; Bonneau, *Breast Cancer Res.*, 2020). Our current study is dedicated to the stromal compartment. We hope the Reviewer and the Editor will agree that studying the role of CAF marker in epithelial cells is beyond the scope of our study.

-Figure 1F is a composite of screenshots (The upper right grey square gives it away). Please provide with higher quality images.

We are sorry for that. We have now provided higher quality images. We thank the Reviewer for the careful reading of our paper.

-Authors mention the existence of “platinum resistant patients with high residual CAF-S1 or CAF-S4 content after chemotherapy”, which is also mentioned throughout the discussion. To make this claim, please provide with GSEA data comparing responder patients versus non-responders to test whether CAF-S1/4 are indeed enriched in non-responders.

As we observed the importance of maintenance of CAF-S1/CAF-S4 populations after chemotherapy in platinum resistant patients (**Supplementary Fig. 1A**), we first tested if the impact of the content in CAF populations was independent of the residual content in cancer cells after treatment by applying multivariate Cox regression analysis. The hazard ratio for overall survival (OS) and disease-free survival (DFS) was 5.9 (95%CI [1.1-31.3], $p=0.038$) and 4.9 (95%CI [1.1-21.6], $p=0.037$), respectively, in patients enriched in CAF-S1 or CAF-S4 compared to those enriched in CAF-S2 or CAF-S3, after considering the level of residual epithelial content following treatment in multivariate analysis. By this way, we validated that the impact observed with CAF populations is independent of the residual epithelial content after treatment. This multivariate analysis is now included in the **Supplementary Fig. S1B** and described **p7** and in the corresponding legend, **p47**.

p7: Results: “Indeed, platinum-resistant patients with high residual CAF-S1 or CAF-S4 content after chemotherapy survived less and relapsed earlier than patients enriched in CAF-S2 or CAF-S3 after treatment (P-value = 0.01 for overall survival and P-value = 0.018 for disease-free survival by Kaplan-Meier test) (**Supplementary Fig. S1A**). Importantly, multivariate Cox regression analysis showed that this effect was independent on the level of residual epithelial content after treatment (**Supplementary Fig. S1B**), thereby highlighting the interest of variations of these CAF populations after treatment.”

p47: Corresponding legend of Sup Fig. S1B: “(B) Multivariate Cox regression analysis for OS (left) and DFS (right) considering enrichment of CAF populations and residual epithelium content after chemotherapy.”

Moreover, as we did not have access to RNAseq data from the Retrospective Curie 1 cohort, we studied an independent cohort of HGSOc patients for whom we had access to a long-term clinical follow up and performed bulk RNA sequencing. As we observed a switch in the composition of specific CAF-S1 clusters in HGSOc after chemotherapy (with a significant reduction of the ECM-myCAF cluster and an increase in the Detox-iCAF cluster, **Fig. 3**), we sought to test if this switch could be associated with response to treatment. By using a deconvolution method to assess the composition of CAF-S1 clusters in each patient before and after treatment, we validated the decrease in ECM-myCAF content and the increase of Detox-iCAF after treatment in this cohort. Interestingly, we found a significant association between the content in these clusters after chemotherapy and the chemotherapy response. Indeed, we observed a lower proportion of ECM-myCAF and a higher proportion of Detox-iCAF after chemotherapy in responder patients compared to non-responder patients ($p = 0.024$ and $p = 0.015$, respectively). We have now integrated these results in **New Fig. 3K, L** and in the new version of the manuscript, **p10-11**.

p10-11: Results: “We then sought to validate these findings by studying bulk RNA-seq data from an independent cohort of ovarian cancer patients (containing 45 samples before treatment and 25 after chemotherapy, see **Table 1**, retrospective SCANDARE Curie 2 cohort). For this cohort of patients, long-term clinical follow-up and information on treatment responses of patients were available. Cellular composition of each sample -before and after chemotherapy- was inferred by using BayesPrism method¹⁰² based on a high-resolution HGSOc cellular atlas that we built and annotated from both Curie and publicly available scRNAseq datasets (**Supplementary Fig. S4A, B**). This HGSOc atlas was composed of 49 909 cells and 24 different cell types and states, including CAF-S1 clusters, thereby constituting a comprehensive HGSOc cellular landscape (**Supplementary Fig. S4A, B**). In this independent cohort, we confirmed that the proportion of CD8⁺ T lymphocytes increased after treatment (**Fig. 3J**). Importantly, we also validated the decrease in ECM-myCAF content and increase in Detox-iCAF after chemotherapy (**Fig. 3K**). Moreover, thanks to the long-term clinical follow-up available in the SCANDARE cohort, we were able to highlight that the decrease in the ECM-myCAF content and the increase in the Detox-

iCAF proportion following chemotherapy were mainly detected in responder patients but not in non-responders (**Fig. 3L** and **Supplementary Fig. S4C**). Moreover, the increase of CD8 T cell density was also detected in responder patients (**Fig. 3M**), showing that variations of these CAF-S1 clusters are concomitant to increased CD8+ T cell content after chemotherapy. Taken as a whole, these data confirmed the clinical interest of the variations of these CAF-S1 clusters upon treatment.”

p43: Corresponding Legend Fig. 3J-M: “(J) Proportion of CD8+ T lymphocytes in the retrospective SCANDARE Curie 2 cohort before (treatment-naïve) and after chemotherapy (N = 70 samples: 45 treatment-naïve, 25 after chemotherapy). P-value from Mann-Whitney test. (K) Same as (J) for the proportion of ECM-myCAF (**Left**) and Detox-iCAF (**Right**) clusters among CAF-S1. (L, M) Same as (J, K) according to response to chemotherapy for ECM-myCAF (**L, Left**), Detox-iCAF (**L, Right**) and CD8+ T cells (**M**). P-values from Mann-Whitney test.”

-According to the methods section, five representative areas (0.105 mm²) per slide were used for T cell quantification. What is the stromal/epithelial content of said areas? How representative are they of the intratumoral variability?

As mentioned in the text, the 5 areas selected for T cell quantification by Pathologists are representative of each slide. We validated that the stromal/epithelial content was representative of the stromal/epithelial proportion assessed for each patient.

-Please provide with percentages in representative images from Figure 2A. Also, please point the magnified region depicted in the lower-right miniatures.

We have now included the percentages of immune cells in stroma and epithelium in representative images and grey squares to point the magnified regions depicted in the insets in **Fig. 2A**, as requested.

-In Figure 2G it seems that samples after chemotherapy include much more counted events. For the sake of comparison, authors should limit to a same number of total T cells in their analyses.

As requested, we have used the same number of events in our analysis in **Fig. 2G**.

-How can authors explain the lack of correlation between T cells or YAP1 and aSMA? In theory, since aSMA stains both CAF-S1 and CAF-S4 populations, the supposed correlation between CAF-S1 and T cells or YAP1 should suffice to see a similar, but probably milder, effect when using aSMA.

SMA is a marker of both CAF-S1 and CAF-S4 populations. Evaluation of SMA provides information on the proportion of both CAF-S1 and CAF-S4. As shown in our previous publications, T cell content is modulated by CAF-S1 but not CAF-S4; and TEAD-target genes are highly expressed in CAF-S1 compared to CAF-S4. Detecting both CAF-S1 and CAF-S4 with SMA prevents us to detect the correlations strictly observed with CAF-S1 (stained by FAP) in **Fig. 2** and **Fig. 4**.

-It is interesting to note that a large proportion of samples in Figure 2F tend to focus around a delta H-score of 0 and above, which would imply that many tumors do not only experience no CAF alterations, but FAP can be increased after chemotherapy. Can this effect be connected with overall responses to chemotherapy?

In **Fig. 2F**, the Delta score of FAP varies from -400 to +400, indicating an important variation upon chemotherapy and a potential link with response to treatment. For the Retrospective Curie 1 cohort (analyzed in **Fig. 1A-I** and **Fig. 2A-F**), this question is addressed in **Supp Fig. 1A, B**. Indeed, platinum-resistant patients with high residual CAF-S1 or CAF-S4 content after chemotherapy survived less and relapsed earlier than patients enriched in CAF-S2 or CAF-S3 after treatment (P-value = 0.01 for overall survival and P-value = 0.018 for disease-free survival by Kaplan-Meier test) (**Supplementary Fig. S1A**). Importantly, multivariate Cox regression analysis showed that this effect was independent on the level of residual epithelial content after treatment (**Supplementary Fig. S1B**), thereby highlighting the interest of variations of these CAF populations after treatment.

In addition, as we observed a switch in the composition of specific CAF-S1 clusters in HGSOc after chemotherapy (with a significant reduction of the ECM-myCAF cluster and an increase in the Detox-iCAF cluster, **Fig. 3**), we sought to test if this switch could be associated with response to treatment. To do so, we have now analyzed an independent cohort (SCANDARE cohort of HGSOc patients) for whom we had access to a long-term clinical follow-up and performed bulk RNA sequencing. By using a deconvolution method to assess the composition of CAF-S1 clusters in each patient, we validated the decrease of ECM-myCAF and the increase of Detox-iCAF after treatment in this cohort. Interestingly, we found a significant association between content in these clusters and chemotherapy response. Indeed, we observed a lower proportion of ECM-myCAF and a higher proportion of Detox-iCAF after chemotherapy in responder patients compared to non-responder patients (p = 0.024 and p = 0.015, respectively). We have now integrated these results in **New Fig. 3K, L** and in the new version of the manuscript, **p11**.

-Since all cells from the scRNAseq analysis from Figure 3F are CAF-S1, what is the relative percentage of this subpopulation within the whole tumor microenvironment, as detected by flow cytometry?

The relative percentage of the CAF-S1 population among total viable cells, including tumor cells and all cells from tumor micro-environment, is around 5-10%.

-What does Residual or Relapse mean in Figures 4C and D, respectively? If the sample after chemotherapy is that of a relapsing patient, we could conclude that they are resistant to chemotherapy. Therefore, it follows that CAF-S1 enrichment does not determine resistance to chemotherapy.

Regarding relapse / residual, we do agree with the Reviewer, and we are sorry for this error. In our paper, we analyzed residual cells after treatment, and not at relapse. The text has now been changed, and we thank the Reviewer for the careful reading of our paper.

-In Figure 5C, while there are clear regions devoid of ECM-myCAFs with increased CD8 intensity, it is not clear whether they are completely excluded from ECM-myCAFs. Could authors focus on particular areas of interest? Also, and since this is the post-chemotherapy sample, would this effect also be observed in the pre-chemotherapy sample? Authors should also include the map for YAP1 expression to reinforce their claim of CD8 T cells being excluded from YAP1+ CAFs.

As described above, we have now provided multiple additional data, including spatial transcriptomics and multiplex IHC in **Fig. 5** to address that question.

-FACS gating strategies in Supplementary Figure 4E: Authors should provide with a control of CAF-S4 to demonstrate that FAP and CD29 levels are truly concordant with those of CAF-S1.

We have previously studied and validated our markers in previous studies, in which we characterized these markers and analyzed CAF populations in details (Costa, *Cancer Cell*, 2018; Givel, *Nat Commun*, 2018; Pelon, *Nat Commun*, 2020; Bonneau, *Breast Cancer Res.*, 2020). We have added all these references in our paper to inform the Reader.

Reviewer #4 (Remarks to the Author): with expertise in cancer associated fibroblasts, transcriptomics

In this study the authors present a comprehensive analysis of the role of the stromal compartment in responding to chemotherapy and T cell infiltration. Notably, the authors use patient samples, single cell and spatial methods to infer a subset of CAF-S1, ECM-myCAF coupled with inhibition of YAP1 signaling pathway as a better therapeutic strategy for HGSOC. Particularly, restraining the expression of YAP1 target gene, CYR61 restored the percentage of Gzmb+ CD8+ T lymphocytes. While the study has significant impact and scope, it will benefit from addressing the following comments.

We thank the Reviewer for the positive evaluation of our work. We have now addressed Reviewer's comments, which have improved our manuscript. We thank the Reviewer for his/her recommendations.

Comments:

1. Figure panels 3F-H with single cell classification of various CAF clusters are not distinct cell types, rather appear as artificial clusters. In order to distinguish which CAF related program is being up regulated, I recommend using topic modeling methods such as cNMF (Kotliar et al., *eLife* 2019) to assess changes in gene expression programs pre- and post- treatment and clearly demonstrate the behaviour of the ECM my CAF program.

As requested by the reviewer, and in addition to the reference-based approach used in our paper, we have now applied the consensus Non-Negative Matrix factorization (cNMF) method (Kotliar et al., *eLife* 2019) on scRNAseq data from the Curie cohort to validate the CAF-S1 identity in an unsupervised manner. The results have been added to **Supplementary Fig. 3D** demonstrating that the factors identified by cNMF coincide both in terms of classification and biological significance with our CAF-S1. We thank the Reviewer for this suggestion, which confirmed our data and improved our manuscript.

p10, Results: “We also confirmed the identity of the most abundant CAF-S1 clusters detected (ECM-myCAF, Detox-iCAF and Wound-myCAF) using an unsupervised method by applying consensus Non-Negative Matrix factorization (cNMF)¹⁰¹ on scRNAseq data from the Curie cohort (**Supplementary Fig. S3D**).”

p29-30: Methods: “**Consensus Non-Negative Matrix factorization:** Identification of gene expression programs from scRNAseq data was performed using consensus Non-Negative Matrix factorization (cNMF) algorithm described in¹⁰¹ and implemented in Python (<https://github.com/dylkot/cNMF>). Range from 5 to 13 factors (K) were tested with 200 iterations for each K. Consensus estimate was obtained by setting optimal K to 10 considering the trade-off between stability and error as described by the authors. Local-density-threshold was fixed to 0.1 using

diagnostic plot, default parameters were used otherwise. Heatmap and clustering (correlation distance and Ward.D2 method) was applied on the usage matrix (cells x K) and pathway analysis was performed using Metascape tool on the first 200 most contributing genes for each factor from the gene expression program matrix.”

p48: Corresponding Legend Supplementary Fig. 3D: “(D) Heatmap showing the relative contribution of the 10 factors obtained after consensus Non-Negative Matrix factorization (cNMF) of CAF-S1 fibroblasts from HGSOc Curie cohort (Prospective cohort 2). Clustering uses correlation distance and Ward D2 agglomeration method. Bars on top of the heatmap represent CAF-S1 cluster identity (colors) and treatment status (grey/black). Representative biological pathways from the top 200 most informative genes of factors 1, 2, 3 and 6 are indicated on the right side for the most abundant CAF-S1 clusters analyzed in this study.”

2. Can the authors comment on the ANTXR1- iCAF sub population of CAF-S1 cluster in relation to chemotherapy? Are consistent levels of these sub populations associated with positive or negative outcomes in HGSOc and other fibrotic solid cancers?

We thank the Reviewer for this recommendation that we have now addressed. As we observed a switch in the composition of specific CAF-S1 clusters in HGSOc after chemotherapy (with a significant reduction of the ECM-myCAF cluster and an increase in the ANTXR1- iCAF cluster, referred to as Detox-iCAF, **Fig. 3**), we sought to test if this switch could be associated with response to treatment. To do so, we now analyzed an independent cohort of HGSOc patients for we had access to a long-term clinical follow-up and performed bulk RNA sequencing. By using a deconvolution method to assess the composition of CAF-S1 clusters in each patient, we validated the decrease of ECM-myCAF and the increase of Detox-iCAF after treatment in this cohort. Interestingly, we found a significant association between content in these clusters and chemotherapy response. Indeed, we observed a lower proportion of ECM-myCAF and a higher proportion of Detox-iCAF after chemotherapy in responder patients compared to non-responder patients ($p = 0.024$ and $p = 0.015$, respectively). We have now integrated these results in **New Fig. 3K, L** and in the new version of the manuscript, **p10-11**.

p10-11: Results: “We then sought to validate these findings by studying bulk RNA-seq data from an independent cohort of ovarian cancer patients (containing 45 samples before treatment and 25 after chemotherapy, see **Table 1**, retrospective SCANDARE Curie 2 cohort). For this cohort of patients, long-term clinical follow-up and information on treatment responses of patients were available. Cellular composition of each sample -before and after chemotherapy- was inferred by using BayesPrism method¹⁰² based on a high-resolution HGSOc cellular atlas that we built and annotated from both Curie and publicly available scRNAseq datasets (**Supplementary Fig. S4A, B**). This HGSOc atlas was composed of 49 909 cells and 24 different cell types and states, including CAF-S1 clusters, thereby constituting a comprehensive HGSOc cellular landscape (**Supplementary Fig. S4A, B**). In this independent cohort, we confirmed that the proportion of CD8⁺ T lymphocytes increased after treatment (**Fig. 3J**). Importantly, we also validated the decrease in ECM-myCAF content and increase in Detox-iCAF after chemotherapy (**Fig. 3K**). Moreover, thanks to the long-term clinical follow-up available in the SCANDARE cohort, we were able to highlight that the decrease in the ECM-myCAF content and the increase in the Detox-iCAF proportion following chemotherapy were mainly detected in responder patients but not in non-responders (**Fig. 3L** and **Supplementary Fig. S4C**). Moreover, the increase of CD8 T cell density was also detected in responder patients (**Fig. 3M**), showing that variations of these CAF-S1 clusters are concomitant to increased CD8⁺ T cell content after chemotherapy. Taken as a whole, these data confirmed the clinical interest of the variations of these CAF-S1 clusters upon treatment.”

p43: Corresponding Legend Fig. 3J-M: “(J) Proportion of CD8⁺ T lymphocytes in the retrospective SCANDARE Curie 2 cohort before (treatment-naïve) and after chemotherapy (N = 70 samples: 45 treatment-naïve, 25 after chemotherapy). P-value from Mann-Whitney test. (K) Same as (J) for the proportion of ECM-myCAF (**Left**) and Detox-iCAF (**Right**) clusters among CAF-S1. (L, M) Same as (J, K) according to response to chemotherapy for ECM-myCAF (**L, Left**), Detox-iCAF (**L, Right**) and CD8⁺ T cells (**M**). P-values from Mann-Whitney test.”

REVIEWERS' COMMENTS

Reviewer #1 (Remarks to the Author):

Overall, the authors have significantly improved the manuscript by adding a considerable amount of new data, as well as by clarifying the description of their approach and the results. They have addressed all my major concerns, and I only have minor concerns, as listed below.

1. Summary section: "Thus, efficient inhibition after treatment of YAP1-signaling pathway in the ECM-myCAF cluster is required to enhance CD8+ T-cell cytotoxicity"

Please modify this statement; change 'required to enhance' to eg. 'could enhance'. 'Require' would mean that there are no other ways to enhance CD9+ T cell cytotoxicity, which is basically impossible to prove.

2. Results section: "We also took advantage of the publicly available scRNAseq data from CAF isolated from untreated and chemotherapy-treated samples from an independent HGSOc cohort of patients treated at Helsinki's hospital 99..."

Even though not apparent from ref 99, these patients were actually treated at Turku University Hospital, Finland, so the cohort should for example "Turku cohort", and the sentence should state "...treated at Turku University Hospital, Finland,...".

3. There is a spatial transcriptomics data set from post-chemo HGSOc available from this preprint:

Denisenko, Elena, Leanne de Kock, Adeline Tan, Aaron B. Beasley, Maria Beilin, Matthew E. Jones, Rui Hou, et al. 2022. "Spatial Transcriptomics Reveals Ovarian Cancer Subclones with Distinct Tumour Microenvironments." bioRxiv. <https://doi.org/10.1101/2022.08.29.505206>. The GeneExpression Omnibus (GEO) accession code is GSE211956. Nevertheless, the authors analyzed a considerable number of additional samples from their own cohort so that there is not absolute need to use this published data set for additional analyses.

4. Please add a reference for the DoRothEA algorithm.

5. Fig 5:H, J, N: CD8+ T lymphocytes co-occurrence probability ratio with CD8+ T lymphocytes, ANTXR1+ CAF and ANTXR1- CAF are shown from two samples; please also show the same plot across all the Visium analyzed samples; or a composite figure showing median and 95% interval across the samples.

6. Discussion: "In addition, we highlight the YAP1-mediated impact of the ECM-myCAF cluster on CD8+ T cell content following chemotherapy."

=> Impact suggests functional evidence; however, functional validation id for CD8 cytotoxicity, while there is associative evidence for the content or infiltration. Please rephrase.

74. Please rephrase the following sentences; their language needs improvement:

"confirming observations obtained by IHC and flow cytometry and validating that samples selected for spatial transcriptomic as representative"

"In these sections, we identified the different CAF-S1 clusters and confirmed that, among them, the proportion of the ANTXR1+ ECM-myCAF cluster was this one, which accumulated before treatment and decreased the most after chemotherapy"

"We already observed that several TEAD-target genes were up-regulated in CAF-S1 compared to CAF-S4 in HGSOc 40."

"Expression of some YAP1/TEAD-target genes was highly detected in CAF-S1 fibroblasts"

"suggesting that the loss of these CAF after treatment may explain -at least in part- the reduced YAP1/TEAD-signaling pathway."

Reviewer #2 (Remarks to the Author):

The authors have done a significant amount of additional work to address many of the previous reviewer comments and the manuscript is greatly improved. I still have one comment, which was mentioned in my previous review but perhaps was not clear. This is that, since the authors show quite clearly that ECM-myCAFs and CD8+ T cells are spatially segregated within patient tumors, the rationale for experiments in which these two cell types are co-cultured is difficult to understand. Since these two cell types don't exist in close proximity to each other in vivo, the results seen when they are forced into proximity in an artificial in vitro assay may not be physiologically relevant. Perhaps there can be some statement added to the discussion to address this contradiction.

An additional comment is related to Supplemental Figure 6 - Characterization of detox-iCAF vs ECM-myCAF. Figure S6C seems to show differential expression between CAFs grown under two different culture conditions, not two different isolated CAF subtypes. Please clarify - how were the two cell types generated and cultured? Why is the gene expression focused around cells grown under different culture conditions?

Reviewer #3 (Remarks to the Author):

I would like to congratulate the authors for the work they have done in addressing this reviewer's concerns. The inclusion of new data, especially the functional assays and the addition of spatially resolved samples, significantly enhances the strength and clarity of the manuscript's message. I fully support the acceptance of this study for publication.

POINT BY POINT RESPONSES TO REVIEWERS' COMMENTS

We would like to thank the Reviewers for their positive assessment of our revised work and acceptance in principle of our manuscript (23-09662A) for publication in *Nature Communications*. We are delighted that the additional work we have provided in the Revised version has convinced all the Reviewers and reinforced the message of our manuscript. Once again, our sincere thanks to all Reviewers for their fruitful suggestions. We have now included in the new version of the text the modifications in apparent for addressing the last concerns of Reviewer #1.

Reviewer #1 (Remarks to the Author)

Overall, the authors have significantly improved the manuscript by adding a considerable amount of new data, as well as by clarifying the description of their approach and the results. They have addressed all my major concerns, and I only have minor concerns, as listed below.

We would like to thank the Reviewer for the positive assessment of our revised work. We are delighted that the modifications and additional data have convinced Reviewer#1.

1. Summary section: "Thus, efficient inhibition after treatment of YAP1-signaling pathway in the ECM-myCAF cluster is required to enhance CD8+ T-cell cytotoxicity"

Please modify this statement; change 'required to enhance' to eg. 'could enhance'. 'Require' would mean that there are no other ways to enhance CD9+ T cell cytotoxicity, which is basically impossible to prove.

As requested, in the new version of our manuscript, we have now changed the text by "could enhance CD8+ T-cell cytotoxicity", **p2**

2. Results section: "We also took advantage of the publicly available scRNAseq data from CAF isolated from untreated and chemotherapy-treated samples from an independent HGSOC cohort of patients treated at Helsinki's hospital 99..."

Even though not apparent from ref 99, these patients were actually treated at Turku University Hospital, Finland, so the cohort should for example "Turku cohort", and the sentence should state "...treated at Turku University Hospital, Finland,..."

As recommended by the Reviewer, we have now changed the text: "an independent HGSOC cohort of patients treated at Turku University Hospital", **p10**. Moreover, we have now called the cohort "Turku Cohort" all along the manuscript, in the new **Fig 3F**, **Fig. 3I**, in the new **Supplementary Fig. 3A** and in the corresponding **legends**.

3. There is a spatial transcriptomics data set from post-chemo HGSOC available from this preprint: Denisenko, Elena, Leanne de Kock, Adeline Tan, Aaron B. Beasley, Maria Beilin, Matthew E. Jones, Rui Hou, et al. 2022. "Spatial Transcriptomics Reveals Ovarian Cancer Subclones with Distinct Tumour Microenvironments." bioRxiv. <https://doi.org/10.1101/2022.08.29.505206>. The GeneExpression Omnibus (GEO) accession code is GSE211956. Nevertheless, the authors analyzed a considerable number of additional samples from their own cohort so that there is not absolute need to use this published data set for additional analyses.

We thank the Reviewer for the mention of this new manuscript in BioRxiv. As stated by the Reviewer, in the revised version of our manuscript, we have included 10 different spatial transcriptomic sections, that the Reviewer considered as a "considerable number of additional samples". Many thanks to the Reviewer for having considered that this was a huge job.

4. Please add a reference for the DoRothEA algorithm.

We thank the reviewer for this comment, and we are sorry for the missing reference. As requested, we have now added this reference in the **Results**' section, **p12**, as well as in the **Methods**' section (**#DoRothEA analyses**), **p30**.

5. Fig 5:H, J, N: CD8+ T lymphocytes co-occurrence probability ratio with CD8+ T lymphocytes, ANTXR1+ CAF and ANTXR1- CAF are shown from two samples; please also show the same plot across all the Visium analyzed samples; or a composite figure showing median and 95% interval across the samples.

We are sorry for not having been clear enough in our Revised version. The mentioned analyses do not concern *Visium* but are analyses made from IHC co-staining. We have now explained in more detail the analyses performed in this part of our manuscript to avoid any misunderstanding. This text has now been added **p13**.

Results p13: “To gain deeper insights into the spatial organization of CD8⁺ T lymphocytes and ANTXR1⁺ CAF at protein level and at single cell resolution, we performed multiplex imaging of ANTXR1, CD8 and cytokeratin proteins in HGSOE (Fig. 5E). After cell segmentation, we were able to classify each cell type (ANTXR1⁺ and ANTXR1⁻ CAF, CD8⁺ T lymphocytes and pan-cytokeratin⁺ cancer cells) based on intensities of their specific proteins using an unsupervised approach (Fig. 5F, G) (See also Methods #Pan-cytokeratin, CD8 and ANTXR1 multiplex immunofluorescence imaging and colocalization analysis). We next compared the spatial co-localization of ANTXR1⁺ and ANTXR1⁻ CAF with CD8⁺ T lymphocytes by applying a co-occurrence method implemented in Squidpy¹⁰⁵, which computes the probability of detecting a cluster of interest (here either ANTXR1⁺ or ANTXR1⁻ CAF) depending on the presence of another cluster (here CD8⁺ T lymphocytes) within an increasing radius. This spatial co-occurrence analysis of these multiplex imaging revealed that ANTXR1⁺ CAF were spatially distant from CD8⁺ T lymphocytes and did not colocalize together, while ANTXR1⁻ CAF were more likely to colocalize with CD8⁺ T cells (Fig. 5H).”

Methods, p24: “Pan-cytokeratin, CD8 and ANTXR1 multiplex immunofluorescence imaging and colocalization analysis”

6. Discussion: “In addition, we highlight the YAP1-mediated impact of the ECM-myCAF cluster on CD8⁺ T cell content following chemotherapy.”

=> Impact suggests functional evidence; however, functional validation is for CD8 cytotoxicity, while there is associative evidence for the content or infiltration. Please rephrase.

As suggested by the reviewer, we have now rephrased the sentence by: “In addition, we highlight the role of YAP1 expressed by the ECM-myCAF cluster on CD8⁺ T cell cytotoxicity following chemotherapy.” in the **Discussion, p18**.

7. Please rephrase the following sentences; their language needs improvement:

We thank the Reviewer for the careful reading of our manuscript and for these recommendations. We have now modified the text, as recommended:

- “confirming observations obtained by IHC and flow cytometry and validating that samples selected for spatial transcriptomic as representative”

p11: “We confirmed that the amount of stroma, as well as the number of CD3⁺ and CD8⁺ T lymphocytes, detected after chemotherapy were higher than in treatment-naïve (Supplementary Fig. S4E, F). These observations confirmed IHC and flow cytometry data and validated that the samples selected for spatial transcriptomic analyses are representative.”

- “In these sections, we identified the different CAF-S1 clusters and confirmed that, among them, the proportion of the ANTXR1⁺ ECM-myCAF cluster was this one, which accumulated before treatment and decreased the most after chemotherapy”

p11: “In these sections, we identified the different CAF-S1 clusters and confirmed that the ANTXR1⁺ ECM-myCAF cluster accumulated the most before treatment and decreased the most after chemotherapy (Fig. 3P).”

“We already observed that several TEAD-target genes were up-regulated in CAF-S1 compared to CAF-S4 in HGSOE.”

p12: “We previously observed that several TEAD-target genes are up-regulated in CAF-S1 compared to CAF-S4 in HGSOE.”

“Expression of some YAP1/TEAD-target genes was highly detected in CAF-S1 fibroblasts”

p12: “Several YAP1/TEAD-target genes were strongly expressed in CAF-S1 fibroblasts ^{40,80}, in particular in the ECM-myCAF cluster (**Supplementary Fig. S5A**)”

“suggesting that the loss of these CAF after treatment may explain -at least in part- the reduced YAP1/TEAD-signaling pathway.”

p13: “Thus, the reduced expression of YAP1 and TEAD-target genes after chemotherapy was concomitant to the decrease in the content of ANTXR1⁺ CAF, marker of ECM-myCAF, suggesting that depletion of this CAF-S1 cluster after treatment may explain -at least in part- this decrease in YAP1/TEAD-signaling pathway.”

Reviewer #2 (Remarks to the Author):

The authors have done a significant amount of additional work to address many of the previous reviewer comments and the manuscript is greatly improved.

We first would like to thank the Reviewer 2 for the positive evaluation of our revised manuscript and the additional experiments we performed. We are delighted that the Reviewer#2 is now fully convinced by our study.

I still have one comment, which was mentioned in my previous review but perhaps was not clear. This is that, since the authors show quite clearly that ECM-myCAFs and CD8⁺ T cells are spatially segregated within patient tumors, the rationale for experiments in which these two cell types are co-cultured is difficult to understand. Since these two cell types don't exist in close proximity to each other in vivo, the results seen when they are forced into proximity in an artificial in vitro assay may not be physiologically relevant. Perhaps there can be some statement added to the discussion to address this contradiction.

We would like to thank the Reviewer for this comment. We agree that ECM-myCAF and CD8⁺ T lymphocytes are rarely found in close proximity in HGSOc at baseline. However, following chemotherapy, we observed that CD8⁺ T cells accumulate within the stromal counterpart. This was observed by spatial transcriptomic, multiplex immunofluorescence imaging, and IHC analysis on HGSOc patient samples. All these spatial analyses confirmed single cell RNAseq, flow cytometry and functional assays. They are thus physiologically relevant as directly obtained from patient samples.

An additional comment is related to Supplemental Figure 6 - Characterization of detox-iCAF vs ECM-myCAF. Figure S6C seems to show differential expression between CAFs grown under two different culture conditions, not two different isolated CAF subtypes. Please clarify - how were the two cell types generated and cultured? Why is the gene expression focused around cells grown under different culture conditions?

As previously shown by several laboratories (Biffi, et al. 2019, Biffi et al. 2021, Nicolas, et al. 2022, Krishnamurty, et al. 2022), CAF exhibit a certain degree of plasticity according to culture conditions. Based on these previous observations, we established specific culture conditions to isolate both ECM-myCAF and iCAF primary fibroblasts from HGSOc and maintain them in culture. As indicated by the Reviewer, we have validated their identity by both flow cytometry and RNA sequencing. We have now modified the text in the **Methods**' section, **p32**, to clarify this point, as recommended.

p32: “Previous studies revealed that myCAF and iCAF exhibit a certain degree of plasticity according to culture conditions ^{50,57,88,110,120}. Based on these previous data, we cut fresh HGSOc samples from the operating room into small pieces and incubated them either on plastic dishes (Falcon, #353003) or on dishes coated with type I collagen at a final concentration of 9 µg/ml (Institut De Biotechnologie Jacques Boy, #207050357) to maintain ECM-myCAF and iCAF identities, respectively (see # below for validation of their identities).”

Reviewer #3 (Remarks to the Author):

I would like to congratulate the authors for the work they have done in addressing this reviewer's concerns. The inclusion of new data, especially the functional assays and the addition of spatially resolved samples, significantly enhances the strength and clarity of the manuscript's message. I fully support the acceptance of this study for publication.

We would like to express our sincere thanks to Reviewer#3 for the very favorable assessment of the additional data we provided and for our revised manuscript in general. Comments from the Reviewer have been helpful in the improvement of our manuscript. We are sincerely grateful for the time the Reviewer has taken in the evaluation of our work.